# Leveraging Flatness to Improve Information-Theoretic Generalization Bounds for SGD

**Ze Peng[†], Jian Zhang[†], Yisen Wang[‡], Lei Qi[◊], Yinghuan Shi[†*], Yang Gao[†]**

[†] State Key Laboratory for Novel Software Technology, Nanjing University
[‡] State Key Lab of General Artificial Intelligence,
 School of Intelligence Science and Technology, Peking University
[◊]School of Computer Science and Engineering, Southeast University
{pengze, zhangjian7369}@smail.nju.edu.cn, yisen.wang@pku.edu.cn,
qilei@seu.edu.cn, {syh, gaoy}@nju.edu.cn

## Abstract

Information-theoretic (IT) generalization bounds have been used to study the generalization of learning algorithms. These bounds are intrinsically data- and algorithm-dependent so that one can exploit the properties of data and algorithm to derive tighter bounds. However, we observe that although the flatness bias is crucial for SGD's generalization, these bounds fail to capture the improved generalization under better flatness and are also numerically loose. This is caused by the inadequate leverage of SGD's flatness bias in existing IT bounds. This paper derives a more flatness-leveraging IT bound for the flatness-favoring SGD. The bound indicates the learned models generalize better if the large-variance directions of the final weight covariance have small local curvatures in the loss landscape. Experiments on deep neural networks show our bound not only correctly reflects the better generalization when flatness is improved, but is also numerically much tighter. This is achieved by a flexible technique called "omniscient trajectory". When applied to Gradient Descent's minimax excess risk on convex-Lipschitz-Bounded problems, it improves representative IT bounds' $\Omega(1)$ rates to $O(1/\sqrt{n})$. It also implies a by-pass of memorization-generalization trade-offs. [1]

## 1 Introduction

Over-parameterized deep models trained by Stochastic Gradient Descent (SGD) are observed to generalize well, contradicting classic statistical learning theories that over-parameterization leads to overfitting (Vapnik et al., 1998; Bartlett & Mendelson, 2002; Zhang, 2002). The drawback of these theories is that they are too general and unable to leverage the specific properties of learning algorithms and data (Nagarajan & Kolter, 2019). Therefore, modern learning theories have turned to data- and algorithm-dependent bounds that leverage the properties of data and popular algorithms (*e.g.,* the limited hypothesis subset reached by SGD, various norms of matrix weights, low-rankness and sparsity of parameters or hidden representations, *etc.*) to derive tighter bounds specific to them (Brutzkus et al., 2018; Allen-Zhu et al., 2019; Arora et al., 2019; Cao & Gu, 2019; Neyshabur et al., 2019; Pesme et al., 2021; Muthukumar & Sulam, 2023; Alquier, 2023).

Recently, generalization bounds have been developed using information-theoretic measures (Hellström et al., 2024), because these measures are defined with the data distribution and the conditional distributions of the output hypothesis given the training data and are naturally data- and algorithm-dependent. Representative examples include the PAC-Bayesian bounds (McAllester, 1999) and the bounds using mutual information (MI) between the training data and the output of the algorithm (Russo & Zou, 2020; Xu & Raginsky, 2017). Thanks to the dependence, PAC-Bayesian approaches have led to the first non-vacuous numerical generalization bound for deep networks (Dziugaite & Roy, 2017), later tightened (Pérez-Ortiz et al., 2021) and scaled up (Zhou et al., 2019; Lotfi et al., 2022; 2024). MI bounds, the focus of this paper, can be seen as PAC-Bayesian bounds with optimal priors (Alquier, 2023). Tighter variants of the MI bound have been developed, *e.g.,* its chaining

---

[1]Codes are available at https://github.com/peng-ze/omniscient-bounds.

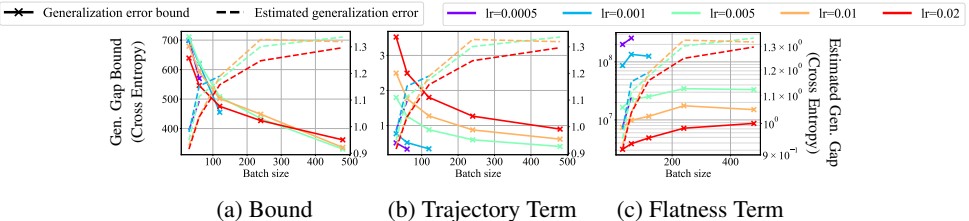

Figure 1: Wang & Mao (2022)'s bound for ResNet-18 on CIFAR-10 under varied flatness.

variants (Asadi et al., 2018), its individual-sample variants (Bu et al., 2020), the conditional MI framework (CMI) (Steinke & Zakynthinou, 2020), the evaluated variants (Harutyunyan et al., 2021; Hellström & Durisi, 2022; Wang & Mao, 2023), and bounds using other measures like rate-distortion (Sefidgaran et al., 2022) and Wasserstein distance (Wang et al., 2019). By bounding the measures for specific algorithms, they have been applied to SGLD (Pensia et al., 2018; Negrea et al., 2019; Wang et al., 2021b; Futami & Fujisawa, 2023), discretized SDE (Wang & Mao, 2024b), and SGD (Neu et al., 2021; Wang & Mao, 2022).

Another line of research on algorithmic properties finds SGD favors flat minima (Hochreiter & Schmidhuber, 1997; Keskar et al., 2017). Flat minima are minima at wide and flat basins of the loss landscape, and they are robust to loss landscape changes between the training and testing set. Consequently, flatness has been used to understand and improve the generalization of SGD-trained deep models (Achille & Soatto, 2018; Jiang et al., 2019; Foret et al., 2020; Cha et al., 2021; Zhao et al., 2022), where it is formulated with the Hessians $\hat{H}_S, H_\mu$ of empirical and population losses (Achille & Soatto, 2018; Orvieto et al., 2022). Therefore, as algorithm-dependent bounds, information-theoretic bounds for SGD should leverage the flatness as an important algorithmic property.

However, existing information-theoretic bounds for SGD do not adequately leverage the flatness bias. By controlling flatness through varying learning rate and batch size (Jastrzębski et al., 2018; He et al., 2019; Wu et al., 2022), we empirically observe how the generalization under varying flatness is captured by Wang & Mao (2022)'s bound (which is tighter than Neu et al. (2021)'s)

$$[\text{Generalization Error}] \leq \inf_{\sigma > 0} \sqrt{\frac{2R^2}{n\sigma^2} \underbrace{\sum_{t \in [T]} \mathbb{V}\left[g_t\right]}_{\text{Trajectory Term}} + \frac{\sigma^2}{2} \cdot \underbrace{T \cdot \operatorname{tr} \mathbb{E}\left[\hat{H}_S - H_\mu\right]}_{\text{Flatness Term}}},$$

with $g_t$ being the update at step $t$. As shown in Figure 1a, as batch size decreases, the actual generalization error decreases while Wang & Mao (2022)'s bound increases. That is, their result misaligns with the true generalization under varied flatness. The bound consists of two terms: the trajectory term and the flatness term. Figures 1b and 1c show the flatness term can capture the generalization to some extent while the trajectory term cannot, causing the misalignment. Unlike the flatness term that depends on Hessians, which are the explicit formulation of flatness, the trajectory term depends on gradient variance that is an implicit measure of flatness and requires extra conditions (*e.g.,* near-zero losses) to approximate the Hessian traces (Zhu et al., 2019; Martens, 2020; Feng et al., 2023). Therefore, we postulate it is the trajectory term's implicit dependence on flatness that causes the misalignment, and we intend to make the whole bound fully explicitly depend on and leverage flatness. To this end, Proposition 8 of Neu et al. (2021) (see Proposition 2) is a good start, which involves finer-grained optimization and depends on more algorithmic properties, potentially including flatness. It is proved by a technique of auxiliary trajectory that is constructed by randomly perturbing the SGD trajectory with *independent* Gaussian noises of covariance $\Sigma$. The bound optimizes $\Sigma$ as a parameter and takes the form

$$[\text{Generalization Error}] \leq \inf_{\Sigma} \mathbb{E}_W\left[f(W; \Sigma)\right],$$

*i.e.,* a mixture of an optimization over $\Sigma$ and an expectation over output weight $W$. By constructing $\Sigma$ from the Hessians of empirical and population losses, the bound will fully explicitly depend on the flatness. However, we find the bound still has two drawbacks: Firstly, it is suboptimal because the optimization is outside the expectation and cannot depend on or leverage the specific properties of $W$ instances. Secondly, accurate estimation is crucial for evaluating, comparing bounds and helping

algorithm design. Yet, the bound is not easy to estimate: It is similar to optimizing the "population risk" of "sample" $W$ over "hypothesis" $\Sigma$, which requires sampling some $W$ and minimizing the "empirical risk" (ERM). Since only a few $W$ can be sampled due to the computation and data cost of training deep models, the "ERM" is prone to "overfitting", *i.e.,* negative bias in estimation.

Both issues can be addressed by moving the optimization inside the expectation (*i.e.,* bounds like $\mathbb{E}_W \left[ \inf_\Sigma f(W; \Sigma) \right]$) so that the optimization depends on $W$: The inside optimization can leverage the specific properties (*e.g.,* population Hessian instance) of $W$, leading to a tighter bound; moreover, the empirical mean $\frac{1}{k} \sum_{i=1}^{k} \inf_{\Sigma^i} f(W^i; \Sigma^i)$ of the inside (instance-overfitting) optimization is an unbiased estimator and overfitting is no longer a source of negative bias in estimations. To implement this interchange, we find it is the independence of the Gaussian auxiliary perturbation that makes $\Sigma$ the same when conditioned on any $W$ instance, *i.e.,* $\Sigma$ cannot adapt to $W$ instances and unable to leverage their specific properties. Therefore, we extend the auxiliary trajectory technique by building it from an auxiliary perturbation that is no longer independent but at least depends on $W$. This is equivalent to moving the optimization inside (check $\inf_{\Sigma(w)} \mathbb{E} \left[ f(W; \Sigma(W)) \right] = \mathbb{E} \left[ \inf_\Sigma f(W; \Sigma) \right]$). Pushing this insight further, we make the perturbation depend on *all* random variables (*e.g.,* the training data) in the SGD training process (*e.g.,* to leverage the instance empirical Hessian that depends on training data), leading to the "omniscient trajectory"[2], which leads to tighter bounds if well optimized.

The technique yields a tighter generalization error bound for SGD that explicitly and fully depends on the flatness of the SGD output instance. Intuitively, it indicates the algorithm generalizes well when the output weights are flat and the flatness aligns with the covariance of output weights. Here, the covariance of output weights is computed after independently training multiple models, and the alignment means that the variance is low along sharp directions and high along flat directions, as illustrated in Figures 2b and 2c. As discussed in Section 3.4, our alignment better leverages the flatness compared to similar notions of alignments (Wang & Mao, 2024b; Wang et al., 2021a). The bound is evaluated on ResNet-18 trained by CIFAR-10. When varying batch size, it aligns well with the actual generalization error, indicating better exploitation of flatness. Moreover, our bound is numerically tight by being only a few percentages looser than the truth across hyperparameters.

Our omniscient trajectory technique has a simple nature, namely, interchanging the order of the expectation and optimization. Thanks to the simplicity, our technique can be flexibly combined with many existing techniques. Furthermore, recent works (Livni, 2024; Haghifam et al., 2023; Attias et al., 2024) have highlighted the limitations of information-theoretic bounds for having an $\Omega(1)$ lower-bound for GD or any accurate learners on some convex-Lipschitz-Bounded (CLB) problems. It can also be seen as an information-accuracy trade-off highlighting the complex relationship between memorization and generalization. We find our simple and flexible technique yields an $O(1/\sqrt{n})$ minimax rate for GD on CLB problems. Therefore, despite being simple, it provides asymptotic improvements and addresses a significant limitation of existing information-theoretic generalization theory. It also implies a by-pass of the trade-off: although accurate learners themselves memorize a lot, they are quite close to some oracle learners that memorize little.

Our contributions are summarized as follows: 1) We derive an information-theoretic generalization bound for SGD that better leverages its flatness bias and is numerically tighter; 2) our bound shows how the direction of flatness affects generalization; 3) we introduce a flexible omniscient trajectory technique that also 4) yields an $O(1/\sqrt{n})$ information-theoretic bound for GD on CLB problems.

## 2 PRELIMINARY

We first introduce basic notations. To present existing information-theoretic bounds for SGD and discuss important insights behind them for our (re)use in Section 2.3, we first introduce the algorithm of interest in Section 2.1 and the formulation and properties of flatness in Section 2.2. For $k \in \mathbb{N}^+$, let $[k] := \{1, 2, \ldots, k\}$. For sequence $a_0, a_1, \ldots,$ let $a_{l:r} := (a_i)_{i=l}^r$ and let $a_{-i}$ denote the rest after excluding $a_i$. For vector $x \in \mathbb{R}^k$ and matrix $A \in \mathbb{R}^{k \times k}$, let $\|x\| := \sqrt{x^\top x}$, and, with an abuse,

---

[2]The above anisotropic Gaussian noises with covariance $\Sigma$ are only an example of auxiliary trajectory and Theorem 1 generalizes the technique to more general auxiliary trajectories. Therefore, note the technique is about neither anisotropic noises (Neu et al., 2021; Wang & Mao, 2024b) nor general noises (Sefidgaran et al., 2022), but the auxiliary trajectory's extra dependence on all randomness and the instance-level optimization.

$\|x\|_A^2 := x^\top A x$. Random variables, realizations, and domains are denoted by capital, lowercase, and calligraphic letters, respectively. For example, $Z$ is a random sample, $\mathcal{Z}$ is the sample space, and $z$ is a realization. Let $\mu$ over $\mathcal{Z}$ be the sample distribution. Let $X'$ denote an I.I.D. copy of random variable $X$. Let $d$ be the number of model parameters and $\mathcal{W} \subseteq \mathbb{R}^d$ be the hypothesis space. If not specified otherwise, assume $\mathcal{W} = \mathbb{R}^d$. Let $S := (Z_1, \ldots, Z_n) \sim \mu^n$ be the training set of $n$ I.I.D. samples. A stochastic learning algorithm is formulated as conditional distribution $P_{W|S}$.

Let $\ell : \mathcal{W} \times \mathcal{Z} \to \mathbb{R}$ be a loss function. The ultimate goal of the learning algorithm is to optimize the *population risk* $\mathcal{L}_\mu(w, \ell) := \mathbb{E}_{Z \sim \mu}[\ell(w, Z)]$ over $w \in \mathcal{W}$. Since $\mu$ is not fully accessible, one uses *empirical risk minimization* (ERM) by sampling a training set $S \in \mathcal{Z}^n$ from $\mu$ and optimizing the *empirical risk* defined by $\hat{\mathcal{L}}_s(w, \ell) := \sum_{i=1}^{|s|} \ell(w, z_i)/|s|$. The difference in expectation over training sets and algorithmic randomness is the *(expected) generalization error (gap)* $\mathrm{gen}(\mu^n, P_{W|S}, \ell) := \mathbb{E}_{(S,W) \sim \mu^n \circ P_{W|S}} \left[ \mathcal{L}_\mu(W, \ell) - \hat{\mathcal{L}}_S(W, \ell) \right]$, where $\mu^n \circ P_{W|S}$ denotes the joint distribution determined by $\mu^n$ and $P_{W|S}$. We may omit the loss function $\ell$ when it is clear from the context. Let the loss difference under transnational perturbation $\gamma \in \mathbb{R}^d$ and Gaussian perturbation $\xi$ with covariance $\Sigma \in \mathbb{R}^{d \times d}$ be $\Delta_\gamma^\Sigma(w, s) := \mathbb{E}_{\xi \sim \mathcal{N}(0, \Sigma)} \left[ \hat{\mathcal{L}}_s(w + \gamma + \xi) - \hat{\mathcal{L}}_s(w) \right]$. We write $\sigma^2$ instead of $\Sigma$ in the superscript if $\Sigma = \sigma^2 I$, and omit $\gamma$ or $\Sigma$ if they are zero.

## 2.1 ITERATIVE STOCHASTIC ALGORITHMS

To facilitate analysis, we assume an abstract form of iterative stochastic algorithms to hide unnecessary details and improve generality. We assume the algorithm first prepares an independent random variable $V$ for internal randomness, then starts from an independent initial hypothesis $W_0 \in \mathcal{W}$ and updates the hypothesis iteratively for $T \in \mathbb{N}^+$ steps by $W_t := W_{t-1} - g_t(W_{t-1}, S, V, W_{0:t-2}) \in \mathcal{W}$ for $t \in [T]$, and finally outputs $W := W_T$. Here, $g_t$ is a deterministic function. The algorithm specifies a random process $W_{0:T}$, referred to as the *original trajectory*. We may omit $g_t$'s dependence on $(W_{t-1}, S, V, W_{0:t-2})$ to save space. SGD with batch size $b$ can be obtained by first generating the indices $B_{1:T} \in ([n]^b)^T$ for each mini-batch, then saving them in $V$ and finally letting $g_t$ compute the gradients in the mini-batch defined by $B_t$. Lastly, with access to past weights $W_{0:t-2}$ and all randomnesses $V$, $g_t$ can recover past gradients and covers momentum or Adam, *etc*. As a result, our theoretical results can be directly applied to these algorithms.

## 2.2 FLATNESS

The flatness at $w \in \mathcal{W}$ is formulated by the Hessians $\hat{H}_S(w)$ and $H_\mu(w)$ of the empirical and population losses, respectively. Empirically, the flatness of the empirical loss is highly anisotropic for deep models: after SGD training, *most* empirical Hessian eigenvectors have small eigenvalues while only the rest *few* have large eigenvalues (Sagun et al., 2018; Papyan, 2019). We distinguish these eigenvectors by "flat" versus "sharp" directions (Jastrzębski et al., 2019; Wu et al., 2022). Perturbations in the weight space along the sharp directions cause large loss changes, while those along the flat directions cause only slight loss changes.

## 2.3 INFORMATION-THEORETIC BOUNDS AND APPLICATION ON SGD

A random variable $X$ is $R$-sub-Gaussian if $\mathbb{E}\left[ e^{\lambda(X - \mathbb{E}[X])} \right] \leq e^{\lambda^2 R^2/2}$ for any $\lambda \in \mathbb{R}$. Let $I(A; B)$ be the mutual information (MI) between a pair of random variables $(A, B)$ (Cover & Thomas, 2006). The MI between the training set and the output weight can bound the generalization error:

**Lemma 1 (Xu & Raginsky (2017))** *Assume $\ell(w, \cdot)$ is $R$-sub-Gaussian on $\mu$ for any $w \in \mathcal{W}$. The generalization error of $P_{W|S}$ on $\mu$ is bounded by* $\mathrm{gen}(\mu^n, P_{W|S}) \leq \sqrt{2R^2 I(W; S)/n}$.

MI is hard to compute for SGD and one must bound it to apply Lemma 1. However, directly bounding MI on SGD is not enough because the MI of SGD can be infinite (Hellström et al., 2024), leading to a vacuous generalization bound. Neu et al. (2021) propose auxiliary trajectories to address this problem. Auxiliary trajectories are perturbed versions of the original trajectory, aiming to decrease the MI between the training set and the output weight. If well designed, the larger the perturbation is, the more the MI decreases. It is often easier to bound the MI of the auxiliary trajectory, which finally

becomes the "trajectory term" in the bound. Given the auxiliary trajectory, to transform the bound of it into a bound for the original trajectory, one must pay a penalty term of the loss difference between their output weights. The penalty comprises of several instances of $\Delta_\gamma^\Sigma$ terms, which are approximated by Taylor expansion to $\Delta_\gamma^\Sigma(w, s) \approx \mathbb{E}\left[\gamma^\top \nabla \hat{\mathcal{L}}_s(w) + (\gamma + \xi)^\top \times \hat{H}_s(w) \times (\gamma + \xi)/2\right]$. As a result, the penalty is related to the flatness: It will be very large if the perturbation $(\gamma + \xi)$ has large projections onto the top eigenvectors of $\hat{H}_s$, *i.e.,* the sharp directions, while it is safe to have large projections onto the flat directions. Trading off between the trajectory and penalty terms, we have Insight 1 for designing auxiliary perturbation and trajectories.

**Insight 1** *The ideal perturbation for the auxiliary trajectory should have large projections onto flat directions to reduce MI and the trajectory term while maintaining small projections onto sharp directions to keep the penalty term small.*

We now present the existing information-theoretic bounds for SGD with the help of auxiliary trajectories. [3] Neu et al. (2021) propose isotropic Gaussian as the perturbation for the auxiliary trajectory:

**Definition 1 (SGLD-Like Trajectory)** *The SGLD-like trajectory $\tilde{W}_{0:T}$ of any trajectory $\bar{W}_{0:T}$ is*

$$\tilde{W}_0 := \bar{W}_0, \quad \tilde{W}_t := \tilde{W}_{t-1} + (\bar{W}_t - \bar{W}_{t-1}) + N_t,$$

*where $N_t \sim \mathcal{N}(0, \sigma_t^2 I)$ is an* independent *isotropic Gaussian noise with $\sigma_t > 0$.*

After building the SGLD-like auxiliary trajectory of the original trajectory, Neu et al. (2021) exploit the properties of Gaussian noises to bound the MI of the SGLD-like output weight. Based on the same SGLD-like trajectory, Wang & Mao (2022) improve the technique for bounding MI, providing a numerically tighter result in Proposition 1.

**Proposition 1 (Theorem 2 of Wang & Mao (2022))** *SGD's generalization error is bounded by*

$$\text{gen}(\mu^n, P_{W|S}) \leq \sqrt{\underbrace{\frac{R^2}{n} \sum_{t=1}^T \frac{1}{\sigma_t^2} \mathbb{E}\left[\|g_t - \mathbb{E}[g_t]\|^2\right]}_{\text{MI bound (Trajectory Term)}} + \underbrace{\mathbb{E}\left[\Delta^{\Sigma_t \sigma_t^2}(W_T, S) - \Delta^{\Sigma_t \sigma_t^2}(W_T, S')\right]}_{\text{Penalty for the SGLD-like trajectory (Flatness Term)}}},$$

*where the expectations are over the randomness of training set sampling, initialization, and $V$.*

Since the trajectory term and the flatness term correspond to the MI bound and the penalty of the SGLD-like trajectory, respectively, these two terminologies will be used interchangeably. The SGLD-like trajectory decreases MI at the cost of the penalty term. The flatness is exploited to ensure the penalty is not very large. However, as shown in Figure 1, Proposition 1's trajectory term does not adequately exploit the flatness. This drawback is partially due to the isotropic Gaussian that adds perturbations of the same strength along all directions, violating Insight 1. To exploit the anisotropy of flatness, one can use non-isotropic Gaussian noises as in Proposition 2.

**Proposition 2 (Proposition 8 of Neu et al. (2021))** *SGD's generalization error is bounded by*

$$\text{gen}(\mu^n, P_{W|S}) \leq \inf_{\Sigma \in \mathcal{S}_+} \sqrt{\frac{R^2}{n} \mathbb{E}\left[\|W_T - \mathbb{E}[W_T]\|_{\Sigma^{-1}}^2\right]} + \mathbb{E}\left[\Delta^\Sigma(W, S) - \Delta^\Sigma(W, S')\right],$$

*where $\mathcal{S}_+ \subseteq \mathbb{R}^{d \times d}$ is the set of symmetric positive definite matrices.*

Here, $\Sigma$ is the covariance of the Gaussian. By setting, for example, $\Sigma = \mathbb{E}\left[\hat{H}_S\right]^{-1}$, the noise has large variances along the directions with small curvatures, better exploiting the flatness. Nevertheless, we find Proposition 2 still has room for improvement, as elaborated in Section 3.1.

---

[3]They are slightly modified to have simpler, similar and more comparable forms to each other. These modifications only make the bounds tighter, and will not cause unfair comparisons.

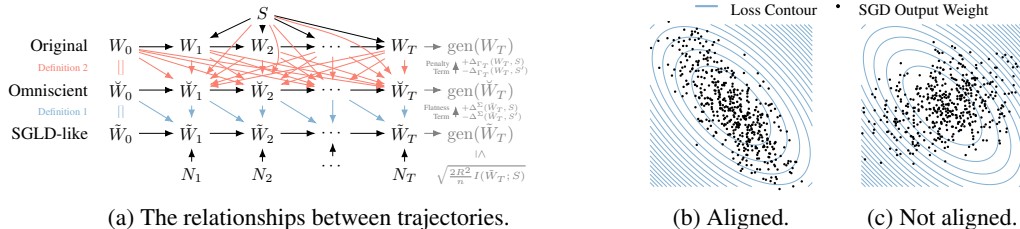

(a) The relationships between trajectories.          (b) Aligned.          (c) Not aligned.

Figure 2: The relationship between the original, omniscient, and SGLD-like trajectories, and illustrative examples of of alignment and misalignment between flatness and output weight covariance. The two trajectories have decoupled roles: the omniscient trajectory optimizes the whole bound while the SGLD-like trajectory bounds the MI for the omniscient trajectory.

## 3 PROPOSED OMNISCIENT TRAJECTORY

### 3.1 ANALYSIS ON EXISTING RESULTS' DRAWBACKS

To see the inefficiencies of Proposition 2, we recall a generic principle:

**Principle 1** *Being specific enhances optimization:* $\min_A \mathbb{E}_X [f(X; A)] \geq \mathbb{E}_X [\min_A f(X; A)]$.

Through the lens of this principle, Proposition 2 is closer to LHS of Principle 1 since $\Sigma$ to be optimized is shared among all instances in the expectation, leading to trade-offs between these instances. As a result, it is suboptimal compared to a fully specific and dependent one.

Moreover, Proposition 2 is prone to overfitting when estimated in numerical studies. Since the distribution of output weights is not fully accessible, one must sample some of them before optimizing over $\Sigma$. This procedure is identical to ERM, leading to potential overfitting to the sampled weights. Sampling output weights involves training multiple models on multiple training sets. The computational costs and data requirements of training deep models limit the number of samples. In contrast, $\mathcal{S}_+$ is a $\Theta(d^2)$-dimensional manifold. Therefore, the overfitting is severe at least in the classic view and the estimated bound has a severe negative bias. Although the overfitting may be benign as it is in deep learning, a meta-generalization theory is needed to ensure it, which is too complex and may even induce a meta-meta-overfitting problem with more parameters. To our knowledge, Proposition 2 lacks numerical results, possibly due to the large parameter count and the overfitting.

Some examples of the two drawbacks can be found in Appendix B.1. As indicated by Principle 1, both issues can be addressed by moving the optimization inside the expectation. Regarding the suboptimality, Principle 1 explicitly shows better tightness. Regarding overfitting, the RHS of Principle 1 is already the result of overfitting to instance trajectories, eliminating overfitting as a bias source. Furthermore, depending on more random variables, such as the training set, makes it easier to access and exploit the empirical flatness. Motivated by these benefits, we propose fully dependent auxiliary trajectories in the form of functions of *all* random variables in the training process. With knowledge of all random variables, the trajectories are termed "omniscient (auxiliary) trajectories".

### 3.2 OMNISCIENT TRAJECTORY

The omniscient trajectory that depends on all random variables in the training is defined as follows:

**Definition 2 (Omniscient Trajectory)** *The omniscient trajectory $\breve{W}_{0:T}$ of $(S, V, g_{1:T}, W_{0:T})$ specified by* omniscient perturbation $\Delta g_{1:T}$ *is given by*

$$\breve{W}_0 := W_0, \quad \breve{W}_t := \breve{W}_{t-1} - (g_t(W_{t-1}, S, V, W_{0:t-2}) - \Delta g_t(S, V, g_{1:T}, W_{0:T})),$$

*where $\Delta g_t$ is a deterministic function. Let $\Gamma_t := \sum_{\tau=1}^{t} \Delta g_\tau(S, V, g_{1:T}, W_{0:T})$, then $\breve{W}_t = W_t + \Gamma_t$.*

To bound the generalization error of the original SGD trajectory, we transform the generalization error of the original output weight into that of the omniscient output weight at the cost of a "penalty term" for the translational perturbation $\Gamma_T$. To bound generalization error and MI of the omniscient

output weight, we construct an SGLD-like auxiliary trajectory $\tilde{W}_{0:T}$ based on the omniscient trajectory and transform the generalization of the omniscient trajectory into that of the SGLD-like trajectory, at the cost of another penalty, *i.e.,* the flatness term. We then bound the generalization error of the SGLD-like output weight using the MI $I(\tilde{W}; S)$. The relationships between the trajectories and their generalization errors are summarized in Figure 2a. To bound the MI, we extend the key Lemma 4 of Wang & Mao (2022) that bounds the mutual information using variances in Lemma B.1 so that it can be used for the double-layered auxiliary trajectories despite its heavy dependence to $S$. Putting these problem transformations and bounds together, we obtain a generalization bound Theorem 1 using trajectory statistics similar to Proposition 1 but modified by the omniscient perturbation.

**Theorem 1** *Assume $R$-sub-Gaussianity for $\ell$. For any $\Delta g_{1:T}$ and any $\sigma_{1:T} \in \left(\mathbb{R}^{>0}\right)^T$, we have*

$$\text{gen}(\mu^n, P_{W|S}) \leq \sqrt{\frac{R^2}{n}\underbrace{\sum_{t=1}^T \frac{1}{\sigma_t^2}\mathbb{E}\left[\|g_t - \mathbb{E}[g_t] - \Delta g_t\|^2\right]}_{\text{MI bound (Trajectory term)}}} + \underbrace{\mathbb{E}\left[\Delta_{\Gamma_T}(W_T, S) - \Delta_{\Gamma_T}(W_T, S')\right]}_{\text{Penalty for the omni. trajectory (Penalty Term)}}$$

$$+ \underbrace{\mathbb{E}\left[\Delta^{\sum_t \sigma_t^2}(W_T + \Gamma_T, S) - \Delta^{\sum_t \sigma_t^2}(W_T + \Gamma_T, S')\right]}_{\text{Penalty for the SGLD-like trajectory (Flatness Term)}}.$$

Setting $\Delta g_t \equiv 0$ recovers Proposition 1. We further optimize it to tighten the bound.

### 3.3 OPTIMIZING THE OMNISCIENT TRAJECTORY AND EXPLOITING FLATNESS

To avoid optimizing over $T$ elements $\Delta g_{1:T}$, we first simplify Theorem 1 with $\Delta g_t = (g_t - \mathbb{E}[g_t]) - \frac{1}{T}\sum_{\tau=1}^T (g_\tau - \mathbb{E}[g_\tau]) + \frac{1}{T}\Delta G$. As a result, for any deterministic function $\Delta G$ of $(S, V, g_{1:T}, W_{0:T})$ and $\sigma > 0$, we have Corollary B.1 that can be informally stated as (with $\Delta W_t := W_t - W_0$)

$$\text{gen} \leq \sqrt{\frac{R^2}{n\sigma^2 T}\mathbb{E}\left[\|\Delta W_T - \mathbb{E}[\Delta W_T] + \Delta G\|^2\right]} + \mathbb{E}\left[\Delta_{\Delta G}^{\sigma^2 T}(W_T, S) - \Delta_{\Delta G}^{\sigma^2 T}(W_T, S')\right].$$

Appendix B.4 verifies such $\Delta g_t$ is optimal under the constraint $\Gamma_T = \Delta G$, and the generality is not harmed by this simplification. Setting $\Delta G = 0$ recovers the isotropic version of Proposition 2[4].

We then optimize over $\Delta G$. It can decrease the trajectory term by canceling $\Delta W_T - \mathbb{E}[\Delta W_T]$. However, large $\Delta G$ would increase the penalty term, leading to a trade-off. Fortunately, near flat minima, most directions are flat, and the penalty term is insensitive to perturbations along them. Following Insight 1, we confine $\Delta G$ to align with the flat directions. This is done by approximating the penalty terms to the second order, where Hessians emerge, and solving an optimization problem formed by the output weights and the Hessians (see Appendix B.6). Theorem 2 presents the result, where the three expectations correspond to the penalty, flatness, and trajectory terms, respectively.

**Theorem 2** *Assume $\ell(w, \cdot)$ is $R$-sub-Gaussian on $\mu$ for any $w \in \mathcal{W}$, $\ell(\cdot, z)$ and $\mathcal{L}_\mu(\cdot)$ are thirdly continuously differentiable w.r.t. $w$ for any $z \in \mathcal{Z}$, and there is a constant $D > 0$ such that $D\|\Delta w\|^4$ bounds the residuals in the third-order Taylor expansions of $\ell(\cdot, z)$ and $\mathcal{L}_\mu(\cdot)$ for any $z \in \mathcal{Z}$. Then for any $\lambda > 0$ trading-off between the trajectory and penalty terms, we have*

$$\text{gen}(\mu^n, P_{W|S}) \leq \mathbb{E}\left[\Delta_{\Delta G}(W_T, S) - \Delta_{\Delta G}(W_T, S')\right] + \frac{3}{2}\sqrt[3]{\frac{R^2}{n}\mathbb{E}\left[\text{tr}\left(\Delta H(W_T + \Delta G)\right)\right]}$$

$$\times \sqrt[3]{\mathbb{E}\left[\left\|(I - (I + \tilde{H}_{\text{pen}}/2\lambda C)^{-1})(\Delta W_T - \mathbb{E}[\Delta W_T]) - (2\lambda CI + \tilde{H}_{\text{pen}})^{-1}J\right\|^2\right]} + r, \quad (1)$$

*where $\Delta H(w) := \hat{H}_S(w) - H_\mu(w)$, $\tilde{H}_{\text{flat}}$ is chosen from $\Delta H(W_T)$ and $\hat{H}_S(W_T)$, $\tilde{H}_{\text{pen}}$ is chosen from $\Delta H(W_T)$ and $\hat{H}_S(W_T)$, and $J$ is chosen from $\nabla(\hat{\mathcal{L}}_S(W_T) - \mathcal{L}_\mu(W_T))$ and $\hat{\mathcal{L}}_S(W_T)$, $C := \frac{3}{2}\left(\frac{R^2}{n}|\text{tr}(\tilde{H}_{\text{flat}})|\right)^{1/3}$, $\Delta G = -(2\lambda CI + \tilde{H}_{\text{pen}})^{-1}(2\lambda C(\Delta W_T - \mathbb{E}[\Delta W_T]) + J)$, and $r = O(d^2\sigma_*^4)$ is the residual in a second-order approximation. See Eq. (B.6) in Appendix B.7 for the form of $\sigma_*$.*

---

[4]Using non-isotropic Gaussian noises for the SGLD-like trajectory fully recovers Proposition 2.

### 3.4 DISCUSSION AND COMPARISON

To see the dependence on the flatness more clearly, we approximate the result of Theorem 2. To simplify the approximation, *only for this subsection*, we assume $J \coloneqq \nabla \hat{\mathcal{L}}_S(W_T) - \nabla \mathcal{L}_\mu(W_T)$, and the model is sufficiently trained so that $\nabla \hat{\mathcal{L}}_S(W_T) \approx 0$. We also assume $\lambda$ is large enough so that $2\lambda C$ surpasses $\tilde{H}_{\text{pen}}$'s top singular value to approximate $I - (I + \tilde{H}_{\text{pen}}/2\lambda C)^{-1} \approx \tilde{H}_{\text{pen}}/2\lambda C$. Be noted that the above assumptions are used *only in this subsection, i.e.,* Theorem 2 and experiments in Section 4 do not require them. Results without some of these assumption can be found in Appendix B.8. Approximating the penalty term to the second order leads to the following result:

$$
\text{gen} \lessapprox \underbrace{\mathbb{E}\left[ \|\Delta W_T - \Delta \mathbb{E}[W_T]\|^2_{|\Delta H(W_T)| - \Delta H(\mathbb{E}[W_T])} + \|\nabla \mathcal{L}_\mu(W_T)\|^2_{\frac{|\Delta H(W_T)|}{4\lambda^2 C^2} - \frac{I}{2\lambda C}} \right]}_{\text{Corresponding to Penalty Term}}
$$

$$
+ \frac{3}{2} \sqrt[3]{\frac{2R^2}{n} \underbrace{\mathbb{E}\left[ |\text{tr}\left(\Delta H(W_T + \Delta G)\right)| \right]}_{\text{Corresponding to Flatness Term}} \cdot \underbrace{\mathbb{E}\left[ \|\Delta W_T - \mathbb{E}[\Delta W_T]\|^2_{\frac{\tilde{H}^2_{\text{pen}}}{4\lambda^2 C^2}} + \|\nabla \mathcal{L}_\mu(W_T)\|^2_{\frac{I}{4\lambda^2 C^2}} \right]}_{\text{Corresponding to Trajectory Term}}},
$$
(2)

where $|\cdot|$ replaces the eigenvalues of a matrix with their absolute values. The details can be found in Appendix B.8. It can be seen that all terms except the two population gradient norms depend on Hessians. Particularly, the "norms" of deviation $\Delta W_T - \mathbb{E}[\Delta W_T]$ are defined by the Hessians. As a result, the components of the deviation along flat directions contribute little to the bound. The flatter the minima are, the more and flatter the flat directions there are, and more components of the deviations contribute little to the bound. Moreover, if the flatness is aligned with the covariance, most large deviations can be found near the "flat subspace", leading to a smaller bound. As a result, generalization is better if minima are flat and the flatness aligns with the covariance.

This focus on alignment is similar to the work by Wang & Mao (2024b) and Wang et al. (2021a). These alignments are compared in Appendix B.8.1 in detail. Briefly, the main difference is that our alignment directly relies on flatness. Our technique is also similar to the rate-distortion bound (Sefidgaran et al., 2022) as both involve weight-dependent perturbations. However, our omniscient trajectory depends on more random variables, such as the training data that is crucial for leveraging the empirical Hessian. Besides, the existing chaining (Asadi et al., 2018) and evaluated (Harutyunyan et al., 2021; Hellström & Durisi, 2022; Wang & Mao, 2023) bounds, which can leverage the similarity between adjacent hypotheses, are potentially useful because the flatness of a minimum reflects a similarity with neighboring weights. However, chaining requires partitioning the hypothesis space, which is difficult for deep neural networks; the evaluated bounds directly rely on model losses, obscuring insights in the language of weights and flatness.

## 4 EXPERIMENTAL STUDY

In this section, we experimentally show how our bound captures the true generalization error/gap measured by Cross Entropy (0-1 loss does not have Hessians and is motivationally incompatible) compared to the existing bounds. We vary the hyperparameter and train 6 independent ResNet-18 models on CIFAR-10 at each hyperparameter. To ensure sub-Gaussianity, capped cross-entropy (CE) is used in testing and bound estimation, while vanilla CE is used for training for efficient training. We estimate Theorem 2 with $\lambda \in \{1, 10^3, 10^9\}$, $\tilde{H}_{\text{flat}} = \tilde{H}_{\text{pen}} = \hat{H}_S(W_T)$ to make estimation easier and $J = \nabla \hat{\mathcal{L}}_S(W_T) - \nabla \mathcal{L}_\mu(W_T)$ for numerical tightness as detailed in Appendix C.2. Estimating the bounds requires splitting datasets: The 6 models are trained by 6 random splits of the training set. Terms involving population statistics (*e.g.,* the population Hessians in the flatness terms and the population gradient in the penalty term) are approximated to the second order and estimated on validation sets. But for existing bounds, we assume $\mathcal{L}_\mu(W) \leq \mathbb{E}_{\xi \sim \mathcal{N}(0, \sigma^2 I)}[\mathcal{L}_\mu(W + \xi)]$ (Wang & Mao, 2022) and avoid the population Hessian. The true generalization error is estimated on a separate test set. See Appendix C for more details and the results with population Hessians.

We first evaluate whether our bound can better capture the generalization under varying flatness. As shown in Figure 3, the bound correctly captures the tendency of generalization error under varied batch size and learning rate. Specifically, the tendency of the trajectory term *w.r.t.* the batch size

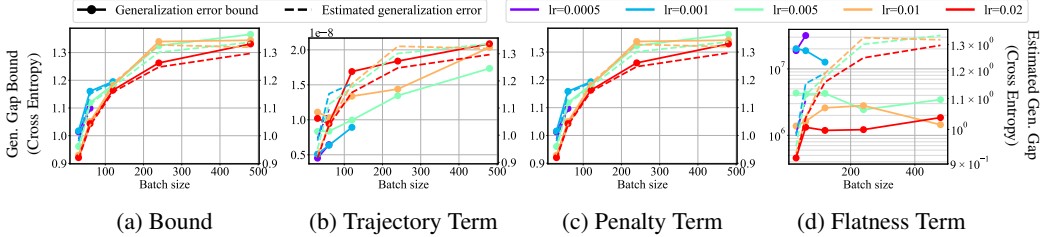

Figure 3: Thm. 2 ($\lambda = 10^9$) on CIFAR-10 and ResNet-18 with varied learning rate and batch size.

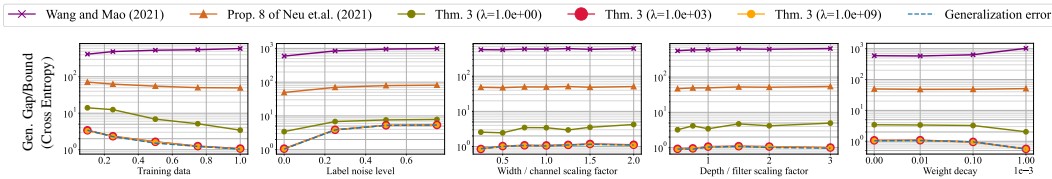

Figure 4: Numerical results for ResNet-18 on CIFAR-10 under varied training data usage, label noise level, width, depth, and weight decay. The isotropic version of Proposition 2 is used.

is corrected. The penalty term for the omniscient trajectory correlates well with the generalization error *w.r.t.* both learning rate and batch size, thanks to the Hessians in Eq. (2). The flatness term also generally has the correct tendency *w.r.t.* both hyperparameters. Unfortunately, the tendency of the trajectory term *w.r.t.* learning rate is still incorrect. As shown in Section 3.4, the trajectory term is essentially the product between Hessians and the variance of the last-step weight. We conjecture it is the increased variance when learning rate increases that overpowers the improved flatness and makes the tendency uncorrected. In contrast, the variance is less sensitive to batch size. Detailed discussion can be found in Appendix C.4.1. Nevertheless, after multiplied together, they contribute little to the bound and the uncorrected tendency does not affect the bound very much.

Figure 4 shows how our bound captures the generalization under varied generalization-critical hyperparameters. It can be seen that our bound with large $\lambda$ (*i.e.,* putting more weights on reducing the trajectory term) can well capture the generalization under these variations, while the some existing bounds fail to capture the improved generalization under stronger weight decay.

Results for MLP on MNIST can be found in Appendix C.4. According to Figures 3 and 4 and results in the appendix, our bounds are numerically much tighter. The main improvement comes from the trajectory term, as a result of both Proposition 2 and the optimized omniscient trajectory leveraging flatness. See Appendix C.4.1 for detailed discussions. In Appendix C.6, we evaluate our bounds under weight scaling, where the generalization is still well captured.

The trajectory term and the flatness almost diminish after multiplied together, and the bound mainly comprises of the penalty term. In Theorem 2 and Eq. (2), the term heavily relies on the population gradients. We consider this reliance to be the major limitation of our bound.

## 5 EXTENSIONS

Our technique is so flexible that it can be combined with existing techniques, such as the individual-sample technique and CMI framework. We can also combine it with the data-dependent prior technique of Negrea et al. (2019) for SGLD to yield an SGD bound. The combined results are listed in Appendix D. Notably, the omniscient trajectories in these results depend on all new random variables introduced by the variants. As a result, the omniscient trajectory can leverage the features of the variants and optimize more specifically.

Our technique can also address the known limitations of representative existing information-theoretic bounds on convex-Lipschitz-Bounded (CLB) problems in stochastic convex optimization (SCO). An SCO problem is a triplet $(\mathcal{W}, \mathcal{Z}, \ell)$, where $\mathcal{W}$ is convex and $\ell(\cdot, z)$ is convex given any $z \in \mathcal{Z}$. The CLB subclass $\mathcal{C}_{L,D}$ of SCO further requires $\mathcal{W}$ is closed and bounded with a diameter

$D$, while $\ell(\cdot, z)$ is $L$-Lipschitz given any $z \in \mathcal{Z}$. The goal is to minimize the *excess* (population) risk $\mathbb{E}_W [\mathcal{L}_\mu(W)] - \inf_{w \in \mathcal{W}} \mathcal{L}_\mu(w)$ under an unknown sample distribution, which can be bounded by the excess optimization error and the generalization error. The minimax excess risk reflects the worst-case generalization: $\inf_{P_{W|S} \in \mathcal{A}} \sup_{\mu \in \mathcal{M}_1(\mathcal{Z})} \mathbb{E}_{W \sim \mu^n \circ P_{W|S}} [\mathcal{L}_\mu(W)] - \inf_{w \in \mathcal{W}} \mathcal{L}_\mu(w)$, where $\mathcal{A}$ is a family of algorithms and $\mathcal{M}_1(\mathcal{Z})$ is the set of all distributions over $\mathcal{Z}$.

An $O(LD/\sqrt{n})$ minimax rate has been obtained for Gradient Descent (GD) using uniform stability. However, Haghifam et al. (2023) have shown there exist CLB instances where PAC-Bayesian bounds, the vanilla (C)MI bounds, their Gaussian-perturbed variants (Propositions 1 and 2 fall in this category if adapted to CLB settings), and representative variants all have an $\Omega(1)$ lower-bound, meaning these representative bounds cannot explain the generalization of GD on CLB problems. Attias et al. (2024) have shown any accurate CLB learner with low excess generalization risk must have high CMI, *i.e.,* the CMI-accuracy trade-off. Recently, Wang & Mao (2024a) have addressed the counterexample by Haghifam et al. (2023) and all CLB problems by augmenting CMI bounds with stability. However, a lot of complicated CMIs are involved and due to this complexity it is hard to intuitively interpret their relationship with the CMI-accuracy trade-off. Therefore, we try our technique to see whether it leads to a similar but simpler result. Combining individual-sample bounds with omniscient trajectory, we can easily resemble the stability argument:

**Theorem 3** *(Informal) For CLB problem $(\mathcal{W}, \mathcal{Z}, \ell)$ and distribution $\mu$ over $\mathcal{Z}$, we have*

$$\text{gen} \leq \inf_{\Delta G^{1:n}} \frac{1}{n} \sum_{i=1}^n \left( LD\sqrt{2I\left(W_T + \Delta G^i; Z_i \mid Z_{-i}\right)} + \mathbb{E}\left[\Delta_{\Delta G^i}(W_T, Z_i) - \Delta_{\Delta G^i}(W_T, S')\right] \right)$$

$$\overset{\substack{\text{(for stable algorithms)} \\ \leq}}{} \frac{2L}{n} \sum_{i=1}^n \mathbb{E}\left[\|W_T - \mathbb{E}\left[W_T \mid Z_{-i}\right]\|\right] \overset{\substack{\text{(for GD)} \\ \leq \\ \left(\text{with } T \text{ steps of size } \eta\right)}}{} 8L^2 \sqrt{T}\eta + 8L^2 T\eta/n$$

This bound recovers that of Bassily et al. (2020) up to a constant. Following similar steps of Haghifam et al. (2023), one can recover the best-achievable $O(LD/\sqrt{n})$ excess risk bound. Theorem 3 and its proof essentially state the original trajectory has a vanishing distance (measured by loss differences) to the omniscient trajectory of zero individual MI. Therefore, Theorem 3 provides an alternative to the trade-off on GD or stable algorithms: although the accurate learners have high CMI, they are quite close to auxiliary (oracle) learners with low MI[5]. Extending this intuition for more general accurate learners in Theorem 4 of Appendix D.5, we explore to replace CMI with a new information measure induced by our technique and break through the information-accuracy trade-off: this generalization-bounding new information measure vanishes together with risks. However, Theorem 4 requires vanishing excess optimization error and is still partial and preliminary. Lastly, when losses are smooth and convex, we also recover a stability-based $O(1/n)$ rate for SGD, as detailed in Appendix D.6.

## 6 Conclusion and Limitation

In this paper, we address the inadequate leverage of flatness in existing information-theoretic generalization bounds for SGD. By continuing the "being more specific" trend in generalization theory, we propose the omniscient trajectory that can be optimized depending on all random variables in the training process to better leverage the flatness. Our bound shows that an algorithm generalizes better if its output flatness aligns with its output covariance. When applied to deep neural networks, our bound aligns well with the generalization under varied flatness and is numerically tighter. When applied to CLB problems for GD, the technique yields an $O(1\sqrt{n})$ minimax rate, addressing a recently highlighted limitation of information-theoretic generalization theory.

However, our bound relies on population gradients and Hessians. Although tolerable for theories (*e.g.,* Neu et al. (2021) and Wang & Mao (2022; 2024b) also rely on population statistics), this problem prevents the bound from being a part of self-certified algorithms (Pérez-Ortiz et al., 2021). Moreover, the most information-theoretic components vanish in both the CLB bound and the experiments (see Eq. (D.5) and Figures 3c and C.4c). We conjecture these two issues may be addressed in future works by information-theoretically bounding the generalization of higher-order statistics. More detailed discussions on the limitations can be found in Appendix E.

---

[5]The auxiliary oracle learners $\mathbb{E}[W_T \mid \mathcal{Z}_{-i}]$ does not obey the MI-accuracy trade-off due to its access to $\mu$.

## REPRODUCIBILITY STATEMENT

All proofs and full versions of informally stated results can be found in the appendices for the corresponding sections. Details of experiments can be found in Appendix C. The codes for the experiments can be found in the supplementary material or at `https://github.com/peng-ze/omniscient-bounds`.

## ACKNOWLEDGMENTS

This work was supported by the NSFC Project (62192783, 92370129, 62222604, 62206052, 62376010), China Postdoctoral Science Foundation (2024M750424), Fundamental Research Funds for the Central Universities (020214380120, 020214380128), State Key Laboratory Fund (ZZKT2024A14), the Postdoctoral Fellowship Program of CPSF (GZC20240252), Jiangsu Funding Program for Excellent Postdoctoral Talent (2024ZB242), Jiangsu Science and Technology Major Project (BG2024031), and Beijing Nova Program (20230484344, 20240484642).

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

## A  Technical Lemmas

**Lemma A.1** *If $X, Y \in \mathbb{R}^d$ are independent random variables, then we have $\mathbb{E}\left[\|X - \mathbb{E}\left[Y\right]\|\right] \leq \mathbb{E}\left[\|X - Y\|\right].$*

**Proof** By the convexity of the $L_2$ norm and the independence between $X$ and $Y$, we have

$$\mathbb{E}\left[\|X - Y\|\right] = \mathbb{E}_X\left[\mathbb{E}_Y\left[\|X - Y\|\right]\right] \geq \mathbb{E}_X\left[\|\mathbb{E}_Y\left[X - Y\right]\|\right] = \mathbb{E}_X\left[\|X - \mathbb{E}\left[Y\right]\|\right].$$

∎

**Corollary A.1** *If $X_1, X_2 \in \mathbb{R}^d$ are I.I.D. copies of $X$, then we have $\mathbb{E}\left[\|X - \mathbb{E}\left[X\right]\|\right] \leq \mathbb{E}\left[\|X_1 - X_2\|\right].$*

**Lemma A.2** *If $X_1, X_2 \in \mathbb{R}^d$ are I.I.D. copies of $X$, then we have $\mathbb{E}\left[\|X - \mathbb{E}\left[X\right]\|^2\right] \leq \frac{1}{2}\mathbb{E}\left[\|X_1 - X_2\|^2\right].$*

**Proof**

$$\begin{aligned}
\mathbb{E}\left[\|X_1 - X_2\|^2\right] &= \mathbb{E}\left[\|X_1\|^2\right] - 2\mathbb{E}\left[X_1^\top X_2\right] + \mathbb{E}\left[\|X_2\|^2\right] \\
&= \mathbb{E}\left[\|X_1\|^2\right] - 2\mathbb{E}\left[X_1\right]^\top \mathbb{E}\left[X_2\right] + \mathbb{E}\left[\|X_2\|^2\right] \\
&= \mathbb{E}\left[\|X\|^2\right] - 2\mathbb{E}\left[X\right]^\top \mathbb{E}\left[X\right] + \mathbb{E}\left[\|X\|^2\right] \\
&= 2\mathbb{E}\left[\|X - \mathbb{E}\left[X\right]\|^2\right].
\end{aligned}$$

∎

**Lemma A.3** *Let $f : \mathcal{W} \to \mathbb{R}_{\geq 0}$ be a non-negative convex function. Let $W \in \mathcal{W}$ be a random variable, then we have*

$$\mathbb{E}\left[|f(\mathbb{E}\left[W\right]) - f(W)|\right] \leq 2\mathbb{E}\left[f(W)\right].$$

**Proof**

$$\begin{aligned}
\mathbb{E}\left[|f(\mathbb{E}\left[W\right]) - f(W)|\right] &\leq \mathbb{E}\left[|f(\mathbb{E}\left[W\right])|\right] + \mathbb{E}\left[|f(W)|\right] \\
&= \mathbb{E}\left[f(\mathbb{E}\left[W\right])\right] + \mathbb{E}\left[f(W)\right] && (f \geq 0) \\
&\leq 2\mathbb{E}\left[f(W)\right]. && \text{(convexity of } f)
\end{aligned}$$

∎

## B  Proofs and Details for Sec. 3 Proposed Omniscient Trajectory

### B.1  Details for Section 3.1

In this section, we present illustrative examples of suboptimality and overfitting of Proposition 2 mentioned in Section 3.1.

#### B.1.1  Suboptimality

Different $(S, W)$ has different empirical and population Hessians with different sharp directions. For instance, assume $v_1$ is the only sharp direction for $(s_1, w_1)$ and $v_2$ for $(s_2, w_2)$. Then the non-specific $\Sigma$ would apply small noises in both directions. If $v_1$ and $v_2$ are nearly orthogonal, the noises along $v_2$ and $v_1$ are insufficient for $(s_1, w_1)$ and $(s_2, w_2)$, respectively.

### B.1.2 OVERFITTING

When $k$ output weights are sampled, they lie in a $k$-dimensional subspace. As a result, noises along the remaining $d - k$ directions will not decrease the trajectory term. The optimal $\Sigma$ induces infinitely small noises along the remaining $d - k$ directions to decrease the penalty terms. Therefore, the optimal $\Sigma$ will have rank at most $k$ when $k$ samples are used. If one additional weight is sampled and it does not reside in the $k$-dimensional subspace, $\Sigma$ cannot reduce the variance along the $(k+1)$-th direction. As a result, the bound optimized for $k$ samples is suboptimal for the $k + 1$ samples, indicating overfitting.

### B.2 LEMMA B.1

Lemma B.1 extends Lemma 4 of Wang & Mao (2022) that bounds the mutual information using gradient variances.

**Lemma B.1** *Let $X, Y \in \mathbb{R}^{d_1}$, $\Delta \in \mathbb{R}^{d_2}$ and $O \in \mathbb{R}^{d_3}$ be arbitrary random variables. Let $N \sim \mathcal{N}(0, I)$ be a $d_1$-dimensional Gaussian noise that is independent of $(X, Y, \Delta, O)$. Then for any $\sigma > 0$, for any deterministic $\mathbb{R}^{d_1}$ function $f$ of $(X, Y, \Delta)$, we have for any function $\Omega$ solely of $(Y, O)$,*

$$I(f(X, Y, \Delta) + \sigma N; X \mid Y, O) \leq \frac{1}{2\sigma^2}\mathbb{E}\left[\|f(X, Y, \Delta) - \Omega(Y, O)\|^2\right], \qquad (B.1)$$

*and for any function $\Omega$ solely of $Y$,*

$$I(f(X, Y, \Delta) + \sigma N; X \mid Y) \leq \frac{1}{2\sigma^2}\mathbb{E}\left[\|f(X, Y, \Delta) - \Omega(Y)\|^2\right]. \qquad (B.2)$$

**Proof** We prove Eq. (B.1) first. Let $0_k \in \mathbb{R}^k$ denote $k$-dimensional zero vector. Define $X' := \begin{bmatrix} X^\top & 0_{d_2}^\top & 0_{d_3}^\top \end{bmatrix}^\top, Y' := \begin{bmatrix} Y^\top & 0_{d_2} & O^\top \end{bmatrix}^\top, \Delta' := \begin{bmatrix} 0_{d_1}^\top & \Delta^\top & 0_{d_3}^\top \end{bmatrix}^\top$ and $f' : \mathbb{R}^{d_1+d_2+d_3} \times \mathbb{R}^{d_1} \to \mathbb{R}^d$ and $\Omega' : \mathbb{R}^{d_1+d_2+d_3} \to \mathbb{R}^{d_1}$ by

$$f'(v, u) := f(u_{1:d_1}, v_{1:d_1}, v_{d_1+1:d_1+d_2}), \quad \Omega'(v) := \Omega(v_{1:d_1}, v_{d_1+d_2+1:d_1+d_2+d_3}),$$

where $0_d \in \mathbb{R}^d$ is the zero vector. By assumption, $(X', Y', \Delta')$ is independent of $N$. Consequently, by Lemma 4 of Wang & Mao (2022), we have

$$I(f'(Y' + \Delta', X') + \sigma N; X' \mid Y') \leq \frac{d}{2}\mathbb{E}\left[\log\left(\frac{\mathbb{E}\left[\|f'(Y' + \Delta', X') - \Omega'(Y')\| \mid Y'\right]}{d\sigma^2} + 1\right)\right]$$

$$\leq \frac{1}{2\sigma^2}\mathbb{E}\left[\|f'(Y' + \Delta', X') - \Omega'(Y')\|\right].$$

By construction, we have $f'(Y' + \Delta', X') = f(X, Y, \Delta)$ and $\Omega'(Y') = \Omega(Y, O)$. Combined with the definition of $Y'$, they lead to Eq. (B.2) of Lemma B.1.

To obtain Eq. (B.2), let $O''$ be the constant $0_{d_2}$ and $\Omega''(y, o'') := \Omega(y)$. Then by Eq. (B.1), we have

$$I(f(X, Y, \Delta) + \sigma N; X \mid Y) = I(f(X, Y, \Delta) + \sigma N; X \mid Y, O'')$$

$$\leq \frac{1}{2\sigma^2}\mathbb{E}\left[\|f(X, Y, \Delta) - \Omega''(Y, O'')\|^2\right]$$

$$= \frac{1}{2\sigma^2}\mathbb{E}\left[\|f(X, Y, \Delta) - \Omega(Y)\|^2\right],$$

which is exactly Eq. (B.2). ∎

B.3   PROOF OF THEOREM 1

By the technique of a change of trajectory, we first transform the generalization error of the original trajectory into that of the omniscient and SGLD-like trajectories at the cost of penalty terms:

$$
\begin{aligned}
&\mathrm{gen}(\mu^n, P_{W|S}) \\
:=&\mathbb{E}\left[\mathcal{L}_\mu(W) - \hat{\mathcal{L}}_S(W)\right] \\
=&\mathbb{E}\left[\mathcal{L}_\mu(\breve{W}_T) - \hat{\mathcal{L}}_S(\breve{W}_T)\right] + \mathbb{E}\left[\hat{\mathcal{L}}_S(\breve{W}_T) - \hat{\mathcal{L}}_S(W_T)\right] - \mathbb{E}\left[\mathcal{L}_\mu(\breve{W}_T) - \mathcal{L}_\mu(W_T)\right] \\
=&\mathbb{E}\left[\mathcal{L}_\mu(\tilde{W}_T) - \hat{\mathcal{L}}_S(\tilde{W}_T)\right] + \mathbb{E}\left[\hat{\mathcal{L}}_S(\tilde{W}_T) - \hat{\mathcal{L}}_S(\breve{W}_T)\right] - \mathbb{E}\left[\mathcal{L}_\mu(\tilde{W}_T) - \mathcal{L}_\mu(\breve{W}_T)\right] \\
&+ \mathbb{E}\left[\Delta_{\Gamma_T}(W_T, S) - \Delta_{\Gamma_T}(W_T, S')\right] \\
=&\mathbb{E}\left[\mathcal{L}_\mu(\tilde{W}_T) - \hat{\mathcal{L}}_S(\tilde{W}_T)\right] + \mathbb{E}\left[\Delta^{\sum_t \sigma_t^2}(W_T + \Gamma_T, S) - \Delta^{\sum_t \sigma_t^2}(W_T + \Gamma_T, S')\right] \\
&+ \mathbb{E}\left[\Delta_{\Gamma_T}(W_T, S) - \Delta_{\Gamma_T}(W_T, S')\right].
\end{aligned}
\tag{B.3}
$$

With the penalty terms matching those in Theorem 1, the goal reduces to bounding the generalization error of $\tilde{W}_T$. To this end, we apply Lemma 1 that bounds the error by mutual information and then decompose the mutual information using the chain rule. Since $\ell(w, \cdot)$ is $R$-sub-Gaussian by assumption, we have

$$
\mathbb{E}\left[\mathcal{L}_\mu(\tilde{W}_T) - \hat{\mathcal{L}}_S(\tilde{W}_T)\right] = \mathrm{gen}(\mu^n, P_{\tilde{W}_T|S}, \ell) \le \sqrt{\frac{2R^2}{n} I(\tilde{W}_T; S)},
$$

where

$$
\begin{aligned}
I(\tilde{W}_T; S) \le& I(\tilde{W}_{0:T}; S) && \text{(Data-processing inequality)} \\
=& I(\tilde{W}_0; S) + \sum_{t=1}^T I(\tilde{W}_t; S \mid \tilde{W}_{0:t-1}) && \text{(Chain rule of MI)} \\
=& \sum_{t=1}^T I(\tilde{W}_t; S \mid \tilde{W}_{0:t-1}) && (\tilde{W}_0 = W_0 \text{ is independent of } S) \\
=& \sum_{t=1}^T I(\tilde{W}_{t-1} - (g_t - \Delta g_t) + N_t; S \mid \tilde{W}_{0:t-1}). && \text{(Definition 1)}
\end{aligned}
$$

To bound the stepwise conditional mutual information, we apply Lemma B.1 with $X = S$, $Y = \tilde{W}_{0:t-1}$, $\Delta = (W_{0:T}, V)$ and

$$
f(X, Y, \Delta) = \tilde{W}_{t-1} - (g_t(W_{t-1}, S, V, W_{0:t-2}) - \Delta g_t(S, V, g_{1:T}, W_{0:T})).
$$

Since $(X, Y, \Delta)$ is a function of $(S, V, g_{1:T}, W_{0:T})$, which is independent with the SGLD noise $N_t \sim \mathcal{N}(0, \sigma^2 I)$, by Lemma B.1, we have

$$
\begin{aligned}
I(\tilde{W}_t; S \mid \tilde{W}_{0:t-1}) =& I(\tilde{W}_{t-1} - (g_t - \Delta g_t) + N_t; S \mid \tilde{W}_{0:t-1}) \\
\le& \frac{1}{2\sigma^2} \mathbb{E}\left[\left\|\tilde{W}_{t-1} - (g_t - \Delta g_t) - \Omega(\tilde{W}_{0:t-1})\right\|^2\right]
\end{aligned}
$$

for any function $\Omega$ of $Y = \tilde{W}_{0:t-1}$. To make things simpler by removing $\tilde{W}_{t-1}$ and minimize the expected squared norm, we set $\Omega(\tilde{w}_{0:t-1}) = \tilde{w}_{t-1} - \mathbb{E}[g_t]$ and obtain

$$
I(\tilde{W}_t; S \mid \tilde{W}_{0:t-1}) \le \frac{1}{2\sigma^2} \mathbb{E}\left[\|g_t - \mathbb{E}[g_t] - \Delta g_t\|^2\right].
\tag{B.4}
$$

Plugging Eq. (B.4) back to Eq. (B.3) leads to Theorem 1.

**Remark B.1** *The bound can be further tightened by setting $\Omega(\tilde{w}_{0:t-1}) = \tilde{w}_{t-1} - \mathbb{E}[g_t] + \mathbb{E}[\Delta g_t]$. However, this eventually leads to an extra $\mathbb{E}[\Delta G]$ in the trajectory terms of Corollary B.1. This extra term creates a "feedback" between instances that hinders optimization: when optimizing $\Delta G$, we must consider all instances of $\Delta G(S, V, g_{1:T}, W_{0:T})$ even if we are optimizing against a single instance of trajectory to form Theorem 2. For simplicity, we only set $\Omega(\tilde{w}_{0:t-1}) = \tilde{w}_{t-1} - \mathbb{E}[g_t]$.*

## B.4 SELECTION OF $\Delta g_t$ GIVEN $\Gamma_T$

In the result of Theorem 1, the penalty and flatness terms only rely on $\Gamma_T$ among omniscient-related variables. Therefore, we can first fix $\Gamma_T$ and optimize $\Delta g_{1:T}$.

Assume for this subsection that $\Gamma_T$ is fixed as $\Delta G$, a function of $(S, V, g_{1:T}, W_{0:T})$, and that $\Delta g_{1:T}$ is constrained to satisfy $\Delta G = \sum_{i=1}^{T} \Delta g_t$. Penalty and flatness terms are now fixed since they only rely on $\Gamma_T$ and we only need to optimize the trajectory term

$$\underset{\Delta g_{1:T}:\sum_t \Delta g_t = \Delta G}{\arg\min} \sqrt{\frac{R^2}{n} \sum_{t=1}^{T} \frac{1}{\sigma_t^2} \mathbb{E}\left[\|g_t - \mathbb{E}\left[g_t\right] - \Delta g_t\|^2\right]}$$

over $\Delta g_{1:T}$. We then set $\sigma_t = \sigma$, throw the square root and coefficients away, and move the summation inside the expectation, arriving at

$$\underset{\Delta g_{1:T}:\sum_t \Delta g_t = \Delta G}{\arg\min} \mathbb{E}\left[\sum_{t=1}^{T} \|g_t - \mathbb{E}\left[g_t\right] - \Delta g_t\|^2\right]$$

Set $\nu_t := g_t - \mathbb{E}\left[g_t\right]$. Since the constraints are instance-wise, we can also optimize $\Delta g_t$ instance-wisely:

$$\underset{\Delta g_{1:T}:\sum_t \Delta g_t = \Delta G}{\arg\min} \sum_{t=1}^{T} \|\nu_t - \Delta g_t\|^2 = T \cdot \mathbb{E}_{t \sim U[T]}\left[\|\nu_t - \Delta g_t\|^2\right] \geq T \cdot \left\|\mathbb{E}_{t \sim U[T]}\left[\nu_t - \Delta g_t\right]\right\|^2$$

$$= T \cdot \left\|\mathbb{E}_{t \sim U[T]}\left[\nu_t\right] - \frac{1}{T}\Delta G\right\|^2,$$

where inequality is due to the convexity of the squared norm and takes equality if and only if $\nu_t - \Delta g_t$ is constant (*w.r.t.* $t$), say $\Delta$. Then we have $\sum_\tau \Delta g_\tau = (\sum_\tau \nu_\tau) - T \cdot \Delta = \Delta G$, which implies $\Delta g_t = g_t - \mathbb{E}\left[g_t\right] - \frac{1}{T}\sum_\tau (g_\tau - \mathbb{E}\left[g_\tau\right]) + \frac{1}{T}\Delta G$.

## B.5 COROLLARY B.1

**Corollary B.1** *Assume $\ell(w, \cdot)$ is R-sub-Gaussian on $\mu$ for any $w \in \mathcal{W}$. For any deterministic function $\Delta G$ of $(S, V, g_{1:T}, W_{0:T})$ and any $\sigma > 0$, we have*

$$\mathrm{gen}(\mu^n, P_{W|S}) \leq \sqrt{\frac{R^2}{n\sigma^2} \frac{\mathbb{E}\left[\|\Delta W_T - \mathbb{E}\left[\Delta W_T\right] + \Delta G\|^2\right]}{T}} + \mathbb{E}\left[\Delta_{\Delta G}^{\sigma^2 T}(W_T, S) - \Delta_{\Delta G}^{\sigma^2 T}(W_T, S')\right],$$

*where $\Delta W_T := W_T - W_0$ is the change before and after the training. If we give up further optimization, i.e., $\Delta G = 0$, then we have*

$$\mathrm{gen}(\mu^n, P_{W|S}) \leq \sqrt{\frac{R^2}{nT\sigma^2} \mathbb{V}\left[\Delta W_T\right]} + \mathbb{E}\left[\Delta^{T\sigma^2}(W_T, S) - \Delta^{T\sigma^2}(W_T, S')\right],$$

*where $\mathbb{V}\left[X\right] := \mathbb{E}\left[\|X - \mathbb{E}\left[X\right]\|^2\right]$ denotes variance.*

**Proof** We prove Corollary B.1 by instantiating $\Delta g_t$ of Theorem 1.

Since $\Delta g_t$ depend on $g_{1:T}$, we can use $\nu_t := g_t - \mathbb{E}\left[g_t\right]$ and $\bar{\nu} = \frac{1}{T}\sum_t \nu_t$ when constructing $\Delta g_t$. Let $\Delta G$ be a deterministic function of $(S, V, g_{1:T}, W_{0:T})$ and $\sigma > 0$ be a constant. Then $\Delta g_t := (\nu_t - \bar{\nu}) + \frac{1}{T}\Delta G$ is indeed a deterministic function of $(S, V, g_{1:T}, W_{0:T})$.

With $\Gamma_T := \sum_t \Delta g_t = \sum_t \nu_t - T \cdot \bar{\nu} + \Delta G = \Delta G$ and $\sigma_t$ set to constant $\sigma$, by Theorem 1 we obtain

$$\mathrm{gen}(\mu^n, P_{W|S}) \leq \sqrt{\frac{R^2}{n\sigma^2} \sum_t \mathbb{E}\left[\left\|\nu_t - (\nu_t - \bar{\nu}) - \frac{1}{T}\Delta G\right\|^2\right]} + \mathbb{E}\left[\Delta_{\Delta G}(W_T, S) - \Delta_{\Delta G}(W_T, S')\right]$$

$$+ \mathbb{E}\left[\Delta^{T\sigma^2}(W_T + \Delta G, S) - \Delta^{T\sigma^2}(W_T + \Delta G, S')\right],$$

where

$$\sum_t \mathbb{E}\left[\left\|\nu_t - (\nu_t - \bar{\nu}) - \frac{1}{T}\Delta G\right\|^2\right] = \sum_t \mathbb{E}\left[\left\|\frac{1}{T}\sum_\tau (g_\tau - \mathbb{E}[g_\tau]) - \frac{1}{T}\Delta G\right\|^2\right]$$

$$= T \cdot \frac{1}{T^2}\mathbb{E}\left[\left\|\sum_t g_t - \mathbb{E}\left[\sum_t g_t\right] - \Delta G\right\|^2\right].$$

Since $W_T = W_0 - \sum_t g_t$, we have $\sum_t g_t = -\Delta W_T$ and

$$\sum_t \mathbb{E}\left[\left\|\nu_t - (\nu_t - \bar{\nu}) - \frac{1}{T}\Delta G\right\|^2\right] = \frac{1}{T}\mathbb{E}\left[\|\Delta W_T - \mathbb{E}[\Delta W_T] + \Delta G\|^2\right].$$

Putting everything together leads to Corollary B.1. ∎

## B.6 DETAILS FOR SECTION 3.3

Firstly, we need a explicit form of the penalty terms to balance the tradeoff. Therefore, the penalty terms are approximated to the second order in the surrogate optimization, where the Hessians at the output weight show up. The approximation is the same as Appendix B.7. Secondly, we assume the empirical and population Hessians in the penalty term for the SGLD-like trajectory does not change too much, namely $\hat{H}_S(W_T + \Delta G) - H_\mu(W_T + \Delta G) \approx \hat{H}_S(W_T) - H_\mu(W_T)$, to simplify the surrogate optimization. These simplifications lead to the following surrogate optimization target (be noted that we have not replacing terms with $\tilde{H}_{\text{flat}}, \tilde{H}_{\text{pen}}$ or $J$):

$$\min_{\Delta G} \frac{3}{2}\sqrt[3]{\frac{R^2}{n}\left|\mathbb{E}\left[\text{tr}\left(\hat{H}_S(W_T) - H_\mu(W_T)\right)\right]\right|\mathbb{E}\left[\|\Delta W_T - \mathbb{E}[\Delta W_T] + \Delta G\|^2\right]}$$

$$+ \mathbb{E}\left[\Delta G^\top \nabla(\hat{\mathcal{L}}_S(W_T) - \mathcal{L}_\mu(W_T)) + \frac{1}{2}\Delta G^\top(\hat{H}_S(W_T) - H_\mu(W_T))\Delta G\right],$$

where $\hat{H}_S(w)$ is the Hessian of the empirical loss at the trajectory terminal and $H_\mu(w)$ is that of the population loss. If we optimize the above surrogate target, we will rely on testing/validation sets to estimate the population gradient and Hessian. Therefore, one may want to partially avoid this reliance as much as possible by simply ignoring them. On the other hand, optimizing against validation sets leads to better numerical tightness. To accommodate both needs, we leave it as an option by letting $\tilde{H}_{\text{flat}} \in \left\{\Delta H(W_T), \hat{H}_S(W_T)\right\}, \tilde{H}_{\text{pen}} \in \left\{\Delta H(W_T), \hat{H}_S(W_T)\right\}$ and $J \in \nabla(\hat{\mathcal{L}}_S(W_T) - \mathcal{L}_\mu(W_T)), \nabla\hat{\mathcal{L}}_S(W_T)$ and replacing the empirical-population differences in the above surrogate losses. Another difficulty is that, one has to consider other independent runs when optimizing $\Delta G$ for one trajectory. To further simplify optimization, we modify the surrogate optimization target so that only one trajectory is considered:

$$\min_{\Delta G} \frac{3}{2}\sqrt[3]{\frac{R^2}{n}\left|\text{tr}\left(\tilde{H}_{\text{flat}}\right)\right|}\|\Delta W_T - \mathbb{E}[\Delta W_T] + \Delta G\|^2 + \Delta G^\top J + \frac{1}{2}\Delta G^\top \tilde{H}_{\text{pen}}\Delta G,$$

where $\Delta G$ is a function of the trajectory. To obtain a convex optimization problem, we partially remove the cubic root, giving

$$\min_{\Delta G} \lambda C\|\Delta W_T - \mathbb{E}[\Delta W_T] + \Delta G\|^2 + \Delta G^\top J + \frac{1}{2}\Delta G^\top \tilde{H}_{\text{pen}}\Delta G,$$

where $C := \frac{3}{2}\sqrt[3]{\frac{R^2}{n}\left|\text{tr}\left(\tilde{H}_{\text{flat}}\right)\right|}$. Bound parameter $\lambda$ here is used to compensate the "weight" change after twisting the target: When the terms under the cubic root become smaller, the derivative of the cubic root increases fast; but after the cubic root is removed, the derivative of the norm becomes smaller when the term becomes smaller. Therefore, we should use large $\lambda$ to emphasize the reduction of trajectory term. The convex optimization problem has a closed form solution

$$\Delta G = -\left(2\lambda C I + \tilde{H}_{\text{pen}}\right)^{-1}(2\lambda C(\Delta W_T - \mathbb{E}[\Delta W_T]) + J). \tag{B.5}$$

Since all variables in Eq. (B.5) is a function of weights, empirical and population gradients and Hessians, which are functions of weights, the training set and the population distribution, Eq. (B.5) indeed specifies a function of $(S, V, g_{1:T}, W_{0:T})$. This omniscient trajectory is used in Theorem 2.

## B.7 PROOF OF THEOREM 2

Based on Corollary B.1, Theorem 2 considers the specific omniscient trajectory defined by $\Delta G = -(2\lambda CI + \tilde{H}_{\text{pen}})^{-1}(2\lambda C(\Delta W_T - \mathbb{E}[\Delta W_T]) + J)$ as Eq. (B.5). We first optimize $\sigma$ given the specific $\Delta G$, for which we approximate the penalty for the SGLD-like trajectory to the third order. Specifically, for the empirical part we have

$$
\begin{aligned}
&\Delta^{\sigma^2 T}(W_T + \Delta G, S) \\
&:= \mathbb{E}_{\xi \sim \mathcal{N}(0, \sigma^2 TI)} \left[ \hat{\mathcal{L}}_S(W_T + \Delta G + \xi) - \hat{\mathcal{L}}_S(W_T + \Delta G) \right] \\
&= \mathbb{E}_\xi \left[ \xi^\top \nabla \hat{\mathcal{L}}_S(W_T + \Delta G) + \frac{1}{2} \xi^\top \hat{H}_S(W_T + \Delta G)\xi + \sum_{i \le j \le k} a_{i,j,k}\xi_i\xi_j\xi_j + r_0 \right] \\
&= \frac{\sigma^2 T}{2} \operatorname{tr}\left( \hat{H}_S(W_T + \Delta G) \right) + \mathbb{E}_\xi[r_0].
\end{aligned}
$$

where $r_0$ is the residual at the third order and the coefficients $a_{i,j,k}$ are independent $\xi$. By the assumption of Theorem 2, we have $|r_0| \le D\|\xi\|^4$ and $|\mathbb{E}_\xi[r_0]| \le D\mathbb{E}\left[\|\xi\|^4\right] = Dd(d+2)\sigma^4$. One can obtain similar results for the population loss difference, and combining the both we have

$$
\mathbb{E}\left[ \Delta^{\sigma^2 T}(W_T + \Delta G, S) - \Delta^{\sigma^2 T}(W_T + \Delta G, S') \right] = \frac{\sigma^2 T}{2} \operatorname{tr}\left( \hat{H}_S(W_T + \Delta G) - H_\mu(W_T + \Delta G) \right) + r,
$$

where $|r| \le Dd(d+2)\sigma^4 = O(d^2\sigma^4)$.

Similarly to Theorem 2 of Wang & Mao (2022), we optimize $\sigma > 0$ to balance

$$
\sqrt{ \frac{R^2}{n\sigma^2} \frac{\mathbb{E}\left[ \|\Delta W_T - \mathbb{E}[\Delta W_T] + \Delta G\|^2 \right]}{T} } =: \frac{A}{\sigma}.
$$

and

$$
\frac{\sigma^2 T}{2} \operatorname{tr}\left( \hat{H}_S(W_T + \Delta G) - H_\mu(W_T + \Delta G) \right) =: \sigma^2 B
$$

For $A, B > 0$, $A/\sigma + \sigma^2 B$ takes the minimum $3(A/2)^{2/3}B^{1/3}$ at

$$
\begin{aligned}
\sigma_* &= (A/2B)^{1/3} \\
&= \left( \frac{ \sqrt{ \frac{R^2}{n} \mathbb{E}\left[ \left\| (I - (I + \tilde{H}_{\text{pen}}/2\lambda C)^{-1})(W_T - \Delta W_T) - (2\lambda CI + \tilde{H}_{\text{pen}})^{-1}J \right\|^2 \right] } }{ T^{3/2} \operatorname{tr}(\Delta H(W_T + \Delta G)) } \right)^{1/3}.
\end{aligned}
$$

(B.6)

The proof is finished by setting $\sigma$ to this value and putting everything together.

## B.8 DETAILS FOR SECTION 3.4

We take $J := \nabla(\hat{\mathcal{L}}_S(W_T) - \mathcal{L}_\mu(W_T))$ and sufficiently large $\lambda > 0$ so that $\left| \lambda_1(\tilde{H}_{\text{pen}}/2\lambda C) \right| \ll 1$. We assume the training is sufficient so that $\nabla \hat{\mathcal{L}}_S(W_T) \approx 0$. We also assume that the initial weight $W_0$ is fixed.

By applying $\|x + y\|^2 \le 2\|x\|^2 + 2\|y\|^2$ and approximating the penalty for the omniscient trajectory to the second order, we obtain

$$\text{gen}(\mu^n, P_{W|S})$$

$$\lesssim \frac{3}{2}\sqrt[3]{\frac{2R^2}{n}\mathbb{E}\left[|\text{tr}\left(\Delta H(W_T + \Delta G)\right)|\right]\mathbb{E}\left[\|\Delta W_T - \mathbb{E}\left[\Delta W_T\right]\|^2_{(I-(I+\tilde{H}_{\text{pen}}/2\lambda C)^{-1})^2} + \|J\|^2_{(2\lambda CI + \tilde{H}_{\text{pen}})^{-2}}\right]}$$

$$+ \mathbb{E}\left[-2\lambda C(\nabla\hat{\mathcal{L}}_S(W_T) - \nabla\mathcal{L}_\mu(W_T))^\top (2\lambda CI + \tilde{H}_{\text{pen}})^{-1}(\Delta W_T - \mathbb{E}\left[\Delta W_T\right])\right]$$

$$+ \mathbb{E}\left[-(\nabla\hat{\mathcal{L}}_S(W_T) - \mathcal{L}_\mu(W_T))^\top (2\lambda CI + \tilde{H}_{\text{pen}})^{-1}J\right]$$

$$+ \mathbb{E}\left[\|\Delta W_T - \mathbb{E}\left[\Delta W_T\right]\|^2_{4\lambda^2 C^2(2\lambda CI + \tilde{H}_{\text{pen}})^{-1}|\Delta H|(2\lambda CI + \tilde{H}_{\text{pen}})^{-1}} + \|J\|^2_{(2\lambda CI + \tilde{H}_{\text{pen}})^{-1}|\Delta H|(2\lambda CI + \tilde{H}_{\text{pen}})^{-1}}\right].$$

By selection of $\lambda$, we have $(2\lambda CI + \tilde{H}_{\text{pen}})^{-1} \approx (2\lambda CI)^{-1} = I/2\lambda C$ and $I - (I + \tilde{H}_{\text{pen}}/2\lambda C)^{-1} = -\sum_{k=1}^{\infty}(-1)^k(\tilde{H}_{\text{pen}}/2\lambda C)^k \approx \tilde{H}_{\text{pen}}/2\lambda C$. These approximations lead to

$$\text{gen}(\mu^n, P_{W|S})$$

$$\lesssim \frac{3}{2}\sqrt[3]{\frac{2R^2}{n}\mathbb{E}\left[|\text{tr}\left(\Delta H(W_T + \Delta G)\right)|\right]\mathbb{E}\left[\|\Delta W_T - \mathbb{E}\left[\Delta W_T\right]\|^2_{\tilde{H}^2_{\text{pen}}/4\lambda^2 C^2} + \|J\|^2/4\lambda^2 C^2\right]}$$

$$+ \mathbb{E}\left[-(\nabla\hat{\mathcal{L}}_S(W_T) - \nabla\mathcal{L}_\mu(W_T))^\top(\Delta W_T - \mathbb{E}\left[\Delta W_T\right])\right]$$

$$+ \mathbb{E}\left[-(\nabla\hat{\mathcal{L}}_S(W_T) - \mathcal{L}_\mu(W_T))^\top J/2\lambda C\right]$$

$$+ \mathbb{E}\left[\|\Delta W_T - \mathbb{E}\left[\Delta W_T\right]\|^2_{|\Delta H|} + \|J\|^2_{|\Delta H|/4\lambda^2 C^2}\right]$$

$$= \frac{3}{2}\sqrt[3]{\frac{2R^2}{n}\mathbb{E}\left[|\text{tr}\left(\Delta H(W_T + \Delta G)\right)|\right]\mathbb{E}\left[\|\Delta W_T - \mathbb{E}\left[\Delta W_T\right]\|^2_{\tilde{H}^2_{\text{pen}}/4\lambda^2 C^2} + \|J\|^2/4\lambda^2 C^2\right]}$$

$$- \mathbb{E}\left[J^\top(\Delta W_T - \mathbb{E}\left[\Delta W_T\right])\right]$$

$$- \mathbb{E}\left[\|J\|^2/2\lambda C\right]$$

$$+ \mathbb{E}\left[\|\Delta W_T - \mathbb{E}\left[\Delta W_T\right]\|^2_{|\Delta H|} + \|J\|^2_{|\Delta H|/4\lambda^2 C^2}\right].$$

The second term $\mathbb{E}\left[\nabla\mathcal{L}_\mu(W_T)^\top(\Delta W_T - \mathbb{E}\left[\Delta W_T\right])\right]$ can also be transformed to Hessian-induced norms by exploiting the symmetry of the terminal deviation $\Delta W_T - \mathbb{E}\left[\Delta W_T\right]$. To this end, approximate the population gradient $\nabla\mathcal{L}_\mu(W_T)$ at the mean terminal weight $\mathbb{E}\left[W_T\right]$ by $\nabla\mathcal{L}_\mu(W_T) \approx \nabla\mathcal{L}_\mu(\mathbb{E}\left[W_T\right]) + H_\mu(\mathbb{E}\left[W_T\right])(W_T - \mathbb{E}\left[W_T\right])$. Then we have

$$\mathbb{E}\left[J^\top(\Delta W_T - \mathbb{E}\left[\Delta W_T\right])\right]$$

$$\approx \mathbb{E}\left[(\nabla\hat{\mathcal{L}}_S(\mathbb{E}\left[W_T\right]) - \nabla\mathcal{L}_\mu(\mathbb{E}\left[W_T\right]) + \Delta H(\mathbb{E}\left[W_T\right])(W_T - \mathbb{E}\left[W_T\right]))^\top(\Delta W_T - \mathbb{E}\left[\Delta W_T\right])\right]$$

$$= \mathbb{E}\left[\|\Delta W_T - \mathbb{E}\left[\Delta W_T\right]\|^2_{\Delta H(\mathbb{E}[W_T])}\right].$$

Plugging this back leads to

$$\text{gen}(\mu^n, P_{W|S})$$

$$\lesssim \frac{3}{2}\sqrt[3]{\frac{2R^2}{n}\mathbb{E}\left[|\text{tr}\left(\Delta H(W_T + \Delta G)\right)|\right]\mathbb{E}\left[\|\Delta W_T - \mathbb{E}\left[\Delta W_T\right]\|^2_{\tilde{H}^2_{\text{pen}}/4\lambda^2 C^2} + \|J\|^2/4\lambda^2 C^2\right]}$$

$$- \mathbb{E}\left[\|\Delta W_T - \mathbb{E}\left[\Delta W_T\right]\|^2_{\Delta H(\mathbb{E}[W_T])}\right]$$

$$- \mathbb{E}\left[\|J\|^2/2\lambda C\right] + \mathbb{E}\left[\|\Delta W_T - \mathbb{E}\left[\Delta W_T\right]\|^2_{|\Delta H|} + \|J\|^2_{|\Delta H|/4\lambda^2 C^2}\right].$$

The above results apply *without* the assumption that $\nabla\hat{\mathcal{L}}_S(W_T) \approx 0$.

Using the assumption $\nabla\hat{\mathcal{L}}_S(W_T) \approx 0$ leads to Eq. (2).

### B.8.1 Comparisons with Similar Alignments with Local Geometry

In Section 3.4, Theorem 2 is shown to connect with an alignment between terminal weight deviations/covariance and Hessians. This is similar to the work by Wang & Mao (2024b) and Wang et al. (2021a) on the alignment and the more fine-grained structure of local geometry.

Wang & Mao (2024b) provides two types of generalization bounds for discretized SDE: the trajectory-state bound based on the statistics along the full trajectories and terminal-state bound based on the statistics after the last update, which are separately compared below.

The trajectory-state bound focuses on the alignment between the population gradient noise covariance (GNC) $\Sigma_t^\mu(W_t) := \mathbb{V}_{Z' \sim \mu} [\nabla \ell(W_t, Z')]$ that depends on $W_t$ and the empirical GNC $C_t(W_t, S) := \frac{n-b}{b(n-1)} \mathbb{V}_{i \sim U[n]} [\nabla \ell(W_t, Z_i)]$ that depends on $(W_t, S)$. Therefore, this alignment and ours are both alignment between weight/gradient distribution and the local geometry. They also have the same dependence: the weight/gradient statistics are both $W_{t-1}$-dependent and the local geometric properties are both $(W_{t-1}, S)$-dependent. Differences mainly lie in the type of geometric properties used and how directly they reflect flatness. Wang & Mao (2024b)'s trajectory-state bound uses the inverted GNC $C_t^{-1}$ to capture local geometry, which is an indirect measure of flatness. By comparing Wang & Mao (2024b)'s Figure 2 and 3, one can see that their bound does not correlate correctly with Hessian traces even along training steps. In contrast, we directly use Hessians, allowing us to better leverage the flatness as required by our motivation. Their choice of $C_t^{-1}$ is a result of the discretized SDE algorithm instead of bound optimization ($\Sigma_t^\mu$ is the ingredient corresponding to bound optimization). Therefore, it is determined by the algorithm and it is difficult to switch to Hessians or other statistics. Another result of the above determining is that $C_t^{-1}$ can only reflect the flatness at step $t$ instead of the flatness of terminal weights. In contrast, our alignment only involves the flatness at terminal weights. According to the empirical results from Wang & Mao (2024b)'s Figure 3 and Jastrzębski et al. (2019)'s Figure 2, the eigenvalues of Hessians during the middle of training are much larger than those at terminal. Therefore, it is better to only use terminal flatness, as it is in our alignment. Regarding Wang & Mao (2024b)'s terminal-state bound, both this bound and our bound involve terminal states. Still, their bound uses the inverted weight covariance instead of Hessians to reflect local geometries.

Wang et al. (2021a) optimize the noise in SGLD, with a new information-theoretic generalization bound as a surrogate for real risks. They show the square root of expected GNC is greedily optimal under their bound and use this noise to improve the optimization and generalization of SGLD and closes the gap with SGD. Their results indicate the importance of direction and alignment of noises. If we put their algorithms and bounds together, then the result will be very similar with Wang & Mao (2024b)'s. Therefore, our alignment has similar differences with Wang et al. (2021a) as with Wang & Mao (2024b).

Moreover, our omniscient trajectory can be seen as a surrogate algorithm, and optimizing it is very similar to designing new algorithms with better generalization. As a result, additional similarities and differences can be found on the goal of introducing local geometry. In Wang et al. (2021a)'s work, before Theorem 1, the only use of local geometry is in Constraint 1 on not rising empirical risks. Therefore, it is used to guide how to increasing noises without sacrificing the empirical risks. In our bound, as discussed in Insight 1, we use local geometry to guide how to pull terminal weights together/closer without changing the losses too much. Therefore, local geometry in alignments has a similar role between Wang et al. (2021a)'s results and ours of guiding information reduction without sacrificing losses. Regarding differences, Wang et al. (2021a) decrease information measures by adding noises, while we do so by mapping things together or closer.

## C  Details for Sec. 4 Experimental Study

Codes are available in the supplementary material or at `https://github.com/peng-ze/omniscient-bounds`.

### C.1  Hessian-Related Details

In experiments, all Hessian traces are computed using PyHessian (Yao et al., 2020).

The optimized omniscient trajectory in Theorem 2 requires products between an Hessian-related inverse matrix $(I + \tilde{H}_{\text{pen}}/2\lambda C)^{-1}$ and vectors, say, $u$. Since Hessians of deep models are too large to store and compute with, it is ubiquitous to directly compute the inverse-vector product (iVP). For iVP, following Dagréou et al. (2024), we use conjugate gradient method that only requires the product $(I + \tilde{H}_{\text{pen}}/2\lambda C)v$ of the original matrix with an arbitrary vector $v$, which is supplied with PyHessian's hv_product. More specifically, we approximate $v$ such that $(I + \tilde{H}_{\text{pen}}/2\lambda C)v = u$ to obtain a $\hat{v}$. We run conjugate gradient method until the error $\left\| (I + \tilde{H}_{\text{pen}}/2\lambda C)\hat{v} - u \right\|^2$ is less than 1% of $\|u\|^2$, or the maximum iteration 20 is reached.

## C.2 BOUND ESTIMATION DETAILS

This section presents the details in estimating Eq. (1) of Theorem 2. For the randomness in weights, we train $k > 1$ models independently to obtain weights $W_T^{1:k}$. To ensure the training data for each model is I.I.D. sampled, we assume the whole training set is sampled in an I.I.D. manner, randomly partition the whole training set into $k$ subsets $\left\{ S^i \right\}_{i=1}^k$ and use $S^i$ to train $W_{0:T}^i$. For fair comparison, we then make unbiased or positively biased estimates for expectations in Eq. (1) separately and then compute the bound using these estimates. Being concave, the cubic root will bring negative bias. Nevertheless, this negative bias can also be found in baseline existing bounds. Moreover, in experiments, the expectation product under the cubic root is extremely small. Therefore, this negative bias will not influence the overall result too much and we leave it as it is for simplicity.

When optimizing the bound to obtain $\Delta G$, we use a validation set $S''$ to estimate $J$.

For the penalty terms, i.e., $\mathbb{E}\left[\Delta_{\Delta G}(W_T, S) - \Delta_{\Delta G}(W_T, S')\right]$ and $\mathbb{E}\left[\text{tr}\left(\Delta H(W_T + \Delta G)\right)\right]$, we unbiasedly estimate them with the test set. The same test set is used for all weights to fully use the data. The loss differences are directly computed by modifying the parameters and forwarding the testing data $S'$ instead of approximating them.

For the flatness-optimized trajectory term, the estimate is more complicated, since we must also estimate $\mathbb{E}\left[\Delta W_T\right]$ within the estimate and handle the matrix inverses and the population gradients in $J$.

To estimate $\mathbb{E}\left[\Delta W_T\right]$ in expectations, one should use samples of $W_T$ that are independent of $W_T^i$. However, another draw of $k$ weights results in further partitioning on the training set, and the data used for each weight is much less. To fully use the data, similarly to cross validation, we estimate the inner expectation by $\Delta \bar{W}_T^{-i} := \frac{1}{k-1} \sum_{i' \in [k]}^{i' \neq i} \Delta W_T^{i'}$, which is independent of $W_T^i$.

To avoid the population Hessian in the inverses and make estimation easier, we select $\tilde{H}_{\text{pen}} := \hat{H}_S(W_T)$. In this way, the inverses solely depend on the training data and the terminal weight, and their interaction with the testing set is only linear, which is much more friendly to expectations. Define $E := I - (I + \tilde{H}_{\text{pen}}/2\lambda C)^{-1}$ and $F := (2\lambda C I + \tilde{H}_{\text{pen}})^{-1}$, which only depend on $(S, W_T)$. Initial experiments on CIFAR-10 indicate that it is necessary to include the population gradients for numerical tightness, and we set $J := \nabla \hat{\mathcal{L}}_S(W_T) - \nabla \mathcal{L}_\mu(W_T)$. Under these specific settings, we will estimate

$$\mathbb{E}\left[ \left\| E(\Delta W_T - \mathbb{E}\left[\Delta W_T\right]) - F(\nabla \hat{\mathcal{L}}_S(W_T) - \nabla \mathcal{L}_\mu(W_T)) \right\|^2 \right] \tag{C.1}$$

Replacing the expectation with the empirical mean and the random variables with I.I.D. copies, we obtain an estimator of the trajectory term

$$\frac{1}{k} \sum_{i=1}^k \left\| E^i(\Delta W_T^i - \Delta \bar{W}_T^{-i}) - F^i(\nabla \hat{\mathcal{L}}_{S^i}(W_T^i) - \nabla \mathcal{L}_{S''}(W_T^i)) \right\|^2,$$

which is positively biased because

$$\mathbb{E}\left[\frac{1}{k}\sum_{i=1}^{k}\left\|E^i(\Delta W_T^i - \Delta\bar{W}_T^{-i}) - F^i(\nabla\hat{\mathcal{L}}_{S^i}(W_T^i) - \nabla\mathcal{L}_{S''}(W_T^i))\right\|^2\right]$$

$$=\frac{1}{k}\sum_{i=1}^{k}\mathbb{E}_{S^i,W_T^i}\left[\mathbb{E}\left[\left\|E^i(\Delta W_T^i - \Delta\bar{W}_T^{-i}) - F^i(\nabla\hat{\mathcal{L}}_{S^i}(W_T^i) - \nabla\mathcal{L}_{S''}(W_T^i))\right\|^2 \mid S^i, W_T^i\right]\right]$$

$$\geq\frac{1}{k}\sum_{i=1}^{k}\mathbb{E}_{S^i,W_T^i}\left[\left\|\mathbb{E}\left[E^i(\Delta W_T^i - \Delta\bar{W}_T^{-i}) - F^i(\nabla\hat{\mathcal{L}}_{S^i}(W_T^i) - \nabla\mathcal{L}_{S''}(W_T^i)) \mid S^i, W_T^i\right]\right\|^2\right]$$

$$=\frac{1}{k}\sum_{i=1}^{k}\mathbb{E}_{S^i,W_T^i}\left[\left\|E^i(\Delta W_T^i - \mathbb{E}\left[\Delta\bar{W}_T^{-i} \mid S^i, W_T^i\right]) - F^i(\nabla\hat{\mathcal{L}}_{S^i}(W_T^i) - \mathbb{E}\left[\nabla\mathcal{L}_{S''}(W_T^i) \mid S^i, W_T^i\right])\right\|^2\right]$$

$$=\frac{1}{k}\sum_{i=1}^{k}\mathbb{E}_{S^i,W_T^i}\left[\left\|E^i(\Delta W_T^i - \mathbb{E}\left[\Delta W_T\right]) - F^i(\nabla\hat{\mathcal{L}}_{S^i}(W_T^i) - \nabla\mathcal{L}_{\mu}(W_T^i))\right\|^2\right]$$

$$=\mathbb{E}\left[\left\|E(\Delta W_T - \mathbb{E}\left[\Delta W_T\right]) - F(\nabla\hat{\mathcal{L}}_{S}(W_T) - \nabla\mathcal{L}_{\mu}(W_T))\right\|^2\right],$$

where the second step is because $\mathbb{E}\left[X^\top X\right] \geq \mathbb{E}\left[X\right]^\top \mathbb{E}\left[X\right]$.

Among the above estimations, the most loose step comes from the last estimation, *i.e.,* the estimation on the trajectory term, because other expectations are estimated unbiasedly and this expectation is positively biased. In principle, we can estimate it without bias. To see this, set $A \coloneqq E\Delta W_T - F(\nabla\hat{\mathcal{L}}_S(W_T) - \nabla\mathcal{L}_\mu(W_T))$ expand Eq. (C.1) as

$$\mathbb{E}\left[\left\|E(\Delta W_T - \mathbb{E}\left[\Delta W_T\right]) - F(\nabla\hat{\mathcal{L}}_S(W_T) - \nabla\mathcal{L}_\mu(W_T))\right\|^2\right]$$

$$=\mathbb{E}\left[\|A - E \times \mathbb{E}\left[\Delta W_T'\right]\|^2\right]$$

$$=\mathbb{E}\left[\|A\|^2\right] - 2\mathbb{E}\left[\mathbb{E}\left[\Delta W_T'\right]^\top E^\top A\right] + \mathbb{E}\left[\|E \times \mathbb{E}\left[\Delta W_T'\right]\|^2\right]$$

$$=\mathbb{E}\left[\|A\|^2\right] - 2\mathbb{E}\left[\Delta W_T'\right]^\top \mathbb{E}\left[E^\top A\right] + \mathbb{E}\left[\|E \times \mathbb{E}\left[\Delta W_T'\right]\|^2\right],$$

which can be term by term estimated without bias. However, since $E$ involves Hessian inversion, this unbiased estimator requires more Hessian iVPs (iHVP): Computing $A$ requires two iHVPs, while $E^\top A$ and $E \times \mathbb{E}\left[\Delta W_T\right]$ require another two iHVPs. As a result, the unbiased estimator requires $4k$ iHVPs. In contrast, the positively biased estimate only requires $2k$ iHVPs. Since IVPs, especially iHVPs, are extremely time-consuming, we still use the *positively biased* estimator in the experiments.

## C.3 Training Details

For both MNIST and CIFAR-10, we train $k = 6$ independent models from the *same* randomly chosen initial weight, which means each model receives $10,000$ training samples. We use 2-layer MLPs for MNIST as Wang & Mao (2022) while ResNet-18 for CIFAR-10. Since a bounded loss is naturally sub-Gaussian (Xu & Raginsky, 2017), we employ cross-entropy (CE) losses capped at $12\log c$ for both testing and bound estimation, where $c$ represents the class number. Here, $\log c$ corresponds to the expected cross-entropy at random initialization, which are heuristically the maximum of the average loss during the training. However, we still use vanilla CE losses for training. As Wang & Mao (2022), we use SGD with momentum of $0.9$. The use of momentum does to violate the assumptions in Section 2.1 because each update can access history weights to recover gradients and compute momentum. ResNet-18 models are trained for 200 epochs while 2-layer MLP models are trained for 500 epochs as Wang & Mao (2022). We start from a base hyperparameter, where the learning rate is $0.01$, the batch size is $60$, and no dropout or weight decay is used. For the 2-layer MLP, the base hidden width is $512$. To enhance the generalization of the networks on CIFAR-10 for evaluating the bounds, we use random horizontal flip and random resized crop. This use of data

augmentations does not violate the assumptions in Section 2.1, as they can be achieved by passing in $V$ the random seeds for the augmentations and letting $g_t$ augment the samples using these seeds before computing the gradients.

For single-hyperparameter variations in Figures 4 and C.5, we vary that single hyperparameter while keeping others the same as the base. For Figures 3 and C.4, we vary the learning rate and the batch size in a grid-search manner while keeping others the same as the base. When the learning rate is too small or too large, or the batch size is too large, the model even cannot fit the training data well. Therefore, we exclude the training records whose final weight has a training accuracy less than $95\%$.

Models are trained on 12 NVidia RTX4090D GPUs for 2 day with auto mixed precision (BF16) and `torch.compile()` to save memory and increase parallelization. To ensure consistency of model and training details with previous results, our codes are modified based on Wang & Mao (2022) with bound estimation modules totally re-implemented.

We must remark that the test accuracy of our models on CIFAR-10 is relatively low, because each model is trained by a one-sixth subset of CIFAR-10.

## C.4 MORE EXPERIMENTAL RESULTS

### C.4.1 MORE RESULTS FOR RESNET-18 ON CIFAR-10 AND MORE DISCUSSIONS

Here, we display more results of existing bounds for ResNet-18 on CIFAR-10. Figure C.1 displays the result of the isotropic Proposition 2 of Neu et al. (2021).

From Figure C.1, we find that the trajectory term of isotropic Proposition 2 is insensitive to batch size, which is an improvement over the incorrect tendency of Proposition 1. However, this improvement is only partial compared to our trajectory term that decreases as the batch size decreases. Thanks to the flatness term that has the correct tendency *w.r.t.* batch size under large learning rates, the bound also has the correct tendency *w.r.t.* batch size if learning rate is large enough. Compared to this result, our bound can even better capture the generalization under all learning rates by having a more aligned trajectory term.

Now we focus on the tendency *w.r.t.* learning rate. Our bound fails to fully correct the wrong tendency of the trajectory term of Propositions 1 and 2 *w.r.t.* learning rate. Nevertheless, the wrong tendency seems partially corrected because the lines corresponding to the trajectory terms under different learning rates are well separated in Figures 1b and C.1b of the existing bounds, while some of those from our bounds twist together in Figure 3b. However, this partial correction is not enough to invert the wrong tendency, unlike the tendency *w.r.t.* batch size. We conjecture it is because the trajectory term is not solely determined by flatness, but is a "product" between the Hessians and the output weight variance (see Eq. (2)). Increasing the learning rate increases the step size, which then increases the variance of the output weight. This variance increase is empirically confirmed by Figure C.1b: we display the trajectory term with $\sigma^{-1}$ excluded in this figure. As a result, what is displayed in this figure is proportional to the output weight variance. It clearly shows that the variance increases as the learning rate increases. Therefore, there is a competition between the increased variance and the improved flatness. Unfortunately, it seems the increased variance overpowers the improved flatness in this process and our bound fails to invert the wrong tendency *w.r.t.* learning rate. On the other hand, decreasing the batch size seems to also increase the variance and we must explain why the tendency *w.r.t.* batch size is fixed. The reasons is that, the variance is, in fact, very *in*sensitive to batch size: From Figure C.1b, we can see that the variance is almost fixed when batch size changes. As a result, the trajectory term is dominated by the improved flatness when the batch size decreases.

Regarding the tightness improvement (mainly due to the trajectory term), Proposition 2 has a much smaller trajectory term than Proposition 1 ($\sim 10^{-2}$ v.s. $\sim 10^{0}$). It mainly happens because gradients of SGD are noisy and gradients from different steps often partially cancel. In Proposition 1, the canceled parts are still counted in the trajectory term while in Proposition 2 where gradients are summed together before taking norms. The components canceled in the trajectory indeed cancel in the trajectory term and do not contribute to the bound. This improvement is pushed further by the omniscient trajectory and the optimization under flatness ($\sim 10^{-2}$ v.s. $\sim 10^{-8}$). Since Theorem 2 can be seen as the combination of Proposition 2 and the omniscient trajectory (see Corollary B.1 and

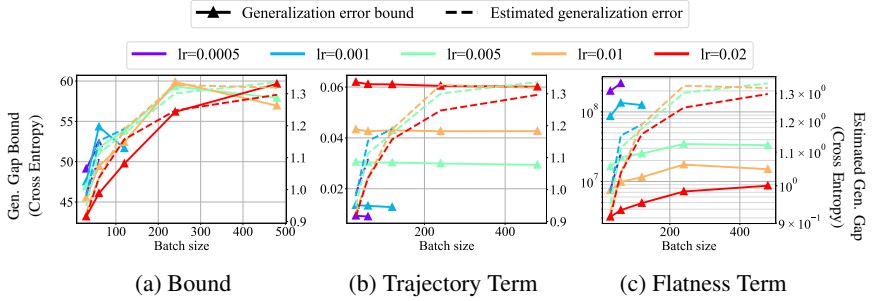

Figure C.1: the isotropic version of Proposition 2 for ResNet-18 on CIFAR-10 under varied flatness.

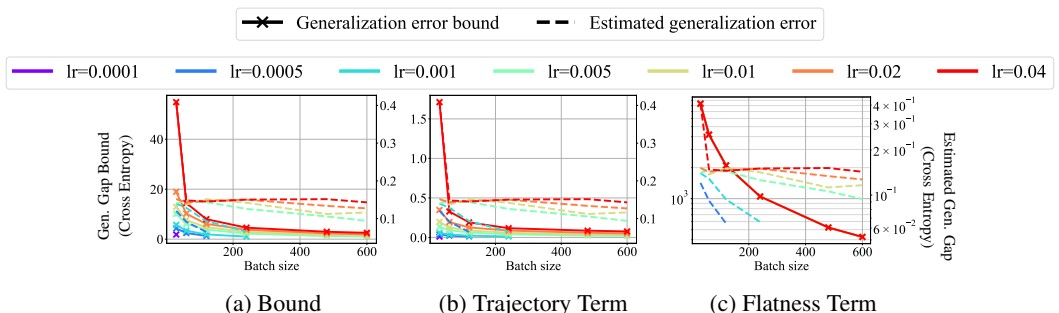

Figure C.2: Wang & Mao (2022)'s bound for 2-layer MLP on MNIST under varied flatness.

how it connects to Proposition 2 by setting $\Delta G \equiv 0$), the improved tightness of our bound compared to Proposition 1 is two-folded, done by both Proposition 2 and the omniscient trajectory.

### C.4.2 RESULTS FOR 2-LAYER MLP ON MNIST

The numerical results on MNIST (LeCun et al., 1998) with the 2-layer MLP are displayed in Figures C.2 to C.5.

### C.5 EXPERIMENTS OF EXISTING BOUNDS WITH POPULATION HESSIAN

In Figures 1, 4 and C.3, we have displayed results of existing bounds *without* using population Hessians. That is, we are computing the bounds whose (approximated) flatness term is

$$\frac{\sigma^2 T}{2} \mathbb{E}\left[\text{tr}\left(\hat{H}_S(W_T)\right)\right]$$

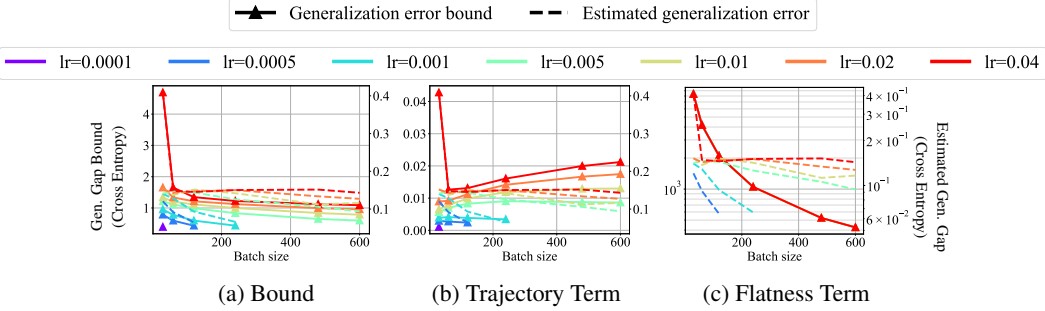

Figure C.3: the isotropic version of Proposition 2 for 2-layer MLP on MNIST under varied flatness.

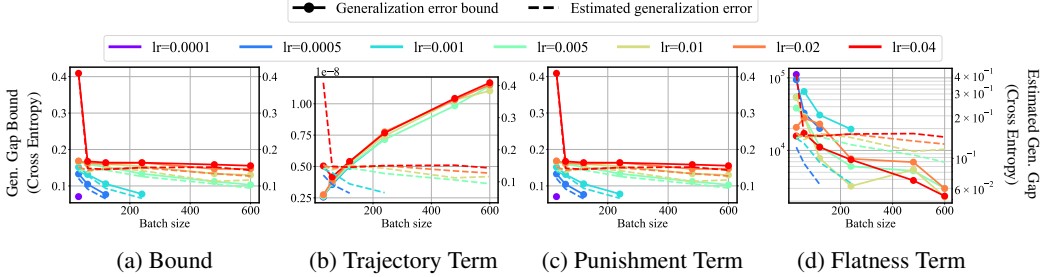

Figure C.4: Numerical results of Theorem 2 with $\lambda = 10^9$ on MNIST and a 2-layer MLP with varied learning rate and batch size.

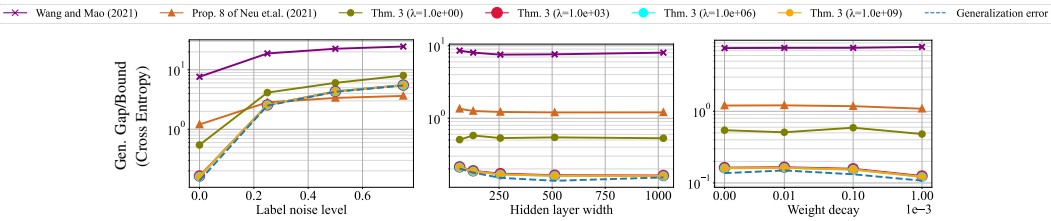

Figure C.5: Numerical results for a 2-layer MLP on MNIST under varied label noise level, width, and weight decay. The isotropic version of Neu et al. (2021)'s Proposition 8 is used.

instead of $\frac{\sigma^2 T}{2} \mathbb{E}\left[\mathrm{tr}\left(\hat{H}_S(W_T)\right) - \mathrm{tr}\left(H_\mu(W_T)\right)\right]$. In contrast, our bound depends heavily on population statistics like the mean gradient and Hessians. One may doubts whether this comparison is unfair due to our extra dependence on population Hessians. This concern is amplified by the fact that the population Hessian potentially improves the numerical tightness of existing bounds: If one assumes as Wang & Mao (2022) that $\mathcal{L}_\mu(W_T) \leq \mathbb{E}_{\xi \sim \mathcal{N}(0,\sigma^2 I)}\left[\mathcal{L}_\mu(W_T + \xi)\right]$, one is essentially assuming $H_\mu(W_T)$ is positive semi-definite, whose trace is non-negative. In Propositions 1 and 2, the non-negative traces are subtracted from the bound, which is tighter than the case where it is not subtracted.

To address this concern, we also allow the existing bounds to depend on the population statistics, especially the population Hessian, and evaluate them in experiments. The results are surprising and confusing: the population Hessians indeed have positive traces but their traces are orders of magnitude larger than the traces of the empirical Hessians. As a result, the flatness term becomes negative. If one optimize the bound parameter $\sigma$, one would arrive at a $-\infty$ bound of the generalization error, which is obviously incorrect. We conjecture this is due to the second order approximation used in the expectation, which becomes inaccurate for population loss changes under the SGLD-like noise of variance $\sigma^2 T I$. This is also a drawback of existing bounds to require the large noises to decrease the impact of the trajectory term in the $\sigma$-optimized bound. In contrast, our bound has an orders-of-magnitude smaller trajectory term and requires orders of magnitude smaller noises. This is supported by empirical results in Table 1, where the value of (average) $\sigma_*^2$ ($\sigma_*$ is the optimal values of $\sigma_t$ in $\sum_t \sigma_t^2$ of Proposition 1, the $\sigma$ in $\Sigma = \sigma^2 T I$ of isotropic Proposition 2, or the $\sigma$ in $\sigma^2 T$ in Corollary B.1 / Theorem 2) in different experiments are presented. Be noted that although $\sigma_*^2$ values themselves seem already small for all experiments, the actual noises in the flatness term has covariance $\sigma_*^2 T I$, where $T$ is the number of training steps ($\sim 10^4$ for CIFAR10) and identity matrix $I$ has a size equal to the number of parameters ($\sim 10^6$ for ResNets). As a result, the magnitude of $\sigma_*^2$ have a significant impact on the second-order approximation in the flatness terms and the second-order approximation is much more accurate for evaluating our bound.

To alleviate this problem, we take absolute values of the flatness terms to make them positive as in the main results of Wang & Mao (2022) and Neu et al. (2021). That is, we now estimate the bound with flatness term replaced by

$$\frac{\sigma^2 T}{2}\left|\mathbb{E}\left[\mathrm{tr}\left(\hat{H}_S(W_T)\right) - \mathrm{tr}\left(H_\mu(W_T)\right)\right]\right|.$$

|  | l.r. and b.s. | label noise | width | weight decay |
|---|---|---|---|---|
| Prop. 1 w/o pop. Hess. | $15381.627 \times 10^{-9}$ | $15372.131 \times 10^{-9}$ | $19904.060 \times 10^{-9}$ | $17674.105 \times 10^{-9}$ |
| Prop. 2 w/o pop. Hess. | $1434.559 \times 10^{-9}$ | $1278.293 \times 10^{-9}$ | $1660.805 \times 10^{-9}$ | $1341.889 \times 10^{-9}$ |
| Prop. 1 w/ pop. Hess. | $6659.479 \times 10^{-9}$ | $6572.998 \times 10^{-9}$ | $9175.748 \times 10^{-9}$ | $9149.648 \times 10^{-9}$ |
| Prop. 2 w/ pop. Hess. | $676.226 \times 10^{-9}$ | $545.839 \times 10^{-9}$ | $767.169 \times 10^{-9}$ | $643.772 \times 10^{-9}$ |
| Thm. 2 ($\lambda = 1$) | $291.674 \times 10^{-9}$ | $595.630 \times 10^{-9}$ | $303.843 \times 10^{-9}$ | $307.434 \times 10^{-9}$ |
| Thm. 2 ($\lambda = 10^3$) | $4.354 \times 10^{-9}$ | $6.269 \times 10^{-9}$ | $3.866 \times 10^{-9}$ | $3.167 \times 10^{-9}$ |
| Thm. 2 ($\lambda = 10^9$) | $\mathbf{0.193} \times 10^{-9}$ | $\mathbf{0.204} \times 10^{-9}$ | $\mathbf{0.244} \times 10^{-9}$ | $\mathbf{0.167} \times 10^{-9}$ |

Table 1: The average value of $\sigma_*^2$ across bounds and CIFAR-10 experiments varying different hyperparameters. It can be seen that our bound requires orders of magnitude smaller noises for the SGLD-like trajectory's Gaussian noises when $\lambda$ is suitably selected. As a result, the distortion due to the second-order approximation has less impact to our bound.

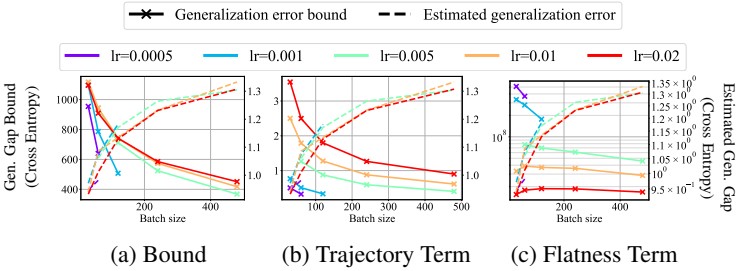

(a) Bound      (b) Trajectory Term      (c) Flatness Term

Figure C.6: Wang & Mao (2022)'s bound with population Hessians for ResNet-18 on CIFAR-10 under varied flatness.

The population Hessian is estimated on a new validation set. After this modification, we display the numerical results for Propositions 1 and 2 in Figures C.6 and C.7.

Since the modification only involves the flatness term, the trajectory term is almost the same as those in Figures 1b and C.1b of the existing bounds without the population Hessian. Therefore, we mainly focus on the flatness term.

Since the subtracted population Hessian traces are orders of magnitude larger than the empirical Hessian, the flatness terms with absolute value are dominated by the population Hessian traces and are orders of magnitude larger than those in Figures 1c and C.1c. As a result, the existing bounds even become looser if fed with validation sets.

When it comes to the tendency, the flatness terms' tendencies *w.r.t.* batch size also become wrong when batch size is large or learning rate is small. As a result, Proposition 2's (Neu et al., 2021) partial improvement of trajectory term *w.r.t.* batch size can not compensate the wrong tendency of the flatness term, and the whole bound scales incorrectly *w.r.t.* batch size.

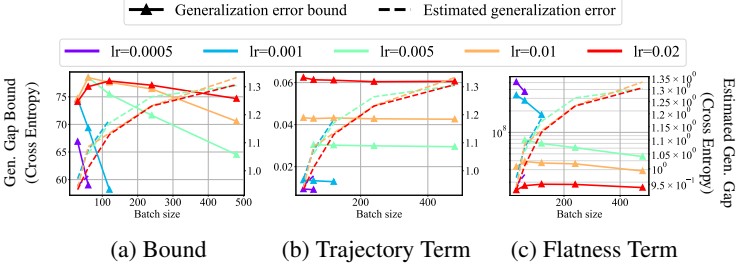

(a) Bound      (b) Trajectory Term      (c) Flatness Term

Figure C.7: The isotropic version of Proposition 2 with population Hessians for ResNet-18 on CIFAR-10 under varied flatness.

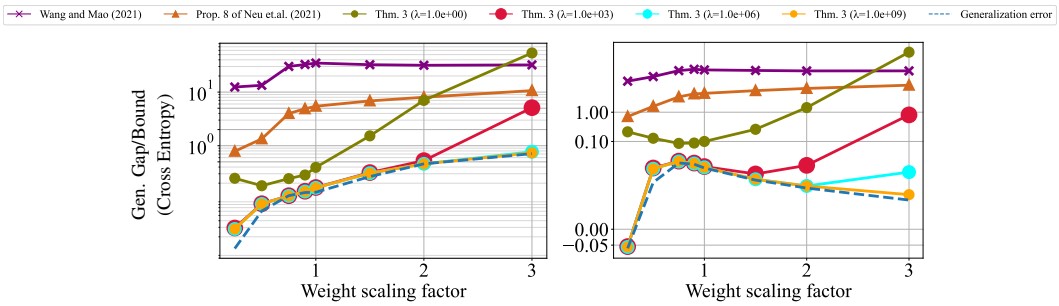

Figure C.8: Numerical results of existing bounds with population Hessians for ResNet-18 on CIFAR-10 under varied training data usage, label noise level, width, depth, and weight decay. The tendency *w.r.t.* weight scaling is also added. However, the results are taken from (unbiased) MLP on MNIST because ResNets are more complicated models and it takes time to develop a weight scaling scheme that does not essentially change the prediction. The isotropic version of Proposition 2 (Neu et al. (2021)'s Prop. 8) is used. If a model has a channel scaling factor of $c$, it means the model has $c$ times the number of channels at each layer compared to a standard ResNet-18.

(a) Generalization gap on all samples.  (b) Gen. gap on correct classification.

Figure C.9: Cross Entropy generalization gap and bounds under weight scaling of MLPs on MNIST. Figure C.9a displays the generalization gap on all testing samples. Figure C.9b displays the generalization gap on only the correctly classified testing samples. Since Figure C.9b involves negative values, we use log-scale above $10^{-4}$ and linear scale below $10^{-4}$.

The correct tendency of the flatness term without population Hessians and the wrong tendency of that with population Hessian indicate population Hessians have wrong tendencies when batch size is large or the learning rate is small. This somehow points to the notion the generalization and overfitting of higher order statistics, as the empirical Hessian has the correct tendency as shown in Figures 1c and C.1c yet the population Hessian has the wrong tendency. Small batch size and large learning rate seem also helpful for this higher-order generalization.

As shown in Figure C.8, when weight decay is used, the existing bounds with population Hessian can better capture the tendency of generalization under regularization. However, when the hyperparameters in Figure C.8 are varied, the existing bounds with population Hessians are times looser than those without the population Hessians.

To sum up, the existing bounds cannot fully exploit the population statistics even if we allow them to access the validation set. Therefore, the comparison between our bound and the existing bounds is still fair, at least for existing bounds in its current form estimated after the conventional (Wang & Mao, 2022) second-order approximation.

## C.6 EXPERIMENTS ON WEIGHT SCALING

To test whether our flatness-related generalization bound captures the generalization under weight scaling, we train homogenous two-layer MLPs, whose output layer has no bias, on MNIST and scale the weight at the end as the last step of the training. The results are displayed in Figure C.9a, where generalization gap is computed for clipped Cross Entropy losses. In this case, our bounds with large $\lambda$ can well capture the generalization.

However, the results in Figure C.9a out of expectation. One would expect the generalization improves as the weight is scaled up because most MNIST samples would be correctly classified and the

generalization risk measured by Cross Entropy would be decreased by increasing the "confidence". However, in Figure C.9a, Cross Entropy increases when scaling up the weights. We conjecture it is because we split the training set into $k = 6$ subsets, the generalization of these MLPs are in fact harmed and about $4\%$ of testing samples are misclassified. As a result, scaling the weight also increases the generalization risk on the misclassified testing samples. The increase on misclassified testing samples are arguably much faster than those on correctly classified samples, because weight scaling on misclassification pushes the loss to infinity while weight scaling on correct classification pushes the loss to zero. As a result, the generalization gap increases in Figure C.9a.

To see how our bound captures the improved generalization on fully correctly classified scenarios, we filter the testing sets and only keep the correctly classified samples, *simulating* the cases where all samples are correctly classified. We implement this simulation using the following loss:

$$\underbrace{\mathbb{I}[\arg\max_i \text{logits}_i = y]}_{\text{Filtering the correct samples}} \cdot \underbrace{\min(\text{CrossEntropy}(\text{logits}, y), 12 \log c)}_{\text{Clipped Cross Entropy}},$$

where $c$ is the number of classes. It is bounded, thus sub-Gaussian and can be rigorously used in the existing and our bounds, whose results can be found in Figure C.9b. In Figure C.9b, when weights are scaled down, the training loss increases and increases faster (maybe the faster increase is related to overfitting), decreasing the gap. However, when weights are sufficiently scaled up, the generalization gap indeed decreases with the decreases of scaling (maybe because now the training loss is almost zero and its decrease slows down). In the latter phase, the bound from Wang & Mao (2022) and Neu et al. (2021) all have weak or wrong dependency *w.r.t.* weight scaling. In contrast, our bound still well captures the improved generalization in both cases, if $\lambda$ is suitably selected. Our bounds even captures the strange negative generalization error. By inspecting the raw data, we find the negative value mainly comes from the empirical part of the penalty term, indicating that the negativeness of the bound value is not mainly due to the dependence on the validation set.

To sum up, our bounds can well capture the generalization under weight scaling in our experiments.

# D  PROOFS AND DETAILS FOR SEC. 5 EXTENSIONS

In this section, we present and prove the combined results of the omniscient trajectory wth existing results to demonstrate the flexibility of the technique. The proofs generally follow those of the existing results. But note that the omniscient and the SGLD-like trajectories may depend on random variables more than the training set $S$. Since the existing results often assume the trajectory solely depends on $S$, we must re-prove some of them instead of directly applying them to handle the extra dependence. Sometimes we must rearrange the order of the steps of the original proofs to insert the omniscient trajectory. Setting the omniscient perturbation to zero will recover the existing results, which we will not bother to restate.

## D.1  COMBINATION WITH THE INDIVIDUAL-SAMPLE TECHNIQUE

**Corollary D.1** *Assume $\ell(w, \cdot)$ is $R$-sub-Gaussian on $\mu$ for any $w \in \mathcal{W}$. Recall $n$ is the number of training samples. Then for any family $\left\{ \Delta g^i_{1:T} \right\}_{i=1}^n$ of omniscient perturbations and any $\sigma_{1:T} \in \left( \mathbb{R}^{>0} \right)^T$, we have*

$$
\begin{aligned}
\text{gen}(\mu^n, P_{W|S}) \leq & \frac{R}{n} \sum_{i=1}^n \sqrt{2I(\tilde{W}^i_T; Z_i)} + \frac{1}{n} \sum_{i=1}^n \mathbb{E}\left[ \Delta^{\sum_t \sigma_t^2}_{\Gamma^i_T}(W_T, Z_i) - \Delta^{\sum_t \sigma_t^2}_{\Gamma^i_T}(W_T, S') \right] \\
\leq & \frac{R}{n} \sum_{i=1}^n \sqrt{2I(\tilde{W}^i_T; Z_i \mid Z_{-i})} + \frac{1}{n} \sum_{i=1}^n \mathbb{E}\left[ \Delta^{\sum_t \sigma_t^2}_{\Gamma^i_T}(W_T, Z_i) - \Delta^{\sum_t \sigma_t^2}_{\Gamma^i_T}(W_T, S') \right] \\
\leq & \frac{R}{n} \sum_{i=1}^n \sqrt{\sum_{t=1}^T \frac{1}{\sigma_t^2} \mathbb{E}\left[ \left\| g_t - \mathbb{E}\left[ g_t \mid Z_{-i} \right] - \Delta g^i_t \right\|^2 \right]} \\
& + \frac{1}{n} \sum_{i=1}^n \mathbb{E}\left[ \Delta^{\sum_t \sigma_t^2}_{\Gamma^i_T}(W_T, Z_i) - \Delta^{\sum_t \sigma_t^2}_{\Gamma^i_T}(W_T, S') \right], \quad \text{(D.1)}
\end{aligned}
$$

*where $\Delta g_t^i$ is a deterministic function of $(S, V, g_{1:T}, W_{0:T}, Z_{-i})$, and $\Gamma_t^i := \sum_{\tau=1}^{t} \Delta g_\tau^i$.*

**Remark D.1** *The omniscient trajectory additionally depends on $i$ and $Z_{-i}$, the two featuring random variables of the individual-sample bounds.*

**Proof** The crux of the individual-sample technique is to first extract the summation in the empirical risk and change its order with the expectation, before other things like using the MI bound or building the auxiliary trajectories:

$$
\begin{aligned}
&\text{gen}(\mu^n, P_{W|S}) \\
&:= \mathbb{E}\left[\mathcal{L}_\mu(W_T) - \frac{1}{n}\sum_{i=1}^{n} \ell(W_T, Z_i)\right] = \frac{1}{n}\sum_{i=1}^{n} \mathbb{E}\left[\mathcal{L}_\mu(W_T) - \ell(W_T, Z_i)\right] \\
&= \frac{1}{n}\sum_{i=1}^{n} \mathbb{E}\left[\mathcal{L}_\mu(\breve{W}_T^i) - \ell(\breve{W}_T^i, Z_i)\right] + \frac{1}{n}\sum_{i=1}^{n} \mathbb{E}\left[\Delta_{\Gamma_T^i}(W_T, Z_i) - \Delta_{\Gamma_T^i}(W_T, S')\right],
\end{aligned}
$$

where $\breve{W}_t^i := W_t + \Gamma_T^i$ is the $i$-th omniscient trajectory. It can be seen that the omniscient penalty term has matched that in Eq. (D.1). Thus, we will then bound the individual-sample generalization error of the omniscient trajectories. The process is similar to the proof of Theorem 1, *i.e.,* adding Gaussian noises to obtain the SGLD trajectories:

$$
\begin{aligned}
\frac{1}{n}\sum_{i=1}^{n} \mathbb{E}\left[\mathcal{L}_\mu(\breve{W}_T^i) - \ell(\breve{W}_T^i, Z_i)\right] &= \frac{1}{n}\sum_{i=1}^{n} \mathbb{E}\left[\mathcal{L}_\mu(\tilde{W}_T^i) - \ell(\tilde{W}_T^i, Z_i)\right] \\
&+ \frac{1}{n}\sum_{i=1}^{n} \mathbb{E}\left[\Delta^{\sum_t \sigma_t^2}(W_T + \Gamma_T^i, Z_i) - \Delta^{\sum_t \sigma_t^2}(W_T + \Gamma_T^i, S')\right],
\end{aligned}
$$

where $\tilde{W}_t^i := \breve{W}_t^i + \sum_{\tau=1}^{t} N_t^i$, and $N_t^i \sim \mathcal{N}(0, \sigma_t^2 I)$ is an independent Gaussian noise. Again we throw away the already matched penalty terms and focus on bounding the individual-sample generalization error of the SGLD-like trajectory. To this end, we see each term in the error as the generalization error of an algorithm that takes one sample (but stealthily samples $n-1$ samples as algorithm's internal randomness) and apply Lemma 1:

$$
\begin{aligned}
\frac{1}{n}\sum_{i=1}^{n} \mathbb{E}\left[\mathcal{L}_\mu(\tilde{W}_T^i) - \ell(\tilde{W}_T^i, Z_i)\right] = \frac{1}{n}\sum_{i=1}^{n} \text{gen}(\mu^n, P_{\tilde{W}_T^i|Z_i}, \ell) &\leq \frac{1}{n}\sum_{i=1}^{n}\sqrt{\frac{2R^2}{1} I(\tilde{W}_T^i; Z_i)} \\
&\leq \frac{R}{n}\sum_{i=1}^{n}\sqrt{2I(\tilde{W}_T^i; Z_i \mid Z_{-i})}.
\end{aligned}
$$

We proceed by a similar arguments as in Theorem 1 to bound the mutual information: for any $i \in [n]$, we have

$$
\begin{aligned}
I(\tilde{W}; Z_i \mid Z_{-i}) &\leq I(\tilde{W}_0^i; Z_i \mid Z_{-i}) + \sum_{t=1}^{T} I(\tilde{W}_t^i; Z_i \mid \tilde{W}_{0:t-1}^i, Z_{-i}) \\
&= \sum_{t=1}^{T} I(\tilde{W}_{t-1}^i - (g_t - \Delta g_t^i) + N_t^i; Z_i \mid \tilde{W}_{0:t-1}^i, Z_{-i}).
\end{aligned}
$$

By instantiating $X^i = Z_i, Y^i = \tilde{W}_{0:t-1}^i, \Delta^i = (W_{0:T}, V, Z_{-i}), O^i = Z_{-i}$ and

$$
\begin{aligned}
f^i(X^i, Y^i, \Delta^i) = \tilde{W}_{t-1}^i - \Big( & g_t(W_{t-1}, (Z_1, \ldots, Z_{i-1}, Z_i, Z_{i+1}, \ldots, Z_n), V, W_{0:t-2}) \\
& - \Delta g_t^i(S, V, g_{1:T}, W_{0:T}, Z_{-i})\Big)
\end{aligned}
$$

and applying Eq. (B.1) in Lemma B.1, for any deterministic function $\Omega^i$ of solely $(\tilde{W}_{0:t-1}^i, Z_{-i})$, respectively, we have

$$
I(\tilde{W}_{t-1}^i - (g_t - \Delta g_t^i) + N_t^i; Z_i \mid \tilde{W}_{0:t-1}^i, Z_{-i}) \leq \frac{1}{2\sigma_t^2}\mathbb{E}\left[\left\|\tilde{W}_{t-1}^i - (g_t - \Delta g_t^i) - \Omega^i(\tilde{W}_{0:t-1}^i, Z_{-i})\right\|^2\right].
$$

Setting $\Omega^i(\tilde{w}^i_{0:t-1}, z_{-i}) = \tilde{w}^i_{t-1} - \mathbb{E}\left[g_t \mid Z_{-i} = z_{-i}\right]$ leads to

$$I(\tilde{W}^i_{t-1} - (g_t - \Delta g^i_t) + N^i_t; Z_i \mid \tilde{W}^i_{0:t-1}, Z_{-i}) \leq \frac{1}{2\sigma^2_t} \mathbb{E}\left[\left\|g_t - \mathbb{E}\left[g_t \mid Z_{-i}\right] - \Delta g^i_t\right\|^2\right].$$

After putting everything together, the corollary is proved. ∎

## D.2 COMBINATION WITH THE CMI FRAMEWORK

Conditional MI (CMI) framework (Steinke & Zakynthinou, 2020) is developed to fundamentally solve the potential unboundedness of MI on some algorithms. CMI framework considers the following random process: Firstly, a supersample $\tilde{S} = \tilde{Z}_{1:n,0:1} \in \mathcal{Z}^{n \times 2}$ of $2n$ samples are sampled as an effective discrete "sample space". Then, $n$ independent indices $U_{1:n} \in \{0,1\}^n$ are sampled as the discrete "sample". The training set is constructed by $S := (\tilde{Z}_{i,U_i})^n_{i=1}$ using the supersample and indices. CMI then bounds the generalization by how much the output reveals the indices given the supersample. We can combine the omniscient trajectory with CMI framework and give the following result:

**Corollary D.2** *Assume $\ell(w, \cdot)$ is $R$-sub-Gaussian on $\mu$ for any $w \in \mathcal{W}$. Then for any omniscient trajectory $\Delta g_{1:T}$ that additionally depends on $(\tilde{S}, U)$, and any $\sigma_{1:T} \in \left(\mathbb{R}^{>0}\right)^T$, we have*

$$\mathrm{gen}(\mu^n, P_{W|S}) \leq 2\sqrt{\frac{2R^2}{n} I(\tilde{W}_T; U \mid \tilde{S})} + \mathbb{E}\left[\Delta^{\sum_t \sigma^2_t}_{\Gamma_T}(W_T, Z_i) - \Delta^{\sum_t \sigma^2_t}_{\Gamma_T}(W_T, S')\right]$$

$$\leq 2\sqrt{\frac{R^2}{n} \sum^T_{t=1} \frac{1}{\sigma^2_t} \mathbb{E}\left[\left\|g_t - \mathbb{E}\left[g_t \mid \tilde{S}\right] - \Delta g_t\right\|^2\right]}$$

$$+ \mathbb{E}\left[\Delta^{\sum_t \sigma^2_t}_{\Gamma_T}(W_T, Z_i) - \Delta^{\sum_t \sigma^2_t}_{\Gamma_T}(W_T, S')\right],$$

*where $\Delta g_t$ is a deterministic function of $(S, V, g_{1:T}, W_{0:T}, \tilde{S}, U)$.*

**Remark D.2** *The omniscient perturbation additionally depends on $(\tilde{S}, U)$, which are the featuring random variables of CMI bounds.*

**Proof** After paying the penalties for changes of trajectories, we can focus on bounding the generalization error on the SGLD-like trajectory. With the omniscient trajectory depending on $(\tilde{S}, U)$, the distribution of the SGLD-like trajectory is determined by $P_{\tilde{W}_{0:T}|\tilde{S},U}$. We will apply the following re-proved CMI bound that handles the extra dependence on $(\tilde{S}, U)$ of $\tilde{W}_T$:

$$\mathbb{E}\left[\mathcal{L}_\mu(\tilde{W}_T) - \hat{\mathcal{L}}_S(\tilde{W}_T)\right] = 2\mathbb{E}\left[\hat{\mathcal{L}}_{\tilde{S}}(\tilde{W}_T) - \hat{\mathcal{L}}_S(\tilde{W}_T)\right]$$

$$= 2\mathbb{E}_{\tilde{S}}\left[\mathbb{E}\left[\hat{\mathcal{L}}_{\tilde{S}}(\tilde{W}_T) - \hat{\mathcal{L}}_S(\tilde{W}_T) \mid \tilde{S}\right]\right] = 2\mathbb{E}_{\tilde{S}}\left[\mathrm{gen}(P_U, P_{\tilde{W}_T|U,\tilde{S}}) \mid \tilde{S}\right]$$

$$\leq 2\mathbb{E}_{\tilde{S}}\left[\sqrt{\frac{2R^2}{n} I(\tilde{W}_T; U \mid \tilde{S} = \tilde{S})} \mid \tilde{S}\right] \qquad \text{(Lemma 1)}$$

$$\leq 2\sqrt{\frac{2R^2}{n} \mathbb{E}_{\tilde{S}}\left[I(\tilde{W}_T; U \mid \tilde{S} = \tilde{S})\right]} = 2\sqrt{\frac{2R^2}{n} I(\tilde{W}_T; U \mid \tilde{S})},$$

where the first step is because

$$\mathbb{E}\left[\hat{\mathcal{L}}_{\tilde{S}}(\tilde{W}_T) - \hat{\mathcal{L}}_S(\tilde{W}_T)\right]$$

$$= \mathbb{E}\left[\frac{1}{n}\sum^n_{i=1} \frac{1}{2}\left(\hat{\mathcal{L}}_{Z_{2(i-1)+U_i}}(\tilde{W}_T) + \hat{\mathcal{L}}_{Z_{2(i-1)+1-U_i}}(\tilde{W}_T)\right) - \frac{1}{n}\sum^n_{i=1} \hat{\mathcal{L}}_{Z_{2(i-1)+U_i}}(\tilde{W}_T)\right]$$

$$= \frac{1}{2}\mathbb{E}\left[\frac{1}{n}\sum^n_{i=1} \hat{\mathcal{L}}_{Z_{2(i-1)+1-U_i}}(\tilde{W}_T) - \hat{\mathcal{L}}_S(\tilde{W}_T)\right] = \frac{1}{2}\mathbb{E}\left[\mathcal{L}_\mu(\tilde{W}_T) - \hat{\mathcal{L}}_S(\tilde{W}_T)\right].$$

The following steps are similar to the proof of Theorem 1 and Corollary D.1, including an application of the chain rule to the conditional MI and an application of Lemma B.1, where $X = S, Y = \tilde{W}_{0:t-1}\ O = (W_{0:T}, V, \tilde{S}, U)$ and $\Omega = \tilde{W}_t - \mathbb{E}\left[g_t \mid \tilde{S}\right]$. ∎

### D.3 COMBINATION WITH NEGREA ET AL.(2019)'S DATA-DEPENDENT PRIOR

To state Corollary D.3, we need extra notations. For an array $a_{1:l}$ of length $l$ and a list $U \in [l]^k$ of indices, define $a_U := (a_{U_i})_{i=1}^k$. For two lists of non-repeated indices $U^1 \in \mathbb{N}^{k_1}$ and $U^2 \in \mathbb{N}^{k_2}$, let $U^1 \setminus U^2$ be the list of indices that can be found in $U^1$ but not in $U^2$, ordered as in $U^1$. Let $U^1 \cap U^2$ be their intersection ordered as in $U^1$.

**Corollary D.3** *Assume $\ell(w, \cdot)$ is R-sub-Gaussian on $\mu$ for any $w \in \mathcal{W}$. Assume the following specific form for the algorithm:*

- *The randomness $V = B_{1:T}$, where $B_t \in [n]^b$ specifies the samples in the b-sized batch at step t with* no repeated *entries;*

- *The update function can be written as*
$$g_t(W_{t-1}, S, V, W_{0:t-2}) = \bar{\nabla}\hat{\mathcal{L}}_{S_{B_t}}(W_{t-1}),$$
  *where $\bar{\nabla}$ stands for* samplewisely *clipped gradient operator (i.e., compute gradients for each sample, clip and then average).*

*Let $U \in [n]^m$ be the result of m uniform (over the remaining indices) sample indices* without *replacement in $[n]$, independent of previous random variables. Define*
$$\xi_t := \frac{b - |U \cap B_t|}{b}\left(\bar{\nabla}\hat{\mathcal{L}}_{S_{B_t \setminus U}}(W_{t-1}) - \bar{\nabla}\hat{\mathcal{L}}_{S_U}(W_{t-1})\right)$$
*to be the gradient* incoherence *(see it as the difference between the true update at step t and a prediction of the update using only samples in U). For omniscient trajectories defined by* bounded *$\Delta g_{1:T}$ that additionally* depends on U, *then we have*

$$\text{gen}(\mu^n, P_{W|S}) \leq \sqrt{\frac{R^2}{n-m}\sum_{t=1}^T \frac{1}{\sigma_t^2}\mathbb{E}\left[\|\xi_t - \Delta g_t\|^2\right] + \mathbb{E}\left[\Delta_{\Gamma_T}^{\sum_t \sigma_t^2}(W_T, S_U^c) - \Delta_{\Gamma_T}^{\sum_t \sigma_t^2}(W_T, S')\right]}.$$

Corollary D.3 is an application of the following more general result:

**Corollary D.4** *Assume $\ell(w, \cdot)$ is R-sub-Gaussian on $\mu$ for any $w \in \mathcal{W}$. Assume samples have no interactions in updates in the following sense:*

- *The randomness $V$ in the update is $V = B_{1:T}$, where $B_t \in [n]^b$ specifies the samples in the b-sized batch at step t with* no repeated *entries;*

- *The update function can be written as*
$$g_t(W_{t-1}, S, V, W_{0:t-2}) = p_t(W_{t-1}, S_{B_t}, W_{0:t-2}),$$
  *where $p_t : \mathcal{W} \times \bigcup_{i=0}^\infty \mathcal{Z}^i \times \mathcal{W}^{t-1} \to \mathcal{W}$ is defined by*
$$p_t(w_{t-1}, s, w_{0:t-2}) := \begin{cases} 0 & |s| = 0, \\ \frac{1}{|s|}\sum_{i=1}^{|s|} f_t(w_{t-1}, s_i, w_{0:t-2}) & |s| > 0, \end{cases}$$
  *and $f_t : \mathcal{W} \times \mathcal{Z} \times \mathcal{W}^{t-1} \to \mathcal{W}$ is a deterministic function.*

*Also, let $U \in [n]^m$ be the result of m uniform (over the remaining indices) samples* without *replacement in $[n]$, independent of previous random variables. Lastly, assume that given $(U, S_U, S_U^c, V, \tilde{W}_{0:t-1})$, then $p_t(W_{t-1}, S_{B_t}, W_{0:t-2}), p_t(W_{t-1}, S_U, W_{0:t-2})$ and*

$p_t(W_{t-1}, S_{B_t \cap U}, W_{0:t-2})$ *all have bounded second moments (e.g., by samplewise gradient clipping).*

*Define*

$$\xi_t := \frac{b - |U \cap B_t|}{b} \left( p_t(W_{t-1}, S_{B_t \setminus U}, W_{0:t-2}) - p_t(W_{t-1}, S_U, W_{0:t-2}) \right)$$

*to be the gradient* incoherence*. For omniscient trajectories defined by $\Delta g_{1:T}$ that additionally depend on $U$, if $\Delta g_t$ also satisfies the above conditional bounded second moments (e.g., bounded in value), then we have*

$$\text{gen}(\mu^n, P_{W|S}) \le \sqrt{\frac{R^2}{n-m} \sum_{t=1}^{T} \frac{1}{\sigma_t^2} \mathbb{E}\left[ \|\xi_t - \Delta g_t\|^2 \right] + \mathbb{E}\left[ \Delta_{\Gamma_T}^{\sum_t \sigma_t^2}(W_T, S_U^c) - \Delta_{\Gamma_T}^{\sum_t \sigma_t^2}(W_T, S') \right]}.$$

**Remark D.3** *The omniscient trajectory additionally depends on $U$ and then $\xi_t$, the two featuring random variables of Negrea et al. (2019)'s SGLD bound.*

**Proof** In this proof, we combine Theorem 2.4 and Theorem 3.1 of Negrea et al. (2019) with our technique. As Negrea et al. (2019), let $S_U^c := S_{(1,2,\ldots,n) \setminus U}$. Due to extra dependence on $(U, V)$ of the omniscient trajectory, the SGLD-like trajectory is determined by $P_{\tilde{W}_{0:T}|(U,S,V)}$, violating the assumptions of Negrea et al. (2019)'s Theorem 2.4. Therefore we must re-prove this data-dependent prior bound, during which we insert the auxiliary trajectories:

$$\begin{aligned} \text{gen}(\mu^n, P_{W|S}) &= \mathbb{E}\left[ \mathcal{L}_\mu(W_T) - \hat{\mathcal{L}}_{S_U^c}(W_T) \right] \\ &= \mathbb{E}\left[ \mathcal{L}_\mu(\tilde{W}_T) - \hat{\mathcal{L}}_{S_U^c}(\tilde{W}_T) \right] + \mathbb{E}\left[ \Delta_{\Gamma_T}^{\sum_t \sigma_t^2}(W_T, S_U^c) - \Delta_{\Gamma_T}^{\sum_t \sigma_t^2}(W_T, S') \right], \end{aligned}$$

where the first equality is because given $(S, W_T)$, $\hat{\mathcal{L}}_{S_U^c}(W_T)$ is an unbiased estimator on the empirical loss. With the penalty term matching with the statement of the corollary, we focus on the generalization error of the SGLD-like trajectory, which can be bounded by

$$\begin{aligned} \mathbb{E}\left[ \mathcal{L}_\mu(\tilde{W}_T) - \hat{\mathcal{L}}_{S_U^c}(\tilde{W}_T) \right] &= \mathbb{E}_{U,S_U,V}\left[ \mathbb{E}\left[ \mathcal{L}_\mu(\tilde{W}_T) - \hat{\mathcal{L}}_{S_U^c}(\tilde{W}_T) \mid U, S_U, V \right] \right] \\ &= \mathbb{E}_{U,S_U,V}\left[ \text{gen}(\mu^{n-m}, P_{\tilde{W}_T|U,S_U^c,S_U,V}) \right] \\ &\le \mathbb{E}_{U,S_U,V}\left[ \sqrt{\frac{2R^2}{n-m} I(\tilde{W}_T; S_U^c \mid U = U, S_U = S_U, V = V)} \right]. \quad \text{(Lemma 1)} \end{aligned}$$

By the "golden formula" of MI (Eq.(8.7) of Csiszár & Körner (2011)) stating $I(X; Y) = \mathbb{E}_Y\left[ \mathcal{D}_{\text{KL}}\left( P_{X|Y} \| P_X \right) \right] \le \mathbb{E}_Y\left[ \mathcal{D}_{\text{KL}}\left( P_{X|Y} \| Q_X \right) \right]$, for any data-dependent prior $Q_{\tilde{W}'_{0:T}}(u, s_u, v)$ over $\mathcal{W}^{T+1}$ under the Definition 2.1 of Negrea et al. (2019), we have

$$\begin{aligned} &\mathbb{E}\left[ \mathcal{L}_\mu(\tilde{W}_T) - \hat{\mathcal{L}}_{S_U^c}(\tilde{W}_T) \right] \\ &\le \mathbb{E}_{U,S_U,V}\left[ \sqrt{\frac{2R^2}{n-m} \mathbb{E}_{S_U^c|U,S_U,V}\left[ \mathcal{D}_{\text{KL}}\left( P_{\tilde{W}_{0:T}|U,S_U,S_U^c,V} \| Q_{\tilde{W}'_{0:T}}(U, S_U, V) \right) \right]} \right] \end{aligned}$$

Now we turn to bound the KL divergence given any realization $(u, s_u, s_u^c, v)$ of the conditioned variables. We restrict the data-dependent prior to have the same distribution as the real distribution for $\tilde{W}_0$, *i.e.*, $P_{\tilde{W}_0|u,s_u,s_u^c,v} = P_{\tilde{W}_0} = Q_{\tilde{W}'_0}(u, s_u, v)$. By Proposition 2.6 of Negrea et al. (2019), we have

$$\begin{aligned} &\mathcal{D}_{\text{KL}}\left( P_{\tilde{W}_{0:T}|u,s_u,s_u^c,v} \| Q_{\tilde{W}_T}(u, s_u, v) \right) \le \mathcal{D}_{\text{KL}}\left( P_{\tilde{W}_{0:T}|u,s_u,s_u^c,v} \| Q_{\tilde{W}'_{0:T}}(u, s_u, v) \right) \\ &\le \sum_{t=1}^{T} \mathbb{E}_{\tilde{W}_{0:t-1} \sim P_{\tilde{W}_{0:t-1}|u,s_u,s_u^c,v}} \left[ \mathcal{D}_{\text{KL}}\left( P_{\tilde{W}_t|\tilde{W}_{0:t-1},u,s_u,s_u^c,v} \| Q_{\tilde{W}'_t|\tilde{W}_{0:t-1}}(u, s_u, v) \right) \right] \quad \text{(D.2)} \end{aligned}$$

We fix a condition $\boldsymbol{c} = (u, s_u, s_u^c, v, \tilde{w}_{0:t-1})$ and turn to bound the KL divergences in the above expectation. We also instantiate the data-dependent prior to the prior designed by Negrea et al. (2019) for SGLD, which uses samples in $S_U$ to predict the update using $B_t$ so that the error between the predicted and the true update eventually becomes the trajectory term. The gradients and updates depend on the model weight and so do the predictions. Therefore, the data-dependent prior must somehow access the $(t-1)$-th weight of the original trajectory to make predictions. In Negrea et al. (2019)'s proof for SGLD, the data-dependent prior technique is directly applied to the original trajectory and thus the $(t-1)$-th original weight can be directly found in the condition $\boldsymbol{c}$. However, in our case where the technique is applied to the auxiliary SGLD-like trajectory, we only have the latest step of the SGLD-like trajectory in the condition, which is not the original SGD weight but corresponds to many $W_{t-1}$ or $\breve{W}_{t-1}$ on the SGD or the omniscient trajectory, given that the SGLD-like trajectory is constructed by adding Gaussian noises to the omniscient trajectory. Directly predicting and comparing gradients at the SGLD-like trajectory will make the final result related to the SGLD-like trajectory. Switching back using local smoothness along the trajectory will result in a "local gradient sensitivity" term as Neu et al. (2021) that accumulates very fast along the training process (Wang & Mao, 2022). The ideal adapted proof would be somehow "tracing back" to the SGD weight $W_{t-1}$ that leads to the SGLD weight in the condition, making predictions based on the SGD weight $W_{t-1}$, and comparing it with the update $g_t(W_{t-1})$ on the SGD trajectory, followed by averaging over the distribution of $W_{t-1}$ given the SGLD-like trajectory weight. This intuition suggests a coupling between the true and the predicted update through $W_{t-1}$, which reminds us of a lemma by Neu et al. (2021):

**Lemma D.1 (Lemma 4 of Neu et al. (2021))** *Let $X$ and $Y$ be random variables taking values in $\mathbb{R}^d$ with bounded second moments and let $\sigma > 0$. Letting $\epsilon \sim \mathcal{N}(0, \sigma^2 I)$ be independent of $(X, Y)$, the KL divergence between the distributions of $X + \epsilon$ and $Y + \epsilon$ is bounded as*

$$\mathcal{D}_{\mathrm{KL}} \left( P_{X+\epsilon} \parallel P_{Y+\epsilon} \right) \leq \frac{1}{2\sigma^2} \mathbb{E} \left[ \|X - Y\|^2 \right],$$

*where the expectation is taken over any joint distribution with* any coupling *between $X$ and $Y$ as long as the marginals are still $P_X, P_Y$, respectively.*

The arbitrariness of the coupling allows a specific one through $W_{t-1}$. Therefore, we turn to construct the coupling of the data-dependent prior.

Since the coupling involves multiple random variables, it must start from a joint distribution. For any condition $(u, s_u, s_u^c, v, \tilde{w}_{0:t-1})$, let $Q_{M_{0:T}, \breve{M}_t, \breve{M}_t', \tilde{M}_t, \tilde{M}_t' | u, s_u, s_u^c, v, \tilde{w}_{0:t-1}}$ be the joint distribution over $\mathcal{W}^5$ given by the Markov chain $M_{0:T} \to (\breve{M}_t, \breve{M}_t') \to (\tilde{M}_t, \tilde{M}_t')$, where $M_{0:T}$ is sampled from $P_{W_{0:T} | u, s_u, s_u^c, v, \tilde{w}_{0:t-1}}$ and

$$\begin{aligned}
\breve{M}_t :=& \tilde{w}_{t-1} - (g_t(M_{t-1}, s, v, M_{0:t-2}) - \Delta g_t(s, v, g_{1:T}, M_{0:T}, u)), \\
\breve{M}_t' :=& \tilde{w}_{t-1} - \left( \frac{|u_t|}{b} p_t(M_{t-1}, s_{u_t}, M_{0:t-2}) + \frac{b - |u_t|}{b} p_t(M_{t-1}, s_u, M_{0:t-2}) \right), \quad \text{(D.3)} \\
\tilde{M}_t =& \breve{M}_t + N_t', \tilde{M}_t' = \breve{M}_t' + N_t',
\end{aligned}$$

where $j_t := j \cap b_t$ is the samples in $j$ contained in the current batch, $N_t' \sim \mathcal{N}(0, \sigma_t^2 I)$ is independent of other random variables in the random process defining $Q$. By assumption, terms defining $\breve{M}_t$ and $\breve{M}_t'$ have bounded second moments, satisfying the assumption of Lemma D.1. It can be easily verified that the marginal distribution $Q_{\tilde{M}_t | \boldsymbol{c}} = P_{\tilde{W}_t | \boldsymbol{c}}$. Moreover, all used samples in Eq. (D.3) are contained in $S_U$. As a result, the marginal distribution of $M_t'$ is independent of $S_U^c$. Therefore, by setting the data-dependent prior $Q_{\tilde{W}_t' | \tilde{W}_{0:t-1}}(u, s_u, v) = Q_{\tilde{M}_t' | u, s_u, (s_u^c)'(u, \tilde{w}_{0:t-1}, s_u, v), v, \tilde{w}_{0:t-1}}$, where $(s_u^c)'(\cdot)$ is a function that outputs some (conditionally) supported realization of $S_U^c$, we have $Q_{\tilde{W}_t' | \tilde{W}_{0:t-1}}(u, s_u, v) = Q_{\tilde{M}_t' | \boldsymbol{c}}$. With the help of this coupling and Lemma D.1, we can bound the

KL divergence in Eq. (D.2) by

$$
\begin{aligned}
&\mathcal{D}_{\mathrm{KL}}\left(P_{\tilde{W}_t|\boldsymbol{c}} \parallel Q_{\tilde{W}_t'|\tilde{W}_{0:t-1}}(u, s_u, v)\right) = \mathcal{D}_{\mathrm{KL}}\left(Q_{\check{M}_t|\boldsymbol{c}} \parallel Q_{\check{M}_t'|\boldsymbol{c}}\right) \\
&= \mathcal{D}_{\mathrm{KL}}\left(Q_{\check{M}_t+N_t'|\boldsymbol{c}} \parallel Q_{\check{M}_t'+N_t'|\boldsymbol{c}}\right) \leq \frac{1}{2\sigma_t^2}\mathbb{E}_{(\check{M}_t, \check{M}_t') \sim Q_{\check{M}_t, \check{M}_t'|\boldsymbol{c}}}\left[\left\|\check{M}_t - \check{M}_t'\right\|^2\right] \quad \text{(Lemma D.1)} \\
&= \frac{1}{2\sigma_t^2}\mathbb{E}_Q\left[\left\|g_t(M_{t-1}, s, v, M_{0:t-2}) - \Delta g_t(s, v, g_{1:T}, M_{0:T}, u)\right.\right. \\
&\qquad\qquad\left.\left. - \left(\frac{|u_t|}{b}p_t(M_{t-1}, s_{u_t}, M_{0:t-2}) + \frac{b-|u_t|}{b}p_t(M_{t-1}, s_u, M_{0:t-2})\right)\right\|^2 \,\Big|\, \boldsymbol{c}\right] \quad \text{Eq. (D.3)} \\
&= \frac{1}{2\sigma_t^2}\mathbb{E}_P\left[\left\|g_t(W_{t-1}, s, v, W_{0:t-2}) - \Delta g_t(s, v, g_{1:T}, W_{0:T}, u)\right.\right. \\
&\qquad\qquad\left.\left. - \left(\frac{|u_t|}{b}p_t(W_{t-1}, s_{u_t}, W_{0:t-2}) + \frac{b-|u_t|}{b}p_t(W_{t-1}, s_u, W_{0:t-2})\right)\right\|^2 \,\Big|\, \boldsymbol{c}\right] \\
&= \frac{1}{2\sigma_t^2}\mathbb{E}\left[\left\|\xi_t - \Delta g_t\right\|^2 \,\Big|\, \boldsymbol{c}\right],
\end{aligned}
$$

where the penultimate step is because, by construction, we have $Q_{M_{0:T}|\boldsymbol{c}} = P_{W_{0:T}|\boldsymbol{c}}$.

After plugging everything together, interchanging the order of the expectation and the square root by the latter's concavity, and putting expectations together, the corollary is proved. ∎

## D.4 Generalization of Stable Algorithms and GD on CLB Problems

**Theorem 3** *Assume $n, d \in \mathbb{N}^+$, $L, D > 0$ and $\eta > 0$, $T \in \mathbb{N}^+$. Assume the algorithm starts from a fixed initialization $W_0 := w_0 \in \mathcal{W}$ and the whole trajectory remains within $\mathcal{W}$, i.e., $W_{0:T} \in \mathcal{W}^{T+1}$. For any CLB problem $(\mathcal{W}, \mathcal{Z}, \ell) \in \mathcal{C}_{L,D}$ and any data distribution $\mu \in \mathcal{M}_1(\mathcal{Z})$, we have the following data-dependent and -agnostic bounds that recover stability bounds:*

$$
\begin{aligned}
&\mathrm{gen}(\mu^n, P_{W|S}) \\
&\leq \inf_{\Delta G^{1:n}} \frac{LD}{n}\sum_{i=1}^n \sqrt{2I\left(W_T + \Delta G^i; Z_i \mid Z_{-i}\right)} + \frac{1}{n}\sum_{i=1}^n \mathbb{E}\left[\Delta_{\Delta G^i}(W_T, Z_i) - \Delta_{\Delta G^i}(W_T, S')\right] \\
&\hspace{8cm} \leq \frac{2L}{n}\sum_{i=1}^n \mathbb{E}\left[\|W_T - \mathbb{E}\left[W_T \mid Z_{-i}\right]\|\right].
\end{aligned}
$$

*If the algorithm is a GD algorithm using projected subgradients of step size $\eta$ and step count $T$, we have*

$$
\mathrm{gen}(\mu^n, P_{W|S}) \leq 8L^2\sqrt{T}\eta + \frac{8L^2T\eta}{n}.
$$

*Therefore, we have the following worst-case generalization error bound for CLB and GD:*

$$
\sup_{(\mathcal{W}, \mathcal{Z}, \ell) \in \mathcal{C}_{L,D}} \sup_{\mu \in \mathcal{M}_1(\mathcal{Z})} \mathrm{gen}(\mu^n, P_{W|S}^{GD_{n,\eta,T}}) \leq 8L^2\sqrt{T}\eta + \frac{8L^2T\eta}{n}.
$$

**Proof** Let $(\mathcal{W}, \mathcal{Z}, \ell) \in \mathcal{C}_{L,D}$ be an SCO problem and $\mu \in \mathcal{M}_1(\mathcal{Z})$ be a distribution of samples. Therefore, the sample loss $\ell$, the empirical loss $\hat{\mathcal{L}}_S(\cdot)$ and the population loss $\hat{\mathcal{L}}_S(\cdot)$ are $L$-Lipschitz w.r.t. the weight and $\mathcal{W}$ is bounded with a diameter $D$ and convex. Let $W_{0:T}$ be the trajectory given by GD, where $W_t \in \mathcal{W}$.

The proof essentially resembles the uniform stability argument of Bassily et al. (2020) through the superior expressivity added to the MI bounds by the omniscient trajectory. Since uniform stability

considers replacing one sample in the training set, we need to focus on one sample instead of the entire training set. To keep other samples "unchanged" in the replacement, we also need to condition on other samples. Therefore, we will start from the "individual sample" technique (Bu et al., 2020). This observation motivates us to use Corollary D.1. However, losses in CLB problems are not sub-Gaussian in general. Therefore, we need variants similar to Theorem 11 of Haghifam et al. (2023) that replaces sub-Gaussianity with Lipschitzness and boundedness. By repeating the proof of Corollary D.1 but with Theorem 11 of Haghifam et al. (2023) instead of Lemma 1, we obtain

$$
\text{gen}(\mu^n, P_{W|S}) \leq \frac{LD}{n} \sum_{i=1}^{n} \sqrt{2I\left(W_T + \sum_{t=1}^{T} \Delta g_t^i ; Z_i \mid Z_{-i}\right)}
$$
$$
+ \frac{1}{n} \sum_{i=1}^{n} \mathbb{E}\left[\Delta_{\Gamma_T^i}(W_T, Z_i) - \Delta_{\Gamma_T^i}(W_T, S')\right], \quad \text{(D.4)}
$$

where $\left\{\Delta g_{1:T}^i\right\}_{i=1}^{n}$ is a family of omniscient perturbations, which moves the terminal within $\mathcal{W}$, i.e., $W_T + \Gamma_T^i \in \mathcal{W}$.

**Remark D.4** *The SGLD-like trajectory is not used in this proof. This is because the SGLD-like trajectory is used only to bound the MI of the omniscient trajectory. As one will see later, the omniscient trajectory in this proof has a very special form (constant) with a trivial MI bound (0). As a result, an MI bound through the SGLD-like trajectory is no longer needed.*

This proof proceeds by bounding the penalty terms in Eq. (D.4) with the help of the inherited Lipschitzness of $\ell$ in the empirical and the population loss:

$$
\left|\Delta_{\Gamma_T^i}(W_T, Z_i)\right| := \left|\hat{\mathcal{L}}_{Z_i}(W_T + \Gamma_T^i) - \hat{\mathcal{L}}_{Z_i}(W_T)\right| \leq L\left\|\Gamma_T^i\right\|,
$$
$$
\left|\mathbb{E}\left[\Delta_{\Gamma_T^i}(W_T, S')\right]\right| := \left|\mathbb{E}\left[\mathcal{L}_\mu(W_T + \Gamma_T^i) - \mathcal{L}_\mu(W_T)\right]\right| \leq L\mathbb{E}\left[\left\|\Gamma_T^i\right\|\right].
$$

Therefore, we have the following bound for the penalty terms:

$$
\left|\frac{1}{n} \sum_{i=1}^{n} \mathbb{E}\left[\Delta_{\Gamma_T^i}(W_T, Z_i) - \Delta_{\Gamma_T^i}(W_T, S')\right]\right| \leq \frac{2L}{n} \sum_{i=1}^{n} \mathbb{E}\left[\left\|\Gamma_T^i\right\|\right].
$$

Plugging this bound to Eq. (D.4) leads to

$$
\text{gen}(\mu^n, P_{W|S}) \leq \frac{LD}{n} \sum_{i=1}^{n} \sqrt{I(W_T + \Gamma_T^i ; Z_i \mid Z_{-i})} + \frac{2L}{n} \sum_{i=1}^{n} \mathbb{E}\left[\left\|\Gamma_T^i\right\|\right].
$$

By setting $\Delta g_t^i = g_t - \mathbb{E}\left[g_t \mid Z_{-i}\right]$, we have $W_T + \Gamma_T^i = \mathbb{E}\left[W_T \mid Z_{-i}\right]$. By convexity of $\mathcal{W}$ and that $W_T \in \mathcal{W}$, we have $W_T + \Gamma_T^i \in \mathcal{W}$. Notably, given $Z_{-i}$, $W_T + \Gamma_T^i$ is constant regardless of $Z_i$. Therefore, we have $I(W_T + \Gamma_T^i ; Z_i \mid Z_{-i}) = 0$ and

$$
\text{gen}(\mu^n, P_{W|S}) \leq \frac{2L}{n} \sum_{i=1}^{n} \mathbb{E}\left[\left\|\Gamma_T^i\right\|\right]. \quad \text{(D.5)}
$$

Note that $\mathbb{E}\left[\|W_T - \mathbb{E}\left[W_T \mid Z_{-i}\right]\| \mid Z_{-i}\right]$ is the expected distance to the center given $Z_{-i}$. By Corollary A.1, this value is smaller than $\mathbb{E}\left[\|W_T - W_T'\| \mid Z_{-i}\right]$, where $W_T'$ is the terminal weight trained by the same $Z_{-i}$ and an independently sampled $i$-th sample.

Now we turn to GD. By Theorem 3.2 of Bassily et al. (2020), given $Z_{-i}$, $\|W_T - W_T'\|$ of GD has a data-agnostic upperbound $4L\sqrt{T}\eta + \frac{4LT\eta}{n}$. Plugging this bound, we obtain

$$
\text{gen}(\mu^n, P_{W|S}) \leq \frac{2L}{n} \sum_{i=1}^{n} \mathbb{E}_{Z_{-i}}\left[\mathbb{E}\left[\|W_T - \mathbb{E}\left[W_T \mid Z_{-i}\right]\| \mid Z_{-i}\right]\right]
$$
$$
\leq \frac{2L}{n} \sum_{i=1}^{n} \mathbb{E}_{Z_{-i}}\left[\mathbb{E}\left[\|W_T - W_T'\| \mid Z_{-i}\right]\right]
$$
$$
\leq 8L^2\sqrt{T}\eta + \frac{8L^2T\eta}{n}.
$$

**Remark D.5** *Although adding the omniscient trajectory can address the limitation of representative information-theoretic bounds on CLB problems, we currently do not know whether it is exactly our technique that makes it happen. This is because, between the Gaussian-perturbed individual-sample (C)MI bound considered by Haghifam et al. (2023) and the omniscient bound, there exist bounds derived by non-isotropic Gaussian perturbations, bounds by non-Gaussian but general independent perturbations, and bounds by general weight-dependent perturbations (Rate-Distortion bounds (Sefidgaran et al., 2022)). It is possible some of these bounds can already address the limitation but we have not obtained any positive or negative results for them before the submission deadline. Nevertheless, our technique makes the proof extremely simple and without it the proof will be at least much longer. For example, it will be harder to make the (conditional) MI diminish without knowing* $Z_{-i}$.

### D.5 EXTENSION TO $\epsilon$-LEARNERS ON CLB PROBLEMS

Attias et al. (2024) have found that any $\epsilon$-learners have CMI of at least $\Omega(1/\epsilon)$ on CLB problems, indicating a CMI-accuracy trade-off. Here, a learning algorithm is an $\epsilon$-learner is for CLB problem $(\mathcal{W}, \mathcal{Z}, \ell)$ if fir every data distribution $\mu$ over $\mathcal{Z}$, the excess generalization risk of the algorithm is at most $\epsilon$.

Our Theorem 3 and its proof have given an intuitive alternative to the trade-off: although GD or stable learners have large CMI themselves, they are quite close to learners with low CMI. However, Theorem 3 assumes GD or stable algorithms, yet Attias et al. (2024)'s trade-off covers more general algorithms. Therefore, we explore whether our technique and alternative can extend to more (expectation-)$\epsilon$-learners in Theorem 4. It states our technique can indeed extends to more $\epsilon$-learners under some assumptions. It also states if one sees the omniscient trajectory augmented MI as a new information measure, then the information-accuracy trade-off is penetrated because both can vanish as $n \to \infty$. However, the result is still partial and preliminary, because we only covers $\epsilon$-learners that are also $O(\epsilon)$-optimizers, *i.e.,* they are "well-behaved" in the sense that its excess optimization error is not too large compared to the excess generalization error $\epsilon$. Nevertheless, we believe this assumption is rather gentle for well-behaved learners used practically, *e.g.,* deep models.

**Theorem 4** *Assume $d \in \mathbb{N}^+$, $L, D > 0$ and let $(\mathcal{W}, \mathcal{Z}, \ell) \in \mathcal{C}_{L,D}$ be a CLB problem. Assume for any $w \in \mathcal{W}$ and $z \in \mathcal{Z}$, we have $\ell(w, z) \in [-LD, +LD]$. If not, shift the loss functions. This shifting does not affect the excess generalization risks or the excess optimization errors, which this theorem mainly assume on.*

*Let $\epsilon > 0$ be a function of sample number $n$. Let $\{\mathcal{A}_n : \mathcal{Z}^n \to \mathcal{W}\}$ be a family of expectation-$\epsilon$-learners for $(\mathcal{W}, \mathcal{Z}, \ell)$. That is, for every sufficiently large $n$, for every distribution $\mu$ over $\mathcal{Z}$, one has $\mathbb{E}_{S \sim \mu^n}[\mathcal{L}_\mu(\mathcal{A}_n(S))] - \inf_{w \in \mathcal{W}} \mathcal{L}_\mu(w) \le \epsilon$.*

*Assume $\{\mathcal{A}_n\}$ is also an $O(\epsilon)$-optimizer. That is, $\mathcal{A}_n$ has an excess optimization error satisfying $\mathbb{E}_{S \sim \mu^n}\left[\hat{\mathcal{L}}_S(\mathcal{A}_n(S)) - \inf_w \hat{\mathcal{L}}_S(w)\right] \le O(\epsilon)$ for any sufficiently large $n$.*

*Assume $n \in \mathbb{N}^+$ is large enough so that $\epsilon$ bounds the excess generalization risk and $O(\epsilon)$ bounds the excess optimization error. Then, we have*

$$\text{gen}(\mu^n, P_{W|S}) \le \inf_{\{\Delta G^i\}_{i=1}^n} \frac{LD}{n} \sum_{i=1}^n \sqrt{2I\left(W + \Delta G^i; Z_i \mid Z_{-i}\right)} + \frac{1}{n} \sum_{i=1}^n \mathbb{E}\left[\Delta_{\Delta G^i}(W, Z_i) - \Delta_{\Delta G^i}(W, S')\right]$$

$$\le O(\epsilon) + O\left(\frac{LD}{\sqrt{n}}\right),$$

*where $W := \mathcal{A}_n(S)$ is the output of the algorithm, and $\{\Delta G^i\}_{i=1}^n$ is a family of (one-step) omniscient perturbations that additionally depend on $i$ and $Z_{-i}$.*

**Remark D.6** *We assume the algorithm $\{\mathcal{A}_n\}$ is deterministic but the extension to random ones is straightforward.*

**Proof** After repeating the initial steps of the proof and obtaining Eq. (D.4), we have

$$\text{gen}(\mu^n, P_{W|S}) \leq \frac{LD}{n} \sum_{i=1}^{n} \sqrt{2I\left(W + \Delta G^i; Z_i \mid Z_{-i}\right)}$$

$$+ \frac{1}{n} \sum_{i=1}^{n} \mathbb{E}\left[\Delta_{\Delta G^i}(W, Z_i) - \Delta_{\Delta G^i}(W, S')\right]$$

$$\leq \frac{LD}{n} \sum_{i=1}^{n} \sqrt{2I\left(W + \Delta G^i; Z_i \mid Z_{-i}\right)} + \frac{1}{n} \sum_{i=1}^{n} \mathbb{E}\left[\left|\hat{\mathcal{L}}_S(W + \Delta G^i) - \hat{\mathcal{L}}_S(W)\right|\right]$$

$$+ \frac{1}{n} \sum_{i=1}^{n} \mathbb{E}\left[\left|\mathcal{L}_\mu(W + \Delta G^i) - \mathcal{L}_\mu(W)\right|\right],$$

After applying a similar omniscient perturbation $\Delta G^i := \mathbb{E}\left[W \mid Z_{-i}\right] - W$ as in the proof of Theorem 3, we have

$$\text{gen}(\mu^n, P_{W|S}) \leq \frac{LD}{n} \sum_{i=1}^{n} \sqrt{2I\left(\mathbb{E}\left[W \mid Z_{-i}\right]; Z_i \mid Z_{-i}\right)} + \frac{1}{n} \sum_{i=1}^{n} \mathbb{E}\left[\left|\hat{\mathcal{L}}_S(\mathbb{E}\left[W \mid Z_{-i}\right]) - \hat{\mathcal{L}}_S(W)\right|\right]$$

$$+ \frac{1}{n} \sum_{i=1}^{n} \mathbb{E}\left[\left|\mathcal{L}_\mu(\mathbb{E}\left[W \mid Z_{-i}\right]) - \mathcal{L}_\mu(W)\right|\right]$$

$$= \frac{1}{n} \sum_{i=1}^{n} \mathbb{E}\left[\left|\hat{\mathcal{L}}_S(\mathbb{E}\left[W \mid Z_{-i}\right]) - \hat{\mathcal{L}}_S(W)\right|\right] + \frac{1}{n} \sum_{i=1}^{n} \mathbb{E}\left[\left|\mathcal{L}_\mu(\mathbb{E}\left[W \mid Z_{-i}\right]) - \mathcal{L}_\mu(W)\right|\right].$$

$$(D.6)$$

We first bound the population loss difference by applying Lemma A.3 to the difference with convex $f(w) = \mathcal{L}_\mu(w) - \mathcal{L}_\mu(w^*) \geq 0$, where $w^*$ is the weight with optimal generalization risk:

$$\mathbb{E}\left[\left|\mathcal{L}_\mu(\mathbb{E}\left[W \mid Z_{-i}\right]) - \mathcal{L}_\mu(W)\right|\right] = \mathbb{E}_{Z_{-i}}\left[\mathbb{E}\left[\left|(\mathcal{L}_\mu(\mathbb{E}\left[W \mid Z_{-i}\right]) - \mathcal{L}_\mu(w^*)) - (\mathcal{L}_\mu(W) - \mathcal{L}_\mu(w^*))\right| \mid Z_{-i}\right]\right]$$

$$\leq 2\mathbb{E}_{Z_{-i}}\left[\mathbb{E}\left[f(W) \mid Z_{-i}\right]\right] = 2(\mathbb{E}\left[\mathcal{L}_\mu(W)\right] - \mathcal{L}_\mu(w^*)) \leq 2\epsilon.$$

Now we bound the empirical loss difference. Let $w_s^*$ be the *empirical* risk minimizer of training set $s$. The main difficulty is that different $W$ corresponds to different $S$, and one cannot find an $f$ to be $\hat{\mathcal{L}}_S(\cdot) - \hat{\mathcal{L}}_S(w_S^*)$ and $\hat{\mathcal{L}}_{S'}(\cdot) - \hat{\mathcal{L}}_{S'}(w_{S'}^*)$ at the same time. Fortunately, with the individual technique, we can put weights corresponding to $Z_{-i}$ together. Most of their training set is the same, while the only different sample only contributes $1/n$ of the loss, which vanishes as $n \to \infty$. Therefore, we have

$$\mathbb{E}\left[\left|\hat{\mathcal{L}}_S(\mathbb{E}\left[W \mid Z_{-i}\right]) - \hat{\mathcal{L}}_S(W)\right|\right] = \mathbb{E}_{Z_{-i}}\left[\mathbb{E}\left[\left|\hat{\mathcal{L}}_S(\mathbb{E}\left[W \mid Z_{-i}\right]) - \hat{\mathcal{L}}_S(W)\right| \mid Z_{-i}\right]\right],$$

where

$$\mathbb{E}\left[\left|\hat{\mathcal{L}}_S(\mathbb{E}\left[W \mid Z_{-i}\right]) - \hat{\mathcal{L}}_S(W)\right| \mid Z_{-i}\right] \leq \frac{n-1}{n} \mathbb{E}\left[\left|\hat{\mathcal{L}}_{Z_{-i}}(\mathbb{E}\left[W \mid Z_{-i}\right]) - \hat{\mathcal{L}}_{Z_{-i}}(W)\right| \mid Z_{-i}\right]$$

$$+ \frac{1}{n} \mathbb{E}\left[\left|\hat{\mathcal{L}}_{Z_i}(\mathbb{E}\left[W \mid Z_{-i}\right]) - \hat{\mathcal{L}}_{Z_i}(W)\right| \mid Z_{-i}\right]$$

(Lipschitzness and boundedness of CLB problems)

$$\leq \frac{n-1}{n} \mathbb{E}\left[\left|\hat{\mathcal{L}}_{Z_{-i}}(\mathbb{E}\left[W \mid Z_{-i}\right]) - \hat{\mathcal{L}}_{Z_{-i}}(W)\right| \mid Z_{-i}\right] + \frac{LD}{n}.$$

Applying Lemma A.3 with convex $f(w) \coloneqq \hat{\mathcal{L}}_{Z_{-i}}(w) - \hat{\mathcal{L}}_{Z_{-i}}(w^*_{Z_{-i}}) \geq 0$ leads to

$$\mathbb{E}\left[\left|\hat{\mathcal{L}}_S(\mathbb{E}\left[W \mid Z_{-i}\right]) - \hat{\mathcal{L}}_S(W)\right| \mid Z_{-i}\right]$$

$$\leq \frac{n-1}{n} \cdot 2\mathbb{E}\left[f(W) \mid Z_{-i}\right] + \frac{LD}{n}$$

$$\leq \frac{n-1}{n} \cdot 2\mathbb{E}\left[\hat{\mathcal{L}}_{Z_{-i}}(W) - \hat{\mathcal{L}}_{Z_{-i}}(w^*_{Z_{-i}}) \mid Z_{-i}\right] + \frac{LD}{n}$$

$$\left(\frac{1}{n}\mathbb{E}\left[\hat{\mathcal{L}}_{Z_i}(W) - \hat{\mathcal{L}}_{Z_i}(w^*_{Z_i}) \mid Z_{-i}\right] \geq 0\right)$$

$$\leq 2\mathbb{E}\left[\hat{\mathcal{L}}_S(W) - \left(\frac{n-1}{n}\hat{\mathcal{L}}_{Z_{-i}}(w^*_{Z_{-i}}) + \frac{1}{n}\hat{\mathcal{L}}_{Z_i}(w^*_{Z_i})\right) \mid Z_{-i}\right] + \frac{LD}{n}$$

Taking expectation over $Z_{-i}$ leads to

$$\mathbb{E}\left[\left|\hat{\mathcal{L}}_S(\mathbb{E}\left[W \mid Z_{-i}\right]) - \hat{\mathcal{L}}_S(W)\right|\right]$$

$$\leq 2\mathbb{E}\left[\hat{\mathcal{L}}_S(W)\right] - 2\left(\frac{n-1}{n}\mathbb{E}\left[\hat{\mathcal{L}}_{S_{n-1}}(w^*_{S_{n-1}})\right] + \frac{1}{n}\mathbb{E}\left[\hat{\mathcal{L}}_Z(w^*_Z)\right]\right) + \frac{LD}{n}, \quad \text{(D.7)}$$

where $S_{n-1} \sim \mu^{n-1}$. We need to relate $\frac{n-1}{n}\mathbb{E}\left[\hat{\mathcal{L}}_{S_{n-1}}(w^*_{S_{n-1}})\right] + \frac{1}{n}\mathbb{E}\left[\hat{\mathcal{L}}_Z(w^*_Z)\right]$ to $\mathbb{E}\left[\hat{\mathcal{L}}_S(w^*_S)\right]$. We can bound $\frac{1}{n}\mathbb{E}\left[\hat{\mathcal{L}}_Z(w^*_Z)\right]$ by $\frac{LD}{n}$ again by the assumption that losses are bounded in $[-LD, +LD]$, leaving $\frac{n-1}{n}\mathbb{E}\left[\hat{\mathcal{L}}_{S_{n-1}}(w^*_{S_{n-1}})\right]$ and $\mathbb{E}\left[\hat{\mathcal{L}}_S(w^*_S)\right]$, which somehow forms the stability of the empirical risk minimizer. Since (projected) GD with fixed initialization has been proved to be stable and has been proved to approximate the empirical minimizer, we use (projected) GD to bridge the empirical minimizers.

According to Orabona (2020) and Eq.(1) of Haghifam et al. (2023), on CLB problems, (projected) GD algorithm $\text{GD}_{\eta,T} : \mathcal{Z}^* \to \mathcal{W}$ with step size $\eta$ and step count $T$ and a fixed initialization has an excess optimization error $\frac{D^2}{2\eta T} + \frac{(\log T + 2)\eta L^2}{2}$. As a result, we can approximate the empirical minimizers using GDs with errors

$$\left|\mathbb{E}\left[\hat{\mathcal{L}}_{S_{n-1}}(w^*_{S_{n-1}})\right] - \mathbb{E}\left[\hat{\mathcal{L}}_{S_{n-1}}(\text{GD}_{\eta_{n-1},T_{n-1}}(S_{n-1}))\right]\right| \leq \frac{D^2}{2\eta_{n-1}T_{n-1}} + \frac{(\log T_{n-1} + 2)\eta_{n-1}L^2}{2},$$

$$\left|\mathbb{E}\left[\hat{\mathcal{L}}_S(w^*_S)\right] - \mathbb{E}\left[\hat{\mathcal{L}}_{S_n}(\text{GD}_{\eta_n,T_n}(S))\right]\right| \leq \frac{D^2}{2\eta_n T_n} + \frac{(\log T_n + 2)\eta_n L^2}{2},$$

for any $(\eta_{n-1}, T_{n-1})$ and any $(\eta_n, T_n)$. We assign $\eta_{n-1} = \eta_n, T_{n-1} = T_n$ and select them suitably as in Haghifam et al. (2023)'s Eq.(3), which bounds the approximation errors by $O(LD/\sqrt{n})$ at the same time:

$$\left|\mathbb{E}\left[\hat{\mathcal{L}}_{S_{n-1}}(w^*_{S_{n-1}})\right] - \mathbb{E}\left[\hat{\mathcal{L}}_{S_{n-1}}(\text{GD}_{\eta_n,T_n}(S_{n-1}))\right]\right| \leq O\left(\frac{LD}{\sqrt{n}}\right),$$

$$\left|\mathbb{E}\left[\hat{\mathcal{L}}_S(w^*_S)\right] - \mathbb{E}\left[\hat{\mathcal{L}}_{S_n}(\text{GD}_{\eta_n,T_n}(S))\right]\right| \leq O\left(\frac{LD}{\sqrt{n}}\right).$$

We then need to relate $\mathbb{E}\left[\hat{\mathcal{L}}_{S_{n-1}}(\text{GD}_{\eta_n,T_n}(S_{n-1}))\right]$ and $\mathbb{E}\left[\hat{\mathcal{L}}_{S_n}(\text{GD}_{\eta_n,T_n}(S))\right]$, which is the removal-based stability of GD. To this end, for each $s_{n-1}$, we construct an artificial sample $z_{S_{n-1}}$ such that

$$\ell(w, z_{S_{n-1}}) \coloneqq \hat{\mathcal{L}}_{S_{n-1}}(w),$$

and denote $S^+_{n-1} \coloneqq S_{n-1} \cup \{z_{S_{n-1}}\}$. By construction, we have $\hat{\mathcal{L}}_{S^+_{n-1}}(w) = \hat{\mathcal{L}}_{S_{n-1}}(w)$ for any $w \in \mathcal{W}$, *i.e.,* the optimizations using $S_{n-1}$ and $S^+_{n-1}$ happen on the same loss landscape. Since GD relies on (sub-)gradients and thus only relies on loss landscape, we have $\text{GD}_{\eta_n,T_n}(S_{n-1}) = $

$\text{GD}_{\eta_n, T_n}(S_{n-1}^+)$. If we pair $S_{n-1}$ and $S_n$, then $S_{n-1}^+$ and $S_n$ only differs by one sample, allowing us to apply the replacement-based uniform stability for GD on CLB from Bassily et al. (2020):

$$\left| \mathbb{E}\left[ \hat{\mathcal{L}}_{S_{n-1}}(\text{GD}_{\eta_n, T_n}(S_{n-1})) \right] - \mathbb{E}\left[ \hat{\mathcal{L}}_{S_n}(\text{GD}_{\eta_n, T_n}(S)) \right] \right|$$

$$\leq \mathbb{E}_{S_{n-1}} \left[ \mathbb{E}\left[ \left| \hat{\mathcal{L}}_{S_{n-1}}(\text{GD}_{\eta_n, T_n}(S_{n-1})) - \hat{\mathcal{L}}_{S_n}(\text{GD}_{\eta_n, T_n}(S)) \right| \mid S_{n-1} \right] \right]$$

$$\leq L \cdot \mathbb{E}_{S_{n-1}} \left[ \mathbb{E}\left[ \|\text{GD}_{\eta_n, T_n}(S_{n-1}) - \text{GD}_{\eta_n, T_n}(S)\| \mid S_{n-1} \right] \right]$$

$$= L \cdot \mathbb{E}_{S_{n-1}} \left[ \mathbb{E}\left[ \|\text{GD}_{\eta_n, T_n}(S_{n-1}^+) - \text{GD}_{\eta_n, T_n}(S)\| \mid S_{n-1} \right] \right]$$

$$\leq O\left( L^2 \sqrt{T_n} \eta_n + \frac{L^2 T_n \eta_n}{n} \right).$$

With the same selection of $(T_n, \eta_n)$, the above difference can be bounded by $O\left( \frac{LD}{\sqrt{n}} \right)$.

As a result, Eq. (D.7) can be bounded by

$$\mathbb{E}\left[ \left| \hat{\mathcal{L}}_S(\mathbb{E}\left[ W \mid Z_{-i} \right]) - \hat{\mathcal{L}}_S(W) \right| \right]$$

$$\leq 2 \left( \mathbb{E}\left[ \hat{\mathcal{L}}_S(W) \right] - \frac{n-1}{n} \mathbb{E}\left[ \hat{\mathcal{L}}_S(w_S^*) \right] \right) + O\left( \frac{LD}{n} \right) + O\left( \frac{LD}{\sqrt{n}} \right)$$

$$\leq 2 \left( \mathbb{E}\left[ \hat{\mathcal{L}}_S(W) \right] - \mathbb{E}\left[ \hat{\mathcal{L}}_S(w_S^*) \right] \right) + O\left( \frac{LD}{n} \right) + O\left( \frac{LD}{\sqrt{n}} \right)$$

$$\leq 2 \cdot O(\epsilon) + O\left( \frac{LD}{n} \right) + O\left( \frac{LD}{\sqrt{n}} \right).$$

Plugging everything back to Eq. (D.6) finishes the proof. ∎

### D.6 EXTENSION TO SGD AND SMOOTH LOSSES

In this subsection, we apply our technique to SGD under smooth losses. We will prove an omniscient information-theoretic bound and then show it recovers some existing stability-based bounds.

To prove the omniscient bound, we need a basic form of information-theoretic bounds like Lemma 1 and Theorem 11 of Haghifam et al. (2023). This is done by results from Lemma D.2 to Corollary D.5. After that, we make the basic bound omniscient in Theorem 5. Finally, we bound the omniscient bound by the stability-based bound in Proposition 3.

**Lemma D.2 (2-Wasserstain Distance Generalization Bound under Smoothness)** *Assume for any sample $z \in \mathcal{Z}$, $\ell(\cdot, z)$ is non-negative, differentiable in $\mathbb{R}^d$ and $\beta$-smooth, i.e., for any $w, w' \in \mathbb{R}^d$*

$$\|\nabla \ell(w, z) - \nabla \ell(w', z)\| \leq \beta \|w - w'\|.$$

*Then we have the following 2-Wasserstain-based information-theoretic (individual-sample) bound:*

$$\text{gen}(\mu^n, P_{W|S}) \leq \frac{\beta}{\gamma} \mathbb{E}\left[ \hat{\mathcal{L}}_S(W) \right] + \frac{\beta + \gamma}{2n} \sum_{i=1}^{n} \mathbb{E}_{Z_i} \left[ \mathbb{W}_2 \left( P_{W|Z_i}, P_W \right) \right],$$

*where $\gamma > 0$ is a constant, and $\mathbb{W}_2\left( \cdot, \cdot \right)$ denotes 2-Wasserstein distance, the optimal transport under squared $L_2$ norm.*

**Proof** Let constant $\gamma > 0$. Given any index $i$ and any instance of the $i$-th training sample $z_i$, let $\pi_{z_i}^\epsilon$ be the coupling that approximates the 2-Wasserstein distance $\mathbb{W}_2\left( P_{W|Z_i = z_i}, P_W \right)$ between

$P_{W|Z_i=z_i}$ and $P_W$ by an error at most $\epsilon > 0$. Then for any $\epsilon > 0$, we have

$$\text{gen}(\mu^n, P_{W|S}) = \frac{1}{n} \sum_{i=1}^{n} \mathbb{E}_{(W,Z_i),W'} \left[ \ell(W', Z_i) - \ell(W, Z_i) \right]$$

$$= \frac{1}{n} \sum_{i=1}^{n} \mathbb{E}_{Z_i} \left[ \mathbb{E}_{(W,W') \sim \pi_{Z_i}^{\epsilon}} \left[ \ell(W', Z_i) - \ell(W, Z_i) \right] \right]$$

$$\leq \frac{1}{n} \sum_{i=1}^{n} \mathbb{E}_{Z_i} \left[ \mathbb{E}_{(W,W') \sim \pi_{Z_i}^{\epsilon}} \left[ (W' - W)^\top \nabla \ell(W, Z_i) + \frac{\beta}{2} \|W' - W\|^2 \right] \right].$$

We then follow Lei & Ying (2020) to handle the inner product as in their Appendix B:

$$(W' - W)^\top \nabla \ell(W, Z_i) \leq \|W' - W\| \cdot \|\nabla \ell(W, Z_i)\|$$

$$\leq \frac{\gamma}{2} \|W' - W\|^2 + \frac{1}{2\gamma} \|\nabla \ell(W, Z_i)\|^2.$$

Thanks to the self-bounding property of positive smooth functions (Lemma A.1 of Lei & Ying (2020)), we have $\|\nabla \ell(W, Z_i)\|^2 \leq 2\beta \cdot \ell(W, Z_i)$ and

$$(W' - W)^\top \nabla \ell(W, Z_i) \leq \frac{\gamma}{2} \|W' - W\|^2 + \frac{\beta}{\gamma} \ell(W, Z_i).$$

Plugging this back leads to

$$\text{gen}(\mu^n, P_{W|S}) \leq \frac{1}{n} \sum_{i=1}^{n} \mathbb{E}_{Z_i} \left[ \mathbb{E}_{(W,W') \sim \pi_{Z_i}^{\epsilon}} \left[ \frac{\beta}{\gamma} \ell(W, Z_i) + \frac{\beta + \gamma}{2} \|W' - W\|^2 \right] \right]$$

$$\leq \frac{\beta}{\gamma} \mathbb{E} \left[ \hat{\mathcal{L}}_S(W) \right] + \frac{\beta + \gamma}{2n} \sum_{i=1}^{n} \mathbb{E}_{Z_i} \left[ \mathbb{W}_2 \left( P_{W|Z_i}, P_W \right) + \epsilon \right].$$

By arbitrariness of $\epsilon > 0$, we have

$$\text{gen}(\mu^n, P_{W|S}) \leq \frac{\beta}{\gamma} \mathbb{E} \left[ \hat{\mathcal{L}}_S(W) \right] + \frac{\beta + \gamma}{2n} \sum_{i=1}^{n} \mathbb{E}_{Z_i} \left[ \mathbb{W}_2 \left( P_{W|Z_i}, P_W \right) \right].$$

∎

**Lemma D.3** *For any random variables $(X, Y, Z)$ such that $Y$ is independent of $Z$, we have*

$$\mathbb{E}_Y \left[ \mathbb{W}_2 \left( P_{X|Y}, P_X \right) \right] \leq \mathbb{E}_Z \left[ \mathbb{E}_{Y \sim P_{Y|Z}} \left[ \mathbb{W}_2 \left( P_{X|Y,Z}, P_{X|Z} \right) \right] \right].$$

**Proof** Let $\pi_{y,z}^{\epsilon}$ be the coupling that approximates the 2-Wasserstein distance $\mathbb{W}_2 \left( P_{X|y,z}, P_{X|y} \right)$ between $P_{X|y,z}$ and $P_{X|y}$ by an error at most $\epsilon > 0$. Then for any $\epsilon > 0$, we have

$$\mathbb{E}_Z \left[ \mathbb{E}_{Y \sim P_{Y|Z}} \left[ \mathbb{W}_2 \left( P_{X|Y,Z}, P_{X|Z} \right) \right] \right] = \mathbb{E}_Y \left[ \mathbb{E}_Z \left[ \mathbb{W}_2 \left( P_{X|Y,Z}, P_{X|Z} \right) \right] \right] \qquad \text{(Independence between } Y, Z)$$

$$\geq \mathbb{E}_Y \left[ \mathbb{E}_Z \left[ \mathbb{E}_{(X,X') \sim \pi_{Y,Z}^{\epsilon}} \left[ \|X - X'\|^2 \right] - \epsilon \right] \right]$$

$$= \mathbb{E}_Y \left[ \mathbb{E}_{(Z,X,X') \sim P_Z \circ \pi_{Y,Z}^{\epsilon}} \left[ \|X - X'\|^2 \right] \right] - \epsilon$$

$$\geq \mathbb{E}_Y \left[ \mathbb{W}_2 \left( P_{X|Y}, P_X \right) \right] - \epsilon.$$

The lemma follows the arbitrariness of $\epsilon > 0$. ∎

Lemma D.2 is very similar to the fact that $I(X; Y) \leq I(X; Y \mid Z)$ if $Y$ is independent of $Z$. Following this similarity, we write (conditional) expected 2-Wasserstain distances similar to (conditional) MI, or equivalently, replace the KL-divergence in MI with the Wasserstain distance to compare the prior and posterior:

**Definition 3** *For any random variables $(X, Y, Z)$, let*

$$I_{\mathbb{W}_2}(X; Y) := \mathbb{E}_Y \left[ \mathbb{W}_2 \left( P_{X|Y}, P_X \right) \right],$$
$$I_{\mathbb{W}_2}(X; Y \mid Z) := \mathbb{E}_Z \left[ \mathbb{E}_{Y \sim P_{Y|Z}} \left[ \mathbb{W}_2 \left( P_{X|Y,Z}, P_{X|Z} \right) \right] \right].$$

**Lemma D.4** *If $X$ and $Y$ are independent given $Z$, then $I_{\mathbb{W}_2}(X; Y \mid Z) = 0$.*

It leads to the following corollary:

**Corollary D.5 (Stability-Style 2-Wasserstain Generalization Bound)** *Under the same assumptions as Lemma D.2, we have*

$$\text{gen}(\mu^n, P_{W|S}) \leq \frac{\beta}{\gamma} \mathbb{E} \left[ \hat{\mathcal{L}}_S(W) \right] + \frac{\beta + \gamma}{2n} \sum_{i=1}^n I_{\mathbb{W}_2}(W; Z_i)$$

$$\leq \frac{\beta}{\gamma} \mathbb{E} \left[ \hat{\mathcal{L}}_S(W) \right] + \frac{\beta + \gamma}{2n} \sum_{i=1}^n I_{\mathbb{W}_2}(W; Z_i \mid Z_{-i})$$

**Proof** $Z_i$ is independent of $Z_{-i}$. ∎

**Theorem 5 (Omniscient 2-Wasserstain Bound under Smoothness)** *Assume $\ell$ is non-negative, differentiable and $\beta$-smooth. Let $\gamma_1 > 0, \gamma_2 > \beta$. Let $\left\{ \Delta G^i \right\}_{i=1}^n$ be a family of omniscient (output-weight) perturbations, each of which additionally depends on $\overline{Z}_{-i}$. Then we have*

$$\text{gen}(\mu^n, P_{W|S}) \leq \frac{\beta}{\gamma_1} \frac{1}{n} \sum_{i=1}^n \mathbb{E} \left[ \ell(W + \Delta G^i, Z_i) \right] + \frac{\beta + \gamma_1}{2n} \sum_{i=1}^n I_{\mathbb{W}_2}(W + \Delta G^i; Z_i \mid Z_{-i})$$

$$+ \frac{1}{n} \sum_{i=1}^n \mathbb{E} \left[ \Delta_{\Delta G^i}(W, Z_i) - \Delta_{\Delta G^i}(W, Z') \right],$$

*and*

$$\text{gen}(\mu^n, P_{W|S}) \leq \frac{1}{1 - \frac{\beta}{\gamma_2}} \left( \frac{\beta}{\gamma_1 n} \sum_{i=1}^n \mathbb{E} \left[ \ell(W + \Delta G^i, Z_i) \right] + \frac{\beta + \gamma_1}{2n} \sum_{i=1}^n I_{\mathbb{W}_2}(W + \Delta G^i; Z_i \mid Z_{-i}) \right.$$

$$\left. + \frac{2\beta}{\gamma_2} \mathbb{E} \left[ \hat{\mathcal{L}}_S(W) \right] + \frac{\beta + \gamma_2}{n} \sum_{i=1}^n \left\| \Delta G^i \right\|^2 \right). \tag{D.8}$$

**Proof** This proof is very similar to the proof of Theorem 3. By repeating the proof of Corollary D.1 but with Corollary D.5 instead of Lemma 1, we obtain the first inequality in the theorem statement:

$$\text{gen}(\mu^n, P_{W|S}) \leq \frac{\beta}{\gamma_1} \frac{1}{n} \sum_{i=1}^n \mathbb{E} \left[ \ell(W + \Delta G^i, Z_i) \right] + \frac{\beta + \gamma_1}{2n} \sum_{i=1}^n I_{\mathbb{W}_2}(W + \Delta G^i; Z_i \mid Z_{-i})$$

$$+ \frac{1}{n} \sum_{i=1}^n \mathbb{E} \left[ \Delta_{\Delta G^i}(W, Z_i) - \Delta_{\Delta G^i}(W, Z') \right].$$

The penalty terms can be bounded the same way as in Lemma D.2:

$$|\Delta_{\Delta G^i}(W, z)| \leq \left| (\Delta G^i)^\top \nabla \ell(W, z) \right| + \frac{\beta}{2} \left\| \Delta G^i \right\|^2$$

$$\leq \left\| \Delta G^i \right\| \left\| \nabla \ell(W, z) \right\| + \frac{\beta}{2} \left\| \Delta G^i \right\|^2$$

$$\leq \frac{\gamma_2}{2} \left\| \Delta G^i \right\|^2 + \frac{\beta}{\gamma_2} \ell(W, z) + \frac{\beta}{2} \left\| \Delta G^i \right\|^2$$

$$= \frac{\beta}{\gamma_2} \ell(W, z) + \frac{\beta + \gamma_2}{2} \left\| \Delta G^i \right\|^2.$$

Therefore, we have

$$\text{gen}(\mu^n, P_{W|S}) \leq \frac{\beta}{\gamma_1} \frac{1}{n} \sum_{i=1}^{n} \mathbb{E}\left[\ell(W + \Delta G^i, Z_i)\right] + \frac{\beta + \gamma_1}{2n} \sum_{i=1}^{n} I_{\mathbb{W}_2}(W + \Delta G^i; Z_i \mid Z_{-i})$$
$$+ \frac{\beta}{\gamma_2}\left(\mathbb{E}\left[\hat{\mathcal{L}}_S(W)\right] + \mathbb{E}\left[\mathcal{L}_\mu(W)\right]\right) + (\beta + \gamma_2)\frac{1}{n} \sum_{i=1}^{n} \mathbb{E}\left[\left\|\Delta G^i\right\|^2\right].$$

Population loss $\mathbb{E}\left[\mathcal{L}_\mu(W)\right]$ appears at the right of the inequality. To move it to the left, we pair it with a virtual empirical loss term and moving the consequent $\text{gen}(\mu^n, P_{W|S})$ to the left:

$$\text{gen}(\mu^n, P_{W|S}) \leq \frac{\beta}{\gamma_1} \frac{1}{n} \sum_{i=1}^{n} \mathbb{E}\left[\ell(W + \Delta G^i, Z_i)\right] + \frac{\beta + \gamma_1}{2n} \sum_{i=1}^{n} I_{\mathbb{W}_2}(W + \Delta G^i; Z_i \mid Z_{-i})$$
$$+ \frac{\beta}{\gamma_2}\left(2\mathbb{E}\left[\hat{\mathcal{L}}_S(W)\right] + \text{gen}(\mu^n, P_{W|S})\right) + \frac{\beta + \gamma_2}{n} \sum_{i=1}^{n} \mathbb{E}\left[\left\|\Delta G^i\right\|^2\right],$$

$$(1 - \frac{\beta}{\gamma_2})\text{gen}(\mu^n, P_{W|S}) \leq \frac{\beta}{\gamma_1} \frac{1}{n} \sum_{i=1}^{n} \mathbb{E}\left[\ell(W + \Delta G^i, Z_i)\right] + \frac{\beta + \gamma_1}{2n} \sum_{i=1}^{n} I_{\mathbb{W}_2}(W + \Delta G^i; Z_i \mid Z_{-i})$$
$$+ \frac{2\beta}{\gamma_2}\mathbb{E}\left[\hat{\mathcal{L}}_S(W)\right] + \frac{\beta + \gamma_2}{n} \sum_{i=1}^{n} \mathbb{E}\left[\left\|\Delta G^i\right\|^2\right].$$

Restricting $\gamma_2 > \beta$ allows us to divide the inequality by $1 - \frac{\beta}{\gamma_2}$ without changing the direction of the inequality:

$$\text{gen}(\mu^n, P_{W|S}) \leq \frac{1}{1 - \frac{\beta}{\gamma_2}}\left(\frac{\beta}{\gamma_1 n} \sum_{i=1}^{n} \mathbb{E}\left[\ell(W + \Delta G^i, Z_i)\right] + \frac{\beta + \gamma_1}{2n} \sum_{i=1}^{n} I_{\mathbb{W}_2}(W + \Delta G^i; Z_i \mid Z_{-i})\right.$$
$$\left. + \frac{2\beta}{\gamma_2}\mathbb{E}\left[\hat{\mathcal{L}}_S(W)\right] + \frac{\beta + \gamma_2}{n} \sum_{i=1}^{n} \mathbb{E}\left[\left\|\Delta G^i\right\|^2\right]\right).$$

$\blacksquare$

Now that we have proved the omniscient bound for smooth losses, we turn to recovering some existing stability-based bounds.

**Proposition 3** *Under the same setting as Theorem 5, we have*

$$\text{gen}(\mu^n, P_{W|S}) \leq \inf_{\{\Delta G^i\}, \gamma_1 > 0, \gamma_2 > \beta}[\textit{RHS of Eq. (D.8)}]$$
$$\leq \inf_{\gamma_2 > \beta} \frac{1}{1 - \frac{\beta}{\gamma_2}}\left(\frac{2\beta}{\gamma_2}\mathbb{E}\left[\hat{\mathcal{L}}_S(W)\right] + \frac{\beta + \gamma_2}{2} \cdot \epsilon^{\text{stability}}\right),$$

*where*

$$\epsilon^{\text{stability}} := \mathbb{E}_{Z_{-i}, V}\left[\frac{1}{n} \sum_{i=1}^{n} \mathbb{E}\left[\left\|W' - W\right\|^2 \mid Z_{-i}, V\right]\right]$$

*(rephrased in our notation) is exactly the $\ell_2$ on-average model stability in Definition 4 of Lei & Ying (2020).*

**Remark D.7** *Proposition 3 recovers the relationship between stability and generalization in Lei & Ying (2020)'s Theorem 2(b) up to constants.*

**Proof** After setting $\Delta G^i = -W + \mathbb{E}\left[W \mid Z_{-i}, V\right]$ as a function of $Z_{-i}$ and $V$, we have $W + \Delta G^i = \mathbb{E}\left[W \mid Z_{-i}, V\right]$, which is a function of $Z_i$-independent $Z_{-i}$ and $V$. As a result, $W + \Delta G^i$ is

independent of $Z_i$ and $I_{\mathbb{W}_2}(W + \Delta G^i; Z_i \mid Z_{-i}) = 0$ according to Lemma D.4. Therefore, we have

$$\text{gen}(\mu^n, P_{W|S}) \leq \inf_{\{\Delta G^i\}, \gamma_1 > 0, \gamma_2 > 0} [\text{RHS of Eq. (D.8)}]$$

$$\leq \inf_{\gamma_1 > 0, \gamma_2 > \beta} \frac{1}{1 - \frac{\beta}{\gamma_2}} \left( \frac{\beta}{\gamma_1 n} \sum_{i=1}^{n} \mathbb{E} \left[ \ell(W + \Delta G^i, Z_i) \right] + \frac{2\beta}{\gamma_2} \mathbb{E} \left[ \hat{\mathcal{L}}_S(W) \right] + \frac{\beta + \gamma_2}{n} \sum_{i=1}^{n} \mathbb{E} \left[ \left\| \Delta G^i \right\|^2 \right] \right)$$

$$\leq \inf_{\gamma_2 > \beta} \frac{1}{1 - \frac{\beta}{\gamma_2}} \left( \frac{2\beta}{\gamma_2} \mathbb{E} \left[ \hat{\mathcal{L}}_S(W) \right] + \frac{\beta + \gamma_2}{n} \sum_{i=1}^{n} \mathbb{E} \left[ \| \mathbb{E} \left[ W \mid Z_{-i}, V \right] - W \|^2 \right] \right) \qquad (\gamma_1 \to +\infty)$$

With a closer look, one can find on-average model stability (Lei & Ying, 2020) term at RHS:

$$\frac{1}{n} \sum_{i=1}^{n} \mathbb{E} \left[ \| \mathbb{E} \left[ W \mid Z_{-i}, V \right] - W \|^2 \right] = \frac{1}{n} \sum_{i=1}^{n} \mathbb{E}_{Z_{-i}, V} \left[ \mathbb{E} \left[ \| \mathbb{E} \left[ W' \mid Z_{-i}, V \right] - W \|^2 \mid Z_{-i}, V \right] \right]$$

$$= \frac{1}{2n} \sum_{i=1}^{n} \mathbb{E}_{Z_{-i}, V} \left[ \mathbb{E} \left[ \| W' - W \|^2 \mid Z_{-i}, V \right] \right] = \frac{1}{2} \underbrace{\mathbb{E}_{S, S', V} \left[ \frac{1}{n} \sum_{i=1}^{n} \left\| \mathcal{A}(S, V) - \mathcal{A}(S^{(i)}, V) \right\|^2 \right]}_{\ell_2 \text{ on-average model stability in Definition 4 of Lei & Ying (2020)}},$$

where the second step follows Lemma A.2, $\mathcal{A}(\cdot, v)$ denotes the SGD when the random seed is $v$ and $S^{(i)}$ means replacing the $i$-th sample of $S$ with the $i$-th sample from $S'$.

$\blacksquare$

Now that we have recovered stability arguments, we can directly borrow stability of SGD to derive excess risk bounds. The following results are based on the on-average model stability bound derived by Lei & Ying (2020).

**Proposition 4** *Assume the loss is non-negative, convex and $\beta$-smooth. Assume the training algorithm is projected SGD that starts from a fixed $W_0 := w_0 \in \mathcal{W}$ and runs $T$ steps with non-increasing step sizes $\{\eta_t\}_{t=1}^{T+1}$ such that $\eta_t \leq 1/2\beta$. Then for any $\gamma > \beta$, we have the following excess risk bound:*

$$\mathbb{E} \left[ \mathcal{L}_\mu(W^{\text{acc}}) - \mathcal{L}_\mu(w^*) \right]$$

$$\leq \frac{2\beta}{\gamma_2 - \beta} \mathcal{L}_\mu(w^*) + \frac{1 + T/n}{n} \frac{4e\gamma_2\beta(\beta + \gamma_2)}{\gamma_2 - \beta} \left( \eta_1 \|w^*\|^2 + 2 \sum_{t=0}^{T} \eta_{t+1} \left( \sum_{\tau=0}^{t-1} \eta_{\tau+1}^2 \mathcal{L}_\mu(w^*) \right) / \sum_{\tau=0}^{T} \eta_{\tau+1} \right)$$

$$+ \frac{\gamma_2 + \beta}{\gamma_2 - \beta} \left( (1/2 + \beta\eta_1) \|w^*\|^2 + 2\beta \sum_{t=0}^{T} \eta_{t+1}^2 \mathcal{L}_\mu(w^*) \right) / \sum_{\tau=0}^{T} \eta_{\tau+1},$$

*where $W^{\text{acc}}$ is the accumulated weight*

$$W^{\text{acc}} := \frac{\sum_{t=0}^{T} \eta_{t+1} W_t}{\sum_{t=0}^{T} \eta_{t+1}}.$$

**Remark D.8** *In separable settings, i.e., when $\mathcal{L}_\mu(w^*) = 0$, the excess risk bound simplifies to*

$$\mathbb{E} \left[ \mathcal{L}_\mu(W^{\text{acc}}) - \mathcal{L}_\mu(w^*) \right]$$

$$\leq \frac{1 + T/n}{n} \frac{4e\gamma_2\beta(\beta + \gamma_2)}{\gamma_2 - \beta} \left( \eta_1 \|w^*\|^2 \right) + \frac{\gamma_2 + \beta}{(\gamma_2 - \beta) \sum_{\tau=0}^{T} \eta_{\tau+1}} \left( (1/2 + \beta\eta_1) \|w^*\|^2 \right).$$

*After setting $\eta_t = \eta \leq 1/2\beta$ and reparameterizing $\gamma_2 = k\beta$ for $k > 1$, we have*

$$\mathbb{E} \left[ \mathcal{L}_\mu(W^{\text{acc}}) - \mathcal{L}_\mu(w^*) \right] \leq \left( \frac{4ek\beta^2\eta}{n} + \frac{T}{n^2} \cdot 4ek\beta^2\eta + \frac{1/2 + \beta\eta}{T\eta} \right) \left( \frac{k+1}{k-1} \|w^*\|^2 \right).$$

*By minimizing over $T$, we can obtain the following:*

$$\mathbb{E}\left[\mathcal{L}_\mu(W^{\text{acc}}) - \mathcal{L}_\mu(w^*)\right] \le \left(\frac{4ek\beta^2\eta}{n} + \frac{2}{n} \cdot \sqrt{4ek\beta^2 \cdot (1/2 + \beta\eta)}\right)\left(\frac{k+1}{k-1}\|w^*\|^2\right).$$

*Now set $\eta = 1/2\beta$ to obtain the following:*

$$\mathbb{E}\left[\mathcal{L}_\mu(W^{\text{acc}}) - \mathcal{L}_\mu(w^*)\right] \le \inf_{k>1} \frac{2\beta}{n}\left(ek + 2\sqrt{ek}\right)\left(\frac{k+1}{k-1}\|w^*\|^2\right) = O(\beta\|w^*\|^2/n),$$

*which indicates an $O(1/n)$ sample complexity for smooth, convex and separable settings. This result recovers the Theorem 5 and the $O(1/n)$ rate in Lei & Ying (2020) up to constants.*

**Proof** This proof is adapted from Appendix C.2 of Lei & Ying (2020). The excess risk can be decomposed into (excess) optimization error and generalization error. The optimization error bound is directly borrowed from Lei & Ying (2020). The generalization error is bounded by combining the recovered stability bound Proposition 3 and the stability of SGD from Lei & Ying (2020).

Let $w^*$ be the weight that achieves the optimal population loss.

### D.6.1  EXCESS OPTIMIZATION ERROR

According to Lemma A.2(c) of Lei & Ying (2020), if the loss is non-negative, convex and $\beta$-smooth, and $\eta_t \le 1/2L$ and non-increasing, then for any constant $\bar{w}$ and constant $s$, one has

$$\sum_{\tau=0}^{t}\eta_{\tau+1}\mathbb{E}\left[\hat{\mathcal{L}}_s(W_\tau) - \hat{\mathcal{L}}_s(\bar{w}) \mid S = s\right] \le (1/2 + \beta\eta_1)\|\bar{w}\|^2 + 2\beta\sum_{\tau=0}^{t}\eta_{\tau+1}^2\hat{\mathcal{L}}_s(\bar{w}).$$

After setting $\bar{w}$ to $w^*$ and taking expectation over training sets, we have

$$\sum_{\tau=0}^{t}\eta_{\tau+1}\mathbb{E}\left[\hat{\mathcal{L}}_S(W_\tau) - \hat{\mathcal{L}}_S(w^*)\right] \le (1/2 + \beta\eta_1)\|w^*\|^2 + 2\beta\sum_{\tau=0}^{t}\eta_{\tau+1}^2\mathcal{L}_\mu(w^*). \tag{D.9}$$

Since the excess training error bound is only given after summing over steps, one has to sum the generalization error bound over steps as well.

### D.6.2  STABILITY AND GENERALIZATION ERROR

Theorem 3 of Lei & Ying (2020) states that if the loss is non-negative, convex and $\beta$-smooth, and SGD has step size $\eta_t \le 2/L$, the for any $p > 0$ one has

$$\epsilon_t^{\text{stability}} \le \frac{8(1+1/p)\beta}{n}\sum_{\tau=0}^{t-1}(1+p/n)^{t-1-\tau}\eta_{\tau+1}^2\mathbb{E}\left[\hat{\mathcal{L}}_S(W_\tau)\right],$$

where $\epsilon_t^{\text{stability}}$ is the $\ell_2$ on-average model stability at step $t$:

$$\epsilon_t^{\text{stability}} := \mathbb{E}_{Z_{-i}, V}\left[\frac{1}{n}\sum_{i=1}^{n}\mathbb{E}\left[\|W_t' - W_t\|^2 \mid Z_{-i}, V\right]\right]$$

Let $\gamma_2 > 0$ be a constant. Plugging this stability bound into Proposition 3 leads to

$$\text{gen}(\mu^n, P_{W_t|S})$$

$$\le \frac{1}{1-\beta/\gamma_2}\left(\frac{2\beta}{\gamma_2}\mathbb{E}\left[\hat{\mathcal{L}}_S(W_t)\right] + \frac{\beta+\gamma_2}{2}\epsilon_t^{\text{stability}}\right)$$

$$\le \frac{1}{1-\beta/\gamma_2}\left(\frac{2\beta}{\gamma_2}\mathbb{E}\left[\hat{\mathcal{L}}_S(W_t)\right] + \frac{\beta+\gamma_2}{2}\frac{8(1+1/p)\beta}{n}\sum_{\tau=0}^{t-1}(1+p/n)^{t-1-\tau}\eta_{\tau+1}^2\mathbb{E}\left[\hat{\mathcal{L}}_S(W_\tau)\right]\right)$$

$$\le \frac{1}{1-\beta/\gamma_2}\left(\frac{2\beta}{\gamma_2}\mathbb{E}\left[\hat{\mathcal{L}}_S(W_t)\right] + \frac{4(1+1/p)(\beta+\gamma_2)\beta(1+p/n)^{t-1}}{n}\sum_{\tau=0}^{t-1}\eta_{\tau+1}^2\mathbb{E}\left[\hat{\mathcal{L}}_S(W_\tau)\right]\right)$$

There are empirical losses on the trajectory at RHS of the above inequality. They can be bounded by Eq. (A.5) of Lei & Ying (2020), which states given any training set $s$ and any constant $\bar{w} \in \mathcal{W}$, one has

$$\sum_{\tau=0}^{t-1} \eta_{\tau+1}^2 \mathbb{E}\left[\hat{\mathcal{L}}_s(W_\tau) \mid S = s\right] \leq \eta_1 \|\bar{w}\|^2 + 2\sum_{\tau=0}^{t-1} \eta_{\tau+1}^2 \hat{\mathcal{L}}_s(\bar{w}).$$

Setting $\bar{w}$ to $w^*$ and taking expectation over training sets, we have

$$\sum_{\tau=0}^{t-1} \eta_{\tau+1}^2 \mathbb{E}\left[\hat{\mathcal{L}}_S(W_\tau)\right] \leq \eta_1 \|w^*\|^2 + 2\sum_{\tau=0}^{t-1} \eta_{\tau+1}^2 \mathcal{L}_\mu(w^*).$$

Plugging it back leads to

$$\text{gen}(\mu^n, P_{W_t|S}) \leq \frac{1}{1 - \beta/\gamma_2} \left(\frac{2\beta}{\gamma_2} \mathbb{E}\left[\hat{\mathcal{L}}_S(W_t)\right] + \frac{4(1 + 1/p)(\beta + \gamma_2)\beta(1 + p/n)^{t-1}}{n}\left(\eta_1 \|w^*\|^2 + 2\sum_{\tau=0}^{t-1} \eta_{\tau+1}^2 \mathcal{L}_\mu(w^*)\right)\right)$$

As in Lei & Ying (2020), one can choose $p = n/T$ to have $(1 + p/n)^{t-1} \leq (1 + p/n)^{T-1} = (1 + 1/T)^{T-1} < e$. As a result, we have

$$\text{gen}(\mu^n, P_{W_t|S}) \leq \frac{1}{1 - \beta/\gamma_2} \left(\frac{2\beta}{\gamma_2} \mathbb{E}\left[\hat{\mathcal{L}}_S(W_t)\right] + \frac{4(1 + T/n)(\beta + \gamma_2)\beta e}{n}\left(\eta_1 \|w^*\|^2 + 2\sum_{\tau=0}^{t-1} \eta_{\tau+1}^2 \mathcal{L}_\mu(w^*)\right)\right)$$

To align the generalization error bounds with the weighted summation form of the optimization error, we weight them by $\eta_{t+1}$ and sum over steps the above inequality:

$$\sum_{t=0}^{T} \eta_{t+1} \text{gen}(\mu^n, P_{W_t|S})$$

$$\leq \sum_{t=0}^{T} \frac{\eta_{t+1}}{1 - \beta/\gamma_2} \left(\frac{2\beta}{\gamma_2} \mathbb{E}\left[\hat{\mathcal{L}}_S(W_t)\right] + \frac{4(1 + T/n)(\beta + \gamma_2)\beta e}{n}\left(\eta_1 \|w^*\|^2 + 2\sum_{\tau=0}^{t-1} \eta_{\tau+1}^2 \mathcal{L}_\mu(w^*)\right)\right)$$

$$= \frac{1}{1 - \beta/\gamma_2} \left(\frac{2\beta}{\gamma_2} \sum_{t=0}^{T} \eta_{t+1} \mathbb{E}\left[\hat{\mathcal{L}}_S(W_t)\right] + \frac{4(1 + T/n)(\beta + \gamma_2)\beta e}{n} \sum_{t=0}^{T} \eta_{t+1}\left(\eta_1 \|w^*\|^2 + 2\sum_{\tau=0}^{t-1} \eta_{\tau+1}^2 \mathcal{L}_\mu(w^*)\right)\right)$$

To get rid of the empirical losses $\sum_{t=0}^{T} \eta_{t+1} \mathbb{E}\left[\hat{\mathcal{L}}_S(W_t)\right]$, we apply Eq. (D.9) again and obtain

$$\sum_{t=0}^{T} \eta_{t+1} \text{gen}(\mu^n, P_{W_t|S})$$

$$\leq \frac{1}{1 - \beta/\gamma_2} \left(\frac{2\beta}{\gamma_2}\left((1/2 + \beta\eta_1)\|w^*\|^2 + 2\beta \sum_{t=0}^{T} \eta_{t+1}^2 \mathcal{L}_\mu(w^*) + \sum_{t=0}^{T} \eta_{t+1} \mathcal{L}_\mu(w^*)\right)\right.$$

$$\left. + \frac{4(1 + T/n)(\beta + \gamma_2)\beta e}{n} \sum_{t=0}^{T} \eta_{t+1}\left(\eta_1 \|w^*\|^2 + 2\sum_{\tau=0}^{t-1} \eta_{\tau+1}^2 \mathcal{L}_\mu(w^*)\right)\right).$$

After obtaining the above generalization error bound, the bound for excess risks can be obtained by summing up it and the optimization error bound Eq. (D.9):

$$
\sum_{t=0}^{T} \eta_{t+1} \mathbb{E}\left[\mathcal{L}_\mu(W_t) - \mathcal{L}_\mu(w^*)\right]
$$

$$
\leq \frac{1}{1 - \beta/\gamma_2} \frac{2\beta}{\gamma_2} \left( (1/2 + \beta\eta_1) \|w^*\|^2 + 2\beta \sum_{t=0}^{T} \eta_{t+1}^2 \mathcal{L}_\mu(w^*) + \sum_{t=0}^{T} \eta_{t+1} \mathcal{L}_\mu(w^*) \right)
$$

$$
+ \frac{1}{1 - \beta/\gamma_2} \frac{4(1 + T/n)(\beta + \gamma_2)\beta e}{n} \sum_{t=0}^{T} \eta_{t+1} \left( \eta_1 \|w^*\|^2 + 2 \sum_{\tau=0}^{t-1} \eta_{\tau+1}^2 \mathcal{L}_\mu(w^*) \right)
$$

$$
+ (1/2 + \beta\eta_1) \|w^*\|^2 + 2\beta \sum_{t=0}^{T} \eta_{t+1}^2 \mathcal{L}_\mu(w^*)
$$

$$
\leq \frac{2\beta}{\gamma_2 - \beta} \sum_{t=0}^{T} \eta_{t+1} \mathcal{L}_\mu(w^*) + \frac{1 + T/n}{n} \frac{4e\gamma_2\beta(\beta + \gamma_2)}{\gamma_2 - \beta} \sum_{t=0}^{T} \eta_{t+1} \left( \eta_1 \|w^*\|^2 + 2 \sum_{\tau=0}^{t-1} \eta_{\tau+1}^2 \mathcal{L}_\mu(w^*) \right)
$$

$$
+ \frac{\gamma_2 + \beta}{\gamma_2 - \beta} \left( (1/2 + \beta\eta_1) \|w^*\|^2 + 2\beta \sum_{t=0}^{T} \eta_{t+1}^2 \mathcal{L}_\mu(w^*) \right)
$$

The stated inequality can be obtained by dividing $\sum_{t=0}^{T} \eta_{t+1}$ and applying the Jensen's inequality to the convex $\hat{\mathcal{L}}_S(\cdot)$. ∎

# E    DISCUSSION ON THE LIMITATION

The major limitation of our bound is that our bound still relies on population gradients and Hessians. This limitation harms the applicability of our bound to self-certified algorithms (Pérez-Ortiz et al., 2021).

However, the limitation is not unique to our bound, but is an inherent limitation of the auxiliary trajectory technique. The most essential step of this technique is to switch from the original trajectory to the auxiliary trajectory with better properties. However, one must relate the auxiliary trajectory back to the original trajectory by adding their differences into the bound. In this process, the loss differences are used to measure such differences, resulting in the population loss difference and population statistics. As a result, previous representative works all have explicit reliance on population statistics. See $S'$ in Propositions 1 and 2. This reliance can be alleviated through some assumptions like $\mathcal{L}_\mu(W_T) \leq \mathbb{E}_{\xi \sim \mathcal{N}(0, \sigma I)}\left[\mathcal{L}_\mu(W_T + \xi)\right]$ of Wang & Mao (2022). Nevertheless, one still must verify this assumptions on the population set to rigorously apply them, especially when the model is under-fitted or the generalization is bad so that the output weight is far from local minima of the population loss. To sum up, existing representative results based on auxiliary trajectory must rely on population statistics at least implicitly.

In terms of dependence to population/validation statistics, we also optimize the omniscient trajectory using validation statistics, which is a heavier dependence. This may forms unfair comparison with existing bounds. However, we have tried to make the comparison fair by allowing the existing bounds to rely on the validation statistics (see Appendix C.5). Even with full access to validation sets, the existing bounds cannot exploit them and are still much numerically looser than ours. Lastly, the results of the existing and our bounds can be seen as not only competitors, but also different trade-offs between the dependence on validation set and bound tightness. Our bound demonstrates how tight a bound can be if one allows heavy dependence on validation sets, while previous works show the looseness when one controls the access to validation sets. Future works can start from these two extremes to achieve a better trade-off or even break it.

