# OpenReview forum: "Leveraging Flatness to Improve Information-Theoretic Generalization Bounds for SGD"
_ICLR.cc/2025/Conference — ICLR 2025 Poster_

### Official Review · Reviewer_aqMJ · 2024-11-03

**Soundness:** 3
**Presentation:** 3
**Contribution:** 3
**Rating:** 8
**Confidence:** 3

**Summary:**

This work introduces a novel information theoretic bound. As is common for bounds of this type, the starting point is a well-known result that states that the generalization error can be bounded in term of the mutual information (MI) between the training set $S$ and the learning algorithm’s output $W$. For SGD algorithms, under Gaussian weight priors and Gaussian noise injection on the final iterate, this boils down to a generalization bound that consists of a flatness term and a trajectory term (see e.g. Neu et al. and Wang et al.).

Some existing MI bounds of this form have a crucial defect: they increase in the batch size even though small batch SGD generally converges to flatter minima due to larger gradient noise. This defect is purported to be caused by the hidden flatness-dependence of the trajectory term in those MI bounds, though this wrong tendency can be addressed with i.i.d. per-iterate auxiliary noise injections (see Neu et al.).

The main contribution of this paper is to improve upon existing MI bounds such as Neu et al. and Wang and Mao via a novel technique. More specifically, the authors of this work conduct a more fine grained analysis wherein, rather than choosing the optimal noise covariance based on the expected trajectory $W$, one chooses the best noise covariance for the specific instances of $W$, and then calculates the expected error bound over $W$. By doing so, one can a) sidestep an overfitting issue with estimating the best noise covariance based on (expensive/scarce samples of) $W$; and b) get a numerically tighter bound by virtue of exchanging the $\inf\mathbb E$ with the $\mathbb E \inf$ operator.

This approach, based on the construction of an "omniscient trajectory", is general enough so that it can be applied to other existing methods. Moreover, it is shown to be empirically tighter than existing, comparable bounds, and it is shown to be theoretically optimal (up to constants) for SGD on Convex-Lipschitz-Bounded (CLB) problems for a specific choice of training iterations and step size.

**Strengths:**

- The proposed approach is sound and very effective in achieving the goal of better modelling the effect of flatness on generalization. For example, it achieves a decrease of the trajectory term on the order of $10^6$ (multiplicative) in the case of CIFAR-10 over the one proposed by Neu et al.. This is a valuable contribution.
- The paper is overall well-written.
- Like the bound of Neu et al., the proposed bound has the right scaling w.r.t. the batch size, while also being numerically tighter.
- In fact, the bound is numerically tight in a very simple setting (MNIST + 2-layer MLP).
- The omniscient-trajectory technique is to a degree general purpose and can be applied to a whole class of similar bounds.

**Weaknesses:**

- While the bounds is quite tight for MNIST, it seems that it is numerically vacuous (i.e. larger than 1 for the 0-1 classification accuracy) already for CIFAR10. So while it is numerically tighter than comparable bounds, it does not provide a major qualitative improvement over bounds such as the one by Neu et al..
- In my view, even if one lets go of requiring tightness of a generalization bound, the next most important goal is to have the right tendency/correlation w.r.t. to the parameters of the learning setup. However, it is my impression that this bound does not “fix” any of such qualitative defects of existing comparable bounds, whereas, for example Neu et al. "fixes" the scaling w.r.t. batch-size and other flatness bounds improve upon prior ones by enforcing invariance w.r.t. rescaling the parameter vector.
- Empirically, applying Theorem 2 is difficult and requires multiple assumptions/heuristics, e.g.:
    - Crucially, multiple population means have to be estimated from i.i.d. samples of $W$, which is difficult. Given a training set of fixed size $N$, one needs to partition it by averaging of $k$ subsets of size $N/k$, yielding a bound for SGD on training sets of size $N/k$. If one heuristically were to apply that obtained bound to the output of SGD over the full training set $S$, it must be quite loose unless $S$ is highly redundant, which is typically not the case.
    - The bound has a hyperparameter $\lambda$ that regulates a trade-off between different terms in the bound and at the same time must upper bound the spectral norm of the Hessians
    - The bound assumes exactly zero training error at the final output. Thus it does not apply in the case of early stopping, and the application to approximately zero training error outputs requires a small leap of faith.
    - In the case of unbounded losses, one must clip the loss function in order to guarantee the sub-Gaussian concentration required for the analysis.
- The result gives a bound on the expected generalization error. This is qualitatively weaker than a high probability upper bound. Arguably, this is a rather minor weakness since deep learning models tend to be exceptionally stable w.r.t. different initializations and draws of the SGD batches.
- These kinds of bounds apply to the generalization error in terms of the expected surrogate loss, rather than the $0-1$ which is the real measure of interest in classification. I acknowledge that this is essentially unavoidable here in order to link generalization to the optimization trajectories, which are obtained from minimization of the surrogate loss. However, it should be noted that this makes the analysis inherently more loose since the surrogate risk is always an upper bound on the $0-1$ risk (i.e. the test accuracy).

**Questions:**

- There exist PAC-Bayes based generalization bounds that are non-vacuous even on ImageNet, while yours seems to be vacuous already for CIFAR10. How do these bounds compare conceptually to your type of bound? Perhaps they have in some sense an “unfair advantage”?
- What is the general tendency of your bound to changes in depth, width, training set size, and scaling of the full weight vector? If the tendency is in the wrong direction, is this shared by other bounds based on MI? I am asking because the bound being numerically tighter does not necessarily rule out the fact that it now scales in the wrong directions for some parameters.
- I suggest that you already introduce some of the notation around the equation in line 82. For example, it might not be clear to some readers unfamiliar with these results that $g_t$ denotes the gradients.
- Could you please expand on the difficulties of evaluating the bound in practise, specifically on which estimation steps you consider the most loose ones? How does the hardness/accuracy of estimation compare to that of the bounds of Neu et al. and Wang and Mao, for example in terms of how the training set has to be partitioned to carry out all the required estimations in a sound manner, and the trade-offs induced by the specific choices of such partitions?
- I would suggest that you include some of the most important aspects of empirical evaluation in the main part of the paper, since this might be considered more important to some readers than some of the more technical/theoretical details.
- Related to the above question: if there are indeed particularly loose estimation steps, do you perhaps have some high level ideas on how to address these inadequacies in future work?
- I see in Appendix C.1 that for your experiments, you evaluate the penalty terms over the test set, which should make the bound incomparable to some of the other competing generalization bounds which work only on the training set. If you disagree with this statement, could you please elaborate? In my view, a fair comparison would require that you set aside a part of the training set for such estimation procedures.

============================================================================

Typos/minor comments:
- Line 34: perhaps "that" instead of "because"
- Line 932 "Hessians" -> "Hessian"

=============================================================================

References:

- Gergely Neu, Gintare Karolina Dziugaite, Mahdi Haghifam, and Daniel M. Roy. Information- theoretic generalization bounds for Stochastic Gradient Descent. In Proceedings of Thirty Fourth Conference on Learning Theory, pp. 3526–3545. PMLR, July 2021. ISSN: 2640-3498.
- Ziqiao Wang and Yongyi Mao. On the generalization of models trained with SGD: Information- theoretic bounds and implications. In Advances in Neural Information Processing Systems, October 2021.

---

> ### Author Response · Authors · 2024-11-21
>
> We thank the reviewer for their feedback and will respond to the raised questions below.
> We have revised the manuscript after these feedbacks. Modifications corresponding to your feedbacks are colored in **purple** or **red**.
>
>
>
> ### **W4.1: The bound is vacuous on CIFAR10, indicating the improvement is not qualitative.**
>
> Thank you for thorough examination and comparison of all experiments.
>
> It is possible that the reviewer assumes the bound is ultimately for 0-1 losses and the Cross Entropy loss only serves as an intermediate upper-bound for the 0-1 loss. In this case, one would conclude that the bound is vacuous for 0-1 loss in the experiments. However, we want to clarify that our bound is intended for Cross Entropy instead of 0-1 loss in the experiments. This is because 0-1 loss is not differentiable and has no Hessian, which is incompatible with our motivation of "leveraging flatness". More detailed discussion can be found in the response to your W4.5. After the clarification, one can conclude that our bound is not vacuous for a Cross Entropy bound:
>
> - **Non-vacuousness**: Actually, we use **Cross Entropy loss instead of 0-1 loss** across **all experiments**, including the CIFAR10 one. Therefore, our bounds are not numerically vacuous by being larger than the trivial upper-bound of the loss.
>
> - **Tightness**: In every result, we mark the true generalization error by dashed lines for comparison. From Figure 3(a), one can see that our bound sticks tightly to the true generalization error. In Figure 4, we even struggled to separate our bound with the true generalization error. Therefore, our bound is numerically tight.
>
> Now in our second draft, we have emphasized the use of Cross Entropy losses at the start of **Sec 4 (Line 419)**.
>
> ### **W4.2: The new bound does not correct more wrong tendencies.**
>
> Thank you very much for carefully reviewing the additional results. However, we consider Neu et al.'s "fix" is only partial and our fix is more adequate:
> - Prop. 8 of Neu et al. indeed "fixes" the scaling w.r.t. batch size to some extent. However, this "fix" is only partial. From Figure C.1(b), one can see that the trajectory term is indeed not increasing as batch size decreases, but becomes almost unrelated to batch size. Therefore, in Neu et al.'s Prop 8, the improved flatness due to smaller batch size is still inadequately exploited by its trajectory term.
> - In contrast, in Figure 3(b) for our bound, the trajectory term is decreasing as batch size decreases. This indicates that we "fix" the scaling w.r.t. batch size to a much larger extent and the improved flatness due to smaller batch size is much more adequately exploited by our bound.
>
> We have added the discussion in **Lines 1371-1373** and **Line 1376**.
>
> ### **W4.3: Empirically, applying Theorem 2 is difficult and requires multiple assumptions/heuristics.**
>
> #### **W4.3.1: The bound requires splitting the dataset.**
>
> Thank you very much for carefully examining our assumptions.
>
> Regarding the requirement of splitting training set, representative existing MI-based works, like that of Neu et al. (2021) and that of Wang and Mao (2021), all share this assumptions.
>
> This drawback can be traced back to MI's definition $I(W; S) := \mathbb{E}\_S[\operatorname{KL}(P\_{W \mid S} \parallel \mathbb{E}\_{S'}[P\_{W\mid S'}])]$, which has hidden expectations over $S$ and $S'$. Even if one turns to its upperbound, arguably, almost any reasonably *tight* and *data-dependent* upperbound of MI inherits these expectations as long as one insists on the optimal prior $\mathbb{E}\_{S'}[P\_{W\mid S'}]$. If one gives up the optimal prior, the data-dependent prior by Negrea et al.(2019) (that depends on random subsets of the training set) can be used to avoid the expectation over $S'$, and then avoid the outer expectation over $S$ at the cost of estimation's variance. In this case, our technique can also be combined with Negrea et al.(2019)'s result in Corollary D.3.

---

> > ### Author Response · Authors · 2024-11-21
> >
> > ### **Q4.1: How do non-vacuous PAC-Bayesian bounds on ImageNet compare conceptually to your type of bound?**
> >
> >
> > Thank you very much for the suggestion of the comparison with PAC-Bayesian bounds, which is closely connected to MI bounds as mentioned in the introduction. But firstly, we must emphasize that our bound is not vacuous on CIFAR10, as mentioned in the response to W4.1.
> >
> > Now we compare the PAC-Bayesian bounds on ImageNet. These large-scale bounds often augment PAC-Bayesian bounds by **compression**. They first design a weight compression scheme (including encoding and/or noise injection) that does not change the output of the network too much, and select or optimize a prior in the compressed-weight space. Then they compress the weight and apply the PAC-Bayesian bounds in the compressed-weight space. The comparisons are summarized in the following table and are elaborated in the bullet points following:
> >
> >   |                             | Weight-Dependence-Only Compression-Augmented PAC-Bayesian | Rate-Distortion             | Learned-Compression-Augmented PAC-Bayesian        | Omniscient                  |
> >   |:---------------------------:|:---------------------------------------------------------:|:---------------------------:|:-------------------------------------------------:|:---------------------------:|
> >   | Sampling                    | random network                                            | training                    | random network                                    | training                    |
> >   | Algorithm                   | ***modified algo.***                                      | given algo. like SGD        | ***modified algo.***                              | given algo. like SGD        |
> >   | Prior/Info. Measure         | manully designed/KL                                       | **optimal/MI**              | manually designed/KL                              | **optimal/MI**              |
> >   | Dependence of Compression   | independent to  training set                              | independent to training set | **exploiting training set**                       | **exploiting training set** |
> >   | Optimization of Compression | manually designed                                         | **optimized**               | partially manually designed and partially learned | **optimized**               |
> >
> >   **Bold** texts are indicate the better choice in terms of tightness while *italic* texts indicate the potential "unfair advantage".

---

> ### Author Response · Authors · 2024-11-21
>
> #### **W4.3.2: The bound introduces a hyperparameter $\lambda$.**
>
> We believe the impact of $\lambda$ is relative small because it can be easily selected. Furthermore, it can be removed.
>
> - **The selection of $\lambda$** is direct: The emphasis on trajectory terms, the assumption when approximating the bound and the experiments all consistently recommend that $\lambda$ should be set very large. The bound is also insensitive to $\lambda$ in terms of looseness when $\lambda$ is large enough. In Figure 4, changing $\lambda$ from $10^9$ to $10^3$ does not affect the tightness very much.
>   - Regarding the assumption that $\lambda$ "must upper bound the spectral norm of the Hessians", we want to clarify that this assumption is only used in Sec 3.4 to facilitate the approximation. It is not used in Theorem 1,2 or the experiments. We did not explicitly remark this since we thought it is not included in Theorem 1 or 2. We have added such remarks in Sec 3.4 (Line 383, Lines 386-388) in the second draft.
>     - We have added such remarks in **Sec 3.4 (Line 383, Lines 386-388)** in the second draft.
> - **Removal of $\lambda$**: The hyperparameter $\lambda$ can be removed by at least three methods:
>   - The hyperparameter $\lambda$ comes from the surrogate optimization target for optimizing the omniscient trajectory. When the bound is only for numerical use, we can get rid of $\lambda$ with black-box optimizers by directly optimizing the omniscient trajectory. However, this sacrifices the interpretability.
>   - We can also develop bounds that optimize $\lambda$ in it, i.e., putting $\lambda$ below $\inf$:
>     - It is potentially possible to modify the bound to allow grid search of $\lambda$ without a negative bias/overfitting. This is already done by Dziugaite et al. (2017) (see "Additional References") to grid-search a hyperparameter in their numerical PAC-Bayesian bound.
>     - In the spirit of exchanging expectation and optimization, we can optimize $\lambda = \lambda(S, V, W\_{0:T}, g\_{1:T})$ in the instance level. This can be done by trying multiple $\lambda$, splitting more validation sets and evaluating the omniscient trajectory with different $\lambda$ on this validation sets. This comes at the cost of a more complicated proofs, bounds and estimation with more data splitting.
>
> #### **W4.3.3: The bound assumes zero training error at the final output.**
>
> This assumption is only used in Sec 3.4 to simplify the calculation of bound approximation. Main results (Theorem 1 and 2) and experiments do not rely on this assumption. We thought it was enough to not include the assumption in Theorem 1 and 2. We have added explicit remarks on the assumptions in **Sec 3.4 (Line 383, Lines 386-388)** of the second draft.
>
> #### **W4.3.4: The loss must be bounded to ensure sub-Gaussianity**
>
> We agree with you that the clipped loss is undesirable.
> Nevertheless, the impact of clipping will be small if the clipping threshold is set to be a large value and the losses of most (training and testing) samples are low so that they are below the threshold. Both of these conditions are actually practical for the following two reasons:
>
> - Firstly, our bound is not sensitive to the clip threshold and thus the threshold can be set very large: In Theorem 2, the clipping threshold is reflected by $R=[\text{threshold}]/2$, which is under a cubic root in the non-penalty terms. Additionally, experiments reveal that the bound is dominated by the penalty term and the sum of non-penalty terms contribute little. Combined together, increasing the threshold and $R$ affects an almost invisible portion of our bound under the cubic root. As a result, the entire numerical bound is very  insensitive to the threshold and $R$, which can be set very large. Particularly, we set it a large value of $12 \ln [\text{number of classes}]$, which is 12 times the expected cross entropy right at initialization.
> - Secondly, most sample-wise losses are below the threshold after training: As pointed out in Sec 4, we used the original unclipped loss for training, resulting in well-learned samples whose losses are far below the large threshold. Only after the training stage, we estimate the bound with the clipped loss.
>
> In a summary, with a large value of threshold and small losses of well-learned samples, the clip is actually only employed to guarantee the sub-Gaussianity while has no impact on the estimation.
>
> Regarding to the removal of the sub-Gaussianity, it is a separate and often discussed topic in MI and PAC-Bayesian community. We believe it should be addressed in separate works and is out of our scope.

---

> ### Author Response · Authors · 2024-11-21
>
> ### **W4.4: The result gives a bound on the expected generalization error, which is qualitatively weaker than a high probability upper bound.**
>
> Thank you for pointing out the problem on the form of the bound.
>
> We selected that expected bounds following existing works.
> It is conventional for MI-based community to present their main results in expected bounds (Xu and Raginsky, 2017; Asadi et al., 2018; Bu et al., 2020; Steinke and Zakynthinou, 2020; Harutyunyan et al., 2021; Hellström & Durisi, 2022; Wang and Mao 2021;2023a;2023b;2024; Pensia et al., 2018; Negrea et al., 2019; Wang et al., 2021b, Neu et al., 2021). The expected bounds are much simpler. By relieving the details (like some $\log \frac{1}{\delta}$ terms and steps leading to these terms) brought by the high-probability bounds, researchers can better demonstrate their methods by expected bounds.
>
> ### **W4.5: The bound applies to generalization error in terms of the expected surrogate loss rather than 0-1 loss.**
>
> Thank you for pointing out the looseness of the surrogate loss.
>
> We use the surrogate loss following previous works. It is conventional to use surrogate losses in MI-based community and flatness-related works, because 1) as the reviewer pointed out, the model is trained under surrogate loss and one must refer to it to link to the training dynamics; and moreover, 2) 0-1 loss is not differentiable and has no Hessian. Since our work "leverages **flatness**", we must use Hessians that formulate the flatness and thus exclude 0-1 loss. Of course, we can try to approximately (by the surrogate loss) optimize against 0-1 loss. However, this extra procedure introduces noises when comparing the bounds' numerical tightness.
>
> Although the surrogate loss bound is an upperbound for 0-1 loss bound, the true generalization surrogate loss is also an upperbound for the true generalization 0-1 loss. Therefore, comparing the surrogate bound with the true losses can also reflect how well the bound captures generalization. Moreover, in some fields, e.g. natural language generation, the ultimate metric, e.g. the perplexity, mainly comprises of the surrogate cross entropy loss.

---

> ### Author Response · Authors · 2024-11-21
>
> - There is a subtle difference in how PAC-Bayesian and MI bounds are used. PAC-Bayesian bounds are usually used for randomized deep networks and the "sampling" for PAC-Bayesian bounds are usually not the training but perturbing the trained weight. In contrast, the "sampling" stage in MI bounds correspond to the training. We have to align this difference to make further comparisons. We only use an abstract "sampling" stage to refer to the random process assumed in both types of bounds.
>
> - After the above alignment, we first relate compression-augmented PAC-Bayesian bounds to existing MI bounds and rate-distortion their variants.
>
>   - Firstly, consider the PAC-Bayesian bounds without compression: As discussed by Alquier (2023), MI bounds can be obtained by some expected-bound (without-compression) version of PAC-Bayesian bounds applied to the optimal prior, i.e., $I(W; S) := \mathbb{E}[\operatorname{KL}(P\_{W\mid S}, P\_{W})] \le \mathbb{E}[\operatorname{KL}(P\_{W\mid S}, Q\_{W})]$. As a result, PAC-Bayesian bounds are in fact conceptually looser than MI bounds.
>
>   - Secondly, insert the compression: The compression is to map the weights with two considerations: 1) decreasing the KL divergence terms, and 2) imitating the output of original weight.
>
>     - To see its correspondence to existing variant MI bounds, we first replace the KL divergence with MI as discussed above, and the first consideration becomes minimizing the MI.
>     - Regarding the second consideration, there is a major difference with MI bounds: Many PAC-Bayesian bounds are not fully devoted to bounding the generalization of a **given** algorithm (e.g., SGD or SGLD), but are allowed to more or less modify the algorithm to better cooperate with the the bounds (i.e., self-certified algorithms). Therefore, the goal of imitating the output is not to relate back to the given algorithm (as it is for auxiliary trajectories) but to retain the low empirical risk and generalization of the uncompressed model. These works often care about and report the certified generalization without the loss difference with the uncompressed model. To facilitate comparison, we must align this difference by conceptually forcing the PAC-Bayesian bounds to focus on given algorithms. Therefore, we make the second consideration explicit by putting constraints on training and population loss differences.
>
>     Putting them together, if the compression scheme only depends on the model weight, then we arrive at the rate-distortion bounds, which essentially looks like $\inf\_{P\_{\tilde{W}\mid W}} I(\tilde{W};S) + \mathbb{E}[\hat{\mathcal{L}}\_S(\tilde{W}) - \hat{\mathcal{L}}\_S(W)] - \mathbb{E}[\mathcal{L}\_\mu(\tilde{W}) - \mathcal{L}\_\mu(W)]$.  Here, the optimized conditional distribution $P\_{\tilde{W} \mid W}$ corresponds to the compression scheme that only depends on the weight, while $I(\tilde{W}; S)$ corresponds to the KL divergence between the optimal compressed weight and the optimal prior, and the remaining terms correspond to the constraints of imitating outputs. The optimization corresponds to the process that researchers design the compression scheme. As a result, weight-dependence-only compression-augmented PAC-Bayesian bounds conceptually align to (and are looser than) rate-distortion bounds in the MI line.
>
> - As discussed in Sec 3.4, the optimized auxiliary trajectory $\tilde{W}$ in the rate-distortion is not allowed to rely on training set $S$ or other random variables. In contrast, our omniscient trajectory can rely on all of the random variables, therefore optimizing better. Therefore, our bound is in fact conceptually tighter than the weight-dependence-only compression-augmented PAC-Bayesian bounds. However, many compression-augmented PAC-Bayesian bounds use learned compression schemes, which depend on the training set as our omniscient trajectory. Therefore, these bounds can be regarded as instances of the fully optimized omniscient MI bounds (our Theorem 1). After the above alignments, they are conceptually looser due to the KL divergence and the compression scheme that is not fully optimized.
>
> To sum up, there is indeed an "unfair advantage" that PAC-Bayesian bounds are allowed to switch to a different and more cooperative training algorithm. If we forbid this, our omniscient bounds conceptually recover and are tighter than those large-scale PAC-Bayesian bounds. However, the advantage is not the main reason behind for their being "non-vacuous even on ImageNet, while yours seems to be vacuous already for CIFAR10". The main reason is that our bound is for Cross Entropy loss instead of 0-1 loss in experiments and they are non-vacuous for CIFAR10 compared to the true Cross Entropy generalization error.

---

> ### Author Response · Authors · 2024-11-21
>
> ### **Q4.2: What is the general tendency of your bound to changes in depth, width, training set size, and scaling of the full weight vector?**
>
> We will add the corresponding experiments as you advised. However, it will come in later revisions because we are rerunning a lot of experiments to answer your Q4.7. These reruns took a lot of time.
>
> ### **Q4.3: It is better to introduce some notation in the introduction.**
>
> Thank you very much for your advices on improving readability. We have added short explanations for these notations in the **Introduction (Line 075, Line 087)**.
>
> ### **Q4.4: Could you please expand on the difficulties of evaluating the bound in practise?**
>
> Thank you very much for your advice. We have found the following loosenesses:
> - **Major looseness**: We consider the estimation of the trajectory term of our bound is the most loose one. This is because this expectation is the only one estimated with positive bias, while other expectations are estimated unbiasedly.
> - **Minor looseness**: Another estimation looseness may arises from optimizing the omniscient trajectory. The optimization relies on the expected population gradient and Hessian. Their estimates on the validation set are definitely different from the real expectation due to random fluctuations, and the optimization will be suboptimal, inducing looseness. However, we believe this looseness has minor impact:
>   - **Minor impact of fluctuations in the estimated Hessians**: After plugging the optimized omniscient trajectory back to the bound, population Hessians always show up in $(I + H / 2\lambda C)^{-1}$, where $\lambda$ very large and an inversion is involved. As a result, the impact of the fluctuations in the estimates will be suppressed by being divided by $2\lambda C$, being shadowed by the identity $I$, and the inversion.
>   - **Minor impact of fluctuations in the estimated gradient**: After plugging the optimized omniscient trajectory back to the bound, the population gradient is directly multiplied with other validation-set-independent matrices and the fluctuations will not be amplified by the optimization. Its looseness reduces to the looseness of the trajectory term.
>
> Regarding the difficulty of estimation, our bound is more difficult to estimate than existing bounds by requiring more validation sets.
> - The existing works have two selections:
>   - Using reasonable assumptions on the population losses to avoid validation sets:
>     - Dataset splits: $k$ training sets + $0$ validation set.
>       - If one wants to verify the assumption, then $1$ validation set.
>     - Numerical Tightness: compromised. Using the assumption gives up the benefit from population loss changes on reducing the flatness term.
>   - Using validation sets:
>     - Dataset splits: $k$ training sets + $1$ validation set
>     - Numerical Tightness: Not compromised but loose as the bounds inherently are.
> - Our bounds also have two typical options:
>   - Ignoring population Hessian and gradient when optimizing omniscient trajectory (setting $\tilde{H}\_{\text{pen}} = \hat{H}\_{S}(W\_T), J = \nabla \hat{\mathcal{L}}\_{S}(W\_T)$)
>     - Dataset splits: $k$ training sets, $1$ validation set (for estimating the penalty term and the flatness term)
>     - Numerical Tightness: compromised.
>   - Using the population Hessian and gradient in optimization (setting $\tilde{H}\_{\text{pen}} = \hat{H}\_{S}(W\_T) - H\_{\mu}(W\_T), J = \nabla \hat{\mathcal{L}}\_{S}(W\_T) - \nabla \mathcal{L}\_\mu(W\_T)$):
>     - Dataset splits: $k$ training sets, $2$ validation set (1 for optimizing the omniscient trajectory, 1 for evaluating the penalty term and the flatness term)
>     - Numerical Tightness: very tight.
> - Driven by your question, we came up with a more complicated estimation scheme for the second option for our bound. The second validation set is used to evaluate the penalty term, i.e., evaluate the "population loss" of the omniscient trajectory optimization. That is, it acts as the "testing set" for the optimization. Since in Theorem 2, the (surrogate) optimization is a convex optimization with explicit solution, maybe one can use a generalization bound to replace the "testing set". However, this double-layered generalization bound may be too complicated. If you would like to see this result, we are very happy to implement it. However, it requires a lot of time to rerun all experiments so it must wait for the next draft. Moreover, it also takes some time to develop this meta-generalization bound, because the optimization is done on a surrogate second-orderly approximated loss while tested on the original highly non-convex loss.

---

> ### Author Response · Authors · 2024-11-21
>
> ### **Q4.5: It is better to include some of the most important aspects of empirical evaluation in the main part of the paper.**
>
> Thank you very much for advices on improving the credibility of experimental results. We have added how the bounds are estimated and how we split the dataset for estimations in Sec 4 (Lines 426-430).
>
> ### **Q4.6: Do you have some high-level ideas to address the loss estimation steps?**
>
> The most loose step, as mentioned above, is the positively estimated trajectory term.
>
> We can expanding the squared norm of vector differences in it and conduct unbiased estimation term by term. Specifically, the trajectory term looks like $\mathbb{E}[\\|X - E \mathbb{E}[Y]\\|^2]$, where $X$ is correlated with $E$ and $E$ contains inverting Hessians. Expanding it leads to $\mathbb{E}[\\|X\\|^2] - 2 \mathbb{E}[ \mathbb{E}[Y]^\top E^\top X]+ \mathbb{E}[\\|E \times \mathbb{E}[Y]\\|^2] = \mathbb{E}[\\|X\\|^2] - 2 \mathbb{E}[Y]^\top \mathbb{E}[E^\top X]+ \mathbb{E}[\\|E \times \mathbb{E}[Y]\\|^2]$, all of whose terms can be estimated unbiased.
>
> However, this estimator increases the number of inverse-Hessian-vector products (iHVP): The positively biased version requires only $2k$ iHVPs, while the unbiased version requires **extra** $2k$ iHVPs. Since iHVPs cost a lot of computation, we decide to still use the positively biased estimator. After all, the numerical results produced by it is already very tight. A more detailed discussion has been added **at the end of Appendix C.2**.
>
> ### **Q4.7: The bound is evaluated on test set. The use of test/validation set make the comparison unfair for existing bounds using only training sets.**
>
> We are sorry that we applied the test set that evaluates the true error to the evaluation of the penalty term. We thought that no optimization and correlation are involved and the true error and the penalty term will still be unbiasedly estimated. We have corrected the code and experimental details to use another validation set for bound evaluation and rerun the experiments. The experiments are still running and we expect they show similar tightness, due to the unbiasedness of old experiments.
>
> Since we use more validation sets, we will discuss the comparison fairness of our bound to other methods:
> - First of all, we are trying to make the comparisons more fair. Unfortunately, we have not seen a good solution so far for our bounds to avoid the dependence on validation sets. Therefore, we instead allow existing bounds to also depend on validation sets and use them to improve their numerical tightness (as detailed discussed in the next bullet point). We are rerunning experiments and the results will come soon.
> - Secondly, we believe the current comparison is not fully unfair, because existing bounds also make assumptions on validation sets. Methods using auxiliary trajectories (including [Wang and Mao, 2021], [Neu et al., 2021] and ours) all rely on population statistics to measure the difference between the auxiliary trajectory and the original trajectory, as discussed in . Otherwise, the bounds on the auxiliary trajectory is totally unrelated to the original trajectory. The dependence of previous results is **explicit** through $S'$ in Prop. 1 and 2 in our Sec 2.3. Although these bounds can be made validation-avoidable by assuming $L\_\mu(w\_T) \le \mathbb{E}\_{\xi}[L\_{\mu}(w\_T + \xi)]$ on the validation set, we argue that **both our method and previous methods all depend on the validation set**.  Therefore, the comparison is not fully unfair.
> - Lastly, the results on existing and our bounds can be seen as not only competitors, but also different trade-offs between the dependence on validation set and bound tightness. Our bound shows how tight a bound can be if one allows heavy dependence on validation sets, while previous works show the looseness when one controls the access to validation sets.  Future works can start from these two extremes to achieve better trade-off or even break this trade-off.
>
> We have added these discussion in Sec 6 (briefly) and in Appendix E (in detail).
>
> ## **Minor Comments**
>
> Thank you very much for such careful reading. We have corrected these typos in the second draft (**Line 034, Line 1040**).
>
> ## **Additional References**
>
> - Dziugaite et al. "Computing nonvacuous generalization bounds for deep (stochastic) neural networks with many more parameters than training data." In Proceedings of the 33rd Annual Conference on Uncertainty in Artificial Intelligence (UAI), 2017.
> - Jeffrey Negrea et al. "Information-theoretic generalization bounds for SGLD via data-dependent estimates." In Advances in Neural Information Processing Systems, 2019.
> - Pierre Alquier. User-friendly introduction to PAC-Bayes bounds, March 2023. URL http://arxiv.org/abs/2110.11216.

---

> > ### Comment · Reviewer_aqMJ · 2024-11-25
> >
> > Thank you for providing such a comprehensive and detailed response. I reviewed your answers several days ago and have since been awaiting updates regarding your experimental results. However, as the discussion period is nearing its conclusion, I think it is best to share my feedback now.
> >
> > First, I would like to acknowledge that your responses have addressed the majority of my questions satisfactorily—thank you once again for your thoughtful and meticulous explanations. Based on this, I have revised my evaluation and raised my score to 6. In my current assessment, this is a solid theoretical contribution that is both clearly articulated and well-achieved, building meaningfully on prior works such as those by Neu et al. and Wang et al. [1,2], I am inclined to recommend acceptance of the paper on this basis.
> >
> > What prevents me from selecting a more enthusiastic positive evaluation is the fact that there are still some doubts I have regarding applicability, ultimately somewhat limiting the impact of this work. While I understand that this is a theory paper--and I do not expect exhaustive experimental verification--I believe that a more definitive ranking of the bounds’ performance can only be achieved by empirical comparison under a fixed budget of validation samples. Such analysis is essential for assessing whether these bounds can inform why a particular algorithm outperforms another or guide the design of superior algorithms. Based on your own acknowledgments, it seems that evaluating your bound in practical scenarios poses some challenges not shared by other competing bounds.
> >
> > I would greatly appreciate it if you could share updates once your experiments—especially those related to Q4.7—are completed. At this stage, preliminary results (e.g., using a low number of random seeds or a limited range of values) would already suffice. Refinements, such as smoother curves, could then be incorporated into later revisions.
> >
> > Regarding my inquiries about scaling experiments (e.g., with depth or width), I consider these supplementary rather than essential. My primary motivation for raising this point was to explore whether, perhaps counterintuitively, your bound exhibits more accurate tendencies for certain parameters compared to other MI bounds, despite not being explicitly designed for this purpose. In the worst case, your bound might exhibit the same incorrect tendencies as other MI bounds, as highlighted in empirical studies such as [3] and [4]. This outcome would not diminish the value of your theoretical contribution. However, in the best case, your bound could demonstrate even greater practical relevance (even if such desirable empirical properties can not be theoretically explained at present).
> >
> >
> >
> > - [1]: Gergely Neu, Gintare Karolina Dziugaite, Mahdi Haghifam, and Daniel M. Roy. Information- theoretic generalization bounds for Stochastic Gradient Descent. In Proceedings of Thirty Fourth Conference on Learning Theory, pp. 3526–3545. PMLR, July 2021. ISSN: 2640-3498.
> > - [2]: Ziqiao Wang and Yongyi Mao. On the generalization of models trained with SGD: Information- theoretic bounds and implications. In Advances in Neural Information Processing Systems, October 2021.
> > - [3]: Jiang, Y., Neyshabur, B., Mobahi, H., Krishnan, D., & Bengio, S. (2019). Fantastic generalization measures and where to find them. arXiv preprint arXiv:1912.02178.
> > - [4]: Dziugaite, G.K., Drouin, A., Neal, B., Rajkumar, N., Caballero, E., Wang, L., Mitliagkas, I. and Roy, D.M., 2020. In search of robust measures of generalization. Advances in Neural Information Processing Systems, 33, pp.11723-11733.

---

> > > ### Author Response · Authors · 2024-11-25
> > >
> > > We thank the reviewer for carefully evaluating the fairness of our experimental comparison.
> > >
> > > ## **More experiments on Q4.7**
> > >
> > > To address your concern on the fairness, we have uploaded the third draft and added more experiments where the existing bounds are allowed to access the validation set in **Line 429** and **Appendix C.5**, as discussed in our response to your Q4.7.
> > >
> > > Briefly, the with-validation existing bounds are still much looser than ours: We have allowed the existing bounds to subtract population Hessians from the flatness term, which presumably tightens the bound. However, it turns out that the population Hessians are magnitudes larger than the empirical bound, which can make the bound negative. We suspect it is the conventional second-order approximation (also done by Wang and Mao, 2022) that makes the bound estimation incorrect. To alleviate this, we take absolute values to the flatness term as in the main results of Wang and Mao (2022) and Neu et al. (2021). In this form, the existing bounds in fact become looser, because the population Hessian now dominates the (absolute) flatness term and is magnitudes larger than the empirical Hessian. As a result, the existing bounds cannot fully exploit the validation sets even if we feed the validations to them. The comparison with the existing bounds is now fair, where our bound is still magnitudes tighter than existing bounds.
> > >
> > > ## **More experiments on varying other hyperparameters**
> > >
> > > A part of experiments varying width, depth and training data usage on CIFAR10 has been added **in Figure 4 in Sec 4 and in Figure C.8 in Appendix C.5**. Experimental results under varied weight scaling (at the end of training) on MNIST with MLP are also added to **Figure C.8** (ResNets are more complicated models and it is more difficult to scale weight without changing the prediction drastically).
> > >
> > > Among all tested hyperparameters, with suitably selected $\lambda$, our bound can well capture the tendency of generalization. Therefore, we consider our bound "robust" to the selection of hyperparameters, at least for hyperparameters commonly tuned.

---

> ### Comment · Reviewer_aqMJ · 2024-11-25
>
> Thank you for making these additions to the paper. I have a few more questions about those:
>
> - Just to verify, in Table 1, in each column, you average over all the instances of the parameter in question (over multiple random seeds) and display the average noise variance / 10^9? E.g. you calculate the average of the noise for batch sizes 100, ..., 500? Right now the formatting is at little confusing, it looks like you divide the label noise etc. by 10^9. This should be reformatted for the camera ready version.
> - There are also quite a lot of typos in some of the revised/new sections, so please make sure to fix those for the camera-ready version.
> - In figure C.8 (f), the cross-entropy increasing with the rescaling of weights. I apologize for not being more precise earlier in the discussion, but what I had in mind is the rescaling of a homogenous network (e.g. 1 hidden layer ReLU MLP with no bias in the output layer), where a scaling of the weights does not affect the predictions at all, and only *decreases* the cross-entropy, assuming most points are classified correctly. Since in your figure, the cross-entropy increases, you perhaps you used an MLP with output bias? Most flatness-based generalization bounds increase when rescaling homogenous models, which is why I wanted to see if the same happens for your bound. If you cannot rerun the experiment for the homogenous case before the deadline (which is now very close), this will not negatively affect my evaluation, but it would be nice to have in the camera-ready version.
> - Could you please provide the typical/rough range of test accuracy values across the various experiments, including both the older ones and the newly added ones?
> - Just to verify: the bounds by Neu et al. and Wang et al. also require splitting the training set for their empirical evaluation, and you do the splitting in the same manner?

---

> ### Author Response · Authors · 2024-11-27
>
> Thank you very much for carefully reviewing our responses.
>
> We are sorry that we made a lot of typos. Coincidentally, the new experiments finished at almost the same time as your new feedbacks and since you were expecting the results we did not pay enough time to proof-read the new contents.
>
> ## **Q4.8: Magnitudes of the noises.**
>
> One of the typos is the magnitude of noises, which should be $~\mathbf{10^{-9}}$ instead of $10^{9}$. For example, one variance from our bounds should be $0.193 \times 10^{-9}$. We have fixed the magnitudes in the newly uploaded revision. To avoid potential confusion, we have put the magnitude into the table.
>
> The average is indeed calculated as you understand. To give more examples, the noise for "label noise" is calculated by averaging the noises from experiments with label noise 0.0, 0.25, 0.5, 0.75.
>
> ## **Q4.9: There are a lot of typos in the newly added sections.**
>
> Regarding other typos, we will further proof-read and fix them in later revisions.
>
> ## **Q4.10: Regarding the weight scaling experiments**
>
> In the scaling of MLPs, we already used fully bias-less MLPs for homogenous networks. However, since we split the training set in to $k=6$ subsets to estimate variances in the (both existing and ours) bounds, the generalization of the MLP is harmed. As a result, the testing accuracy is about $96\\%$ (if we use $k=3$ splits, then it rises to $97\\%$) and the scaling also increases the loss of wrongly classified testing samples and rises the generalization cross entropy.
>
> We will try to further tune the MLPs for higher testing accuracy.
>
> Additionally, to give a demo for 100% accuracies, we have tried to use only correctly classified testing samples to compute both the generalization cross entropy as well as the bounds while wrongly classified samples' losses are forced to be $0$ so that they do not change in the scaling. This is equivalent to replacing the clipped cross entropy loss with the "filtered" clipped cross entropy $\mathbb{I}[\hat{y} = y] \cdot \min(\text{CrossEntropy}(\text{logits}, y), \tau\_{\text{clipping}})$, which is also sub-Gaussian and works well with the bounds. The results can be found in **Figure C.9(b) in Appendix C.6**, where the generalization gap now decreases when weight scaling is sufficiently large. Our bound still well captures the generalization in this setting. Therefore, our bound can well capture the generalization under weight scaling of homogenous networks.
>
> Lastly, all results on weight scaling have been collected in **Figure C.9**, with a dedicated **Appendix C.6** elaborating on the motivations, differences and results of the two weight scaling experiments.

---

> ### Author Response · Authors · 2024-11-27
>
> ## **Q4.11: The typical range of test accuracies**
>
> Of course. They are summarized in tables below:
>
> ### **Accuracies for old experiments**
>
> |   | min (train)   | mean ± std (train)   | max (train)  |   min (test)    |  mean ± std (test)    |  max (test)    |
> |--- | --:  |--:  |--:  |--:   |--:   |--:  |
> |   cifar10, l.r. + b.s.        | 95.10 | 97.78 ± 1.17 | 99.22 | 68.55 | 76.34 ± 4.01 | 81.69 |
> | cifar10, training set   | 98.18 | 98.55 ± 0.25 | 98.81 | 42.34 | 66.56 ± 14.01 | 80.43 |
> | cifar10, label noise   | 95.21 | 96.92 ± 1.29 | 98.69 | 12.99 | 41.78 ± 25.87 | 80.43 |
> | cifar10, width    | 96.43 | 98.39 ± 0.83 | 99.04 | 77.20 | 79.74 ± 1.30 | 81.37 |
> | cifar10, depth  	        | 98.28 | 98.56 ± 0.18 | 98.79 | 79.31 | 79.98 ± 0.55 | 80.95 |
> | cifar10, weight decay   | 95.05 | 97.77 ± 1.57 | 98.82 | 79.49 | 80.08 ± 0.36 | 80.43 |
> | mnist, l.r. + b.s.  	    | 95.26 | 99.28 ± 1.34 | 100.00 | 93.30 | 95.53 ± 0.96 | 96.78 |
> | mnist, training set  	    | 100.00 | 100.00 ± 0.00 | 100.00 | 96.51 | 96.51 ± 0.00 | 96.51 |
> | mnist, label noise  	    | 99.00 | 99.57 ± 0.37 | 100.00 | 16.92 | 54.37 ± 30.09 | 96.51 |
> | mnist, width  	        | 100.00 | 100.00 ± 0.00 | 100.00 | 95.56 | 96.08 ± 0.34 | 96.51 |
> | mnist, depth  	        | 100.00 | 100.00 ± 0.00 | 100.00 | 96.51 | 96.51 ± 0.00 | 96.51 |
> | mnist, weight decay  	    | 100.00 | 100.00 ± 0.00 | 100.00 | 96.20 | 96.39 ± 0.11 | 96.51 |
> | mnist, weight scaling  	| 100.00 | 100.00 ± 0.00 | 100.00 | 96.51 | 96.51 ± 0.00 | 96.51 |
>
> ### **Accuracies of the new experiments**
>
> |   | min (train)   | mean ± std (train)   | max (train)  |   min (test)    |  mean ± std (test)    |  max (test)    |
> |--- | --:  |--:  |--:  |--:   |--:   |--:  |
> | cifar10, l.r. + b.s.    | 95.56 | 97.94 ± 1.02 | 99.32 | 69.45 | 76.55 ± 3.61 | 81.60 |
> | cifar10, training set  	| 96.64 | 98.22 ± 0.82 | 98.84 | 40.63 | 66.12 ± 14.56 | 80.45 |
> | cifar10, label noise  	| 94.21 | 96.74 ± 1.69 | 98.84 | 13.01 | 41.52 ± 25.94 | 80.45 |
> | cifar10, width  	        | 96.15 | 98.36 ± 0.95 | 98.98 | 76.88 | 80.25 ± 1.54 | 81.56 |
> | cifar10, depth  	        | 98.18 | 98.61 ± 0.21 | 98.84 | 78.73 | 79.93 ± 0.66 | 80.74 |
> | cifar10, weight decay  	| 94.54 | 97.79 ± 1.87 | 98.90 | 79.99 | 80.25 ± 0.26 | 80.56 |
> | mnist, l.r. + b.s.  	    | 95.83 | 99.31 ± 1.25 | 100.00 | 93.58 | 95.60 ± 0.91 | 97.10 |
> | mnist, training set  	    | 100.00 | 100.00 ± 0.00 | 100.00 | 89.48 | 93.85 ± 2.59 | 96.61 |
> | mnist, label noise  	    | 99.03 | 99.48 ± 0.36 | 100.00 | 16.67 | 54.22 ± 30.21 | 96.61 |
> | mnist, width  	        | 100.00 | 100.00 ± 0.00 | 100.00 | 95.56 | 96.15 ± 0.43 | 96.61 |
> | mnist, depth  	        | 96.51 | 99.26 ± 1.35 | 100.00 | 89.18 | 94.88 ± 2.73 | 96.61 |
> | mnist, weight decay  	    | 100.00 | 100.00 ± 0.00 | 100.00 | 96.09 | 96.40 ± 0.23 | 96.64 |
> | mnist, weight scaling  	| 100.00 | 100.00 ± 0.00 | 100.00 | 96.05 | 96.35 ± 0.20 | 96.61 |
>
> The testing accuracies are relatively low because we had to split the training set into $k=6$ subsets.
>
> ## **Q4.12: The dependence on training set splitting of existing bounds.**
>
> Yes, the bounds by Neu et al. and Wang et al. do require splitting the training set for the empirical estimation on the variance-like terms and we did do the splitting in the same manner.

---

> > ### Comment · Reviewer_aqMJ · 2024-11-29
> >
> > Thanks for the answer, I have no more questions regarding empirical evaluation. The difficulties regarding evaluation turn out to be milder than I initially anticipated.
> >
> > I think the paper is now much improved compared to the initial version, due to thoroughly addressing all the reviewers suggestions and substantially expanding the paper. Due to this I will raise my score even further to 8.

---

> ### Author Response · Authors · 2024-11-29
>
> We sincerely thank the reviewer for the constructive feedbacks. Your feedbacks have greatly helped us improve the credibility of our empirical evaluation. We also thank the reviewer for raising the score.
>
> We will continue to proof-read and fix typos. If any further questions or concerns come to mind in the future, please do not hesitate to bring them up.

---

### Official Review · Reviewer_Bn1c · 2024-11-04

**Soundness:** 3
**Presentation:** 3
**Contribution:** 2
**Rating:** 6
**Confidence:** 4

**Summary:**

This work belongs to the line of information-theoretic generalization bound and improves upon the results of [Neu et al., 2021] and [Wang and Mao, 2023]. In particular, it utilizes geometry-aware perturbation and establishes a novel generalization upper bound that better captures the influence of flatness-related terms and outperforms [Wang and Mao, 2023] by more accurately predicting the empirical observations. Moreover, the authors devise a more powerful auxiliary sequence called "omniscient trajectory" to bound the generalization error of the trajectory of (stochastic) gradien-based methods. As an application of the novel result, the authors derive the optimal minimax rate for GD under the CLB setting.

**Strengths:**

Overall, this is a solid and well-written paper. The motivation and intuition behind the proposed methodology are well-explained. It provides a refinement for the information-theoretic generalization bound by exploiting the geometric properties of the objective function and by devising a more powerful auxiliary sequence. This allows to better reflect empirical pbeservations and recovers the minimax rate when applying to CLB-based bounds. In this regard, this paper takes a step further towards the understanding of how local geometry might affect the generalization.

**Weaknesses:**

Despite a rather interesting paper, some concerns arise across different aspects.

First, the major concern comes from the novelty. The paper should be regarded as a refinement over [Wang and Mao, 2021] and the generalization bound is not new and decisively better compared to the prior results. Also, the methodology is also not completely original. For example, the use of anisotropic Gaussian perturbation also appears in previous works like [Neu et al., 2021], as mentioned by the authors themselves.

Second, it seems that the applicability of this result slightly suffers from the form of the iterative update in Sec 2.1 and ingredients like the population Hessian. For instance, the iterative update (it would be great if the authors provided a label for the update) might rule out the use of past gradient information and hence exclude many algorithms. It would be great if the author could provide some intuition to show that these ingredients are indeed necessary.

**Questions:**

The authors apply their result to CLB problem and obtain the optimal rate. I would be particularly interested if the author could show how their bound will imply a sample rate when the stochastic function is smooth and realiable, which is provably to have a better rate like O(1/n^{2/3}) or O(1/n) than the CLB setting from the recent upper or lower bound result. This might be unrelated to the major topic but I would greatly appreciate if the authors could answer this.

---

> ### Author Response · Authors · 2024-11-21
>
> We thank the reviewer for their feedback and will respond to the raised questions below.
> We have revised the manuscript after these feedbacks. Modifications corresponding to your feedbacks are colored in **green**.
>
>
>
> ### **W3.1: The methodology is not completely original and the refinement is not decisively better.**
>
> Thank you very much for pointing these out. However, we believe there are "decisive" improvements in our results, in both deep learning and CLB settings:
>
> - **Improvements for deep learning**: The following table shows representative results of the compared generalization bounds when batch size is $30$:
>
>   | Generalization bounds | lr=0.0005 | lr=0.001 | lr=0.005 | lr=0.01 | lr=0.02 |
>   | --- | ---:| ---:| --:| ---:| --:|
>   | Wang and Mao (2021)| 699.220 | 699.084 | 710.599 | 679.398 | 639.065 |
>   | Neu et al. (2021)| 49.117 | 47.527 | 46.938 | 45.530 | 43.197 |
>   | Ours ($\lambda=1$) | 4.324 | 4.570 | 3.333 | 2.186 | 1.695 |
>   | Ours ($\lambda=10^3$) | 1.051 | 1.051 | 0.980    | 0.941 | 0.927 |
>   | Ours ($\lambda=10^9$) | **1.014** | **1.017** | **0.962** | **0.930** | **0.921** |
>   | Generalization error  | 0.988 | 0.980  | 0.949 | 0.929 | 0.916 |
>
>   One can make the following conclusions from the table and Figures 1(a), C.1(a) and 3(a):
>
>   - Quantitatively, our numerical bound is tens of times tighter than existing bounds. The improvement is hundreds of times when compared to [Wang and Mao, 2021].
>   - Qualitatively, our numerical bound sticks very tight to the (estimated) true generalization error.
>
>   Similar observation for more batch sizes.
>
>   After careful examination on the presented results, we find some potential misguidance in the result presentation and have provided a more clear presentation for our results.
>
>   - Some true generalization error are rescaled when plotted: Existing bounds differ from the true generalization error in magnitudes and displaying them as they were will make the variances in the true generalization error almost invisible. To facilitate comparison between existing bounds and the true generalization error, we shift and rescale the true generalization error to adapt to the scale and variances of the existing bounds. Rescaled figures are marked by $(\cdot)^R$. Note that our bounds are close to the true generalization error and the results for our bound in Figure 3(a) are **not** marked by $(\cdot)^R$, i.e., **not** rescaled
>   - We wonder whether it is the rescaling that makes the existing bounds seem to be already close to the true generalization error and confuses the reviewer. We have put more emphasis on the rescaling by explicitly marking the y-axis for the true generalization bound.
>
> - **Improvements on CLB problems**: We have achieved asymptotic improvements over existing representative information-theoretic generalization bounds, which include [Neu et al., 2021] and [Wang and Mao, 2021]:
>
>   - We have arrived at an $O(1/\sqrt{n})$ generalization bound for SGD.
>   - In contrast, [Haghifam et al., 2023], [Livni, 2024] and [Attias et al., 2024] have shown that existing representative information-theoretic bounds has an $O(1)$ lower-bound on CLB problems. [Neu at al., 2021] and [Wang and Mao, 2021] do not focus on CLB problems. But if we adapt the proof of isotropic version Prop. 8 of [Neu et al., 2021] on CLB settings, we will arrive at the bound ((C)MI + isotropic Gaussian perturbation) considered by [Haghifam et al., 2023]. Therefore, it can be shown that our results is asymptotically and decisively advantageous over these existing works.
>   - In the first draft, we thought that the comparison with existing representative MI bounds mentioned in [Haghifam et al., 2023; Attias et al., 2024; Livni, 2024] are enough and omitted [Neu et al., 2021] and [Wang and Mao, 2021] in their non-focusing CLB problems. We have explicitly compare them in Sec 5 in our second draft.

---

> ### Author Response · Authors · 2024-11-21
>
> Regarding the **methodology**, we want to clarify that our contribution is not anisotropic Gaussian noises (as done by [Neu et al., 2021]). Our contribution lies in the exchange between expectation and optimization, which increases the information that optimization can rely on and thus promotes the optimization. Such additional information includes the training set, which allows us to use the instance-level empirical Hessian for "leveraging flatness". It also includes the index $i$ introduced in the individual-sample bound and the leave-one-out training set $Z\_{-i}$, which allows us to easily prove Theorem 3.
>
>   - After careful inspection, we suspect that it is covariance matrix $\Sigma$ used in the introduction that makes the reviewer focus on anisotropic Gaussian noises (correct us if it is not the reason). As we intended, we only wanted to demonstrate the main idea of "moving the optimization (over bound parameter) inside", and selected $\Sigma$ as an example of the bound parameter conveniently from the context (which already demonstrates what the bound looks like before moving the optimization inside). In the main result, the bound parameter is of a more general form instead of anisotropic Gaussian noises.
>   - Now in the introduction of our second draft, we have explicitly stated that $\Sigma$ is only an example of bound parameter. We have also clearly stated that our contribution is not the anisotropic Gaussians or the non-Gaussian general noises. Hopefully it would clarify our contribution.

---

> ### Author Response · Authors · 2024-11-21
>
> ### **W3.2: The applicability of this result slightly suffers from many ingredients of the assumptions and the bound.**
>
> Thank you very much for carefully examining the applicability. We first handle and clarify the applicability issues other than the population Hessian dependence.
> - Regarding the **iterative update**:
>   - Firstly, the iterative update form in Sec 2.1 covers non-iterative single-step algorithms if one sets $T = 1$.
>   - Secondly, although the intermediate result Theorem 1 uses statistics from iterations, the final Theorem 2 only cares about the state of the final weight. Therefore, Theorem 2 is essentially single-step and the iteration form in Sec 2.1 does not harm its applicability.
> - Regarding the **past gradient information**:
>   - We believe this issue is well addressed in Sec 2.1. In the formulation of the iterative algorithm in Sec 2.1, we allow the update function $g\_t$ to access all past weights $W\_{t-1}$ and $W\_{0:t-2}$, as well as all randomnesses $V$. As a result, the algorithm can infer all past gradients: to recover the gradient/update at time $\tau < t$, it can extract the sample indices of batch $B\_\tau$ from $V$, and extract the weight $W\_{\tau-1}$ before step $\tau$, and re-compute $g\_{\tau}$ or the gradient $\nabla \hat{\mathcal{L}}\_{B\_\tau}(W\_{\tau - 1})$. With access to all past gradient information, the formulation can cover momentum and Adam, etc. Therefore, the iterative updates do not rule out the use of past gradients or exclude algorithms relying on them.
>   - We did not remark this extension to adaptive algorithms to save space in our first draft. We have added such remarks at the end of **Sec 2.1 (Line 198)** in the second draft.
>
> Regarding the **population Hessian**, this ingredient is indeed necessary because one must use auxiliary trajectories to derive MI bounds for SGD (as explained in Sec 2.3) and this technique inherently requires the population Hessian or assumptions on it:
> - **Necessity due to auxiliary trajectory technique:** It is an inherent drawback of the auxiliary trajectory techniques, since after switching to the auxiliary trajectory, one must bound the differences between the original and the auxiliary trajectories **to relate the bound for the auxiliary one to the original one**. This difference is mainly measured through empirical and population loss differences if there is no reasonable assumption on losses such as the Lipschitzness or smoothness. In order to approximate the population loss difference, population Hessian must used.
> - **Existing bounds also depend on population Hessians explicitly or implicitly:** As a result of the inherent necessity, representative works of the auxiliary trajectory all somehow rely on population Hessians or other population statistics. For example, [Neu et al., 2021] explicitly includes population losses or Hessians in their flatness term. The main result Theorem 2 of [Wang and Mao, 2021] also includes an explicit use of population statistics in its first inequality. Later, this use is alleviated in the second inequality of [Wang and Mao, 2021]'s Theorem 2 through an assumption that adding noises to weights only increases population loss ($L\_\mu (w\_T) \le \mathbb{E}\_{\xi}[L\_\mu (w\_T + \xi)]$). However, such alleviation may harms the numerical tightness. To sum up, in existing works, they also include such terms, although the dependence is lighter or is alleviated through assumptions on population statistics.
> - Nevertheless, we believe it may be solved in future works by bounding the generalization of higher order statistics, as conjectured at the end of Sec 6.
> - We have added the discussion in **Lines 535-536 and Line 539** and in **Appendix E**.
>
>
>
> ### **Q3.1: Can you extend the bound to smooth settings, where the sample complexity is better than CLB problems?**
>
> Thank you very much for your interest in the extended results. We, as well, are interested in whether our result can recover more existing fast-rate results, especially those stability-based.
>
> In our second draft, we have added in **Appendix D.5** a result for smooth and convex losses with an $O(1/n)$ rate. Specifically, targeting an existing stability-based result from [Lei et al., 2019] (see latter "Additional References"), the results first prove an omniscient information-theoretic (Wasserstain-based) bound on smooth settings. Then we recover the stability-based generalization bound of [Lei et al., 2019]. Finally, by borrowing their bound on stability under convex losses, we recover their excess risk bound at rate $O(1/n)$ up to constants.
>
> Currently, only one new result is added in **Appendix D.5** since it costs some time. We are very happy to add more extensions.
>
> ## **Additional References**
>
> Yunwen Lei, et al. Fine-grained analysis of stability and generalization for stochastic gradient descent. In International Conference on Machine Learning, 2020.

---

> ### Author Response · Authors · 2024-12-03
>
> We sincerely thank the reviewer for the helpful feedbacks. Since the discussion phase will close soon, do you have further questions to discuss? If you have further questions or remarks, please do not hesitate to bring them up.

---

### Official Review · Reviewer_6UMN · 2024-11-04

**Soundness:** 4
**Presentation:** 2
**Contribution:** 3
**Rating:** 6
**Confidence:** 3

**Summary:**

This paper introduces a new approach to developing Information-theoretic Generalization Bounds for Stochastic Gradient Descent (SGD). The core idea is to incorporate an "omniscient" auxiliary trajectory into the proof, which better leverages the flatness of the loss landscape. Previous information-theoretic bounds struggled to align with generalization performance as batch size varied. This paper addresses that limitation, offering bounds that correlate more accurately with generalization across different batch sizes. Additionally, prior bounds for Gradient Descent (GD) on convex-Lipschitz-Bounded (CLB) problems could not achieve the optimal minimax rate of $O(\frac{1}{\sqrt{n}})$, but this paper’s approach successfully reaches this rate.

**Strengths:**

-The contribution is novel and significant: The "omniscient" auxiliary trajectory is original and allows to overcome 2 limitations of previous information-theoretic generalization bounds:
1) correlation between the bounds and observed generalization behavior under different batch sizes
2) $O(\frac{1}{\sqrt{n}})$ rate for Gradient Descent on convex-Lipschitz-Bounded problems.

-The paper effectively motivates the problem and provides a useful high-level overview of the main ideas in the introduction.

-The paper includes rigorous proofs for the results, along with experiments that support the theory. However, I have not verified the mathematical details of the proofs.

**Weaknesses:**

1) The bound requires an additional penalty term to be introduced. This additional term could  worsen the bound in some applications if the benefits to the trajectory term are not high enough. Also, the proof involves two auxiliary trajectories (omniscient and SGLD-like) instead of one, complicating the proof and the bound.

2) In the introduction, it is postulated that the incorrect behavior of the previous bound in regards to varying batch sizes comes from the trajectory term not capturing the flatness more directly. However, in the experiments, the tendency of the new trajectory term w.r.t. learning rate is still incorrect. The overall bound seems to correlate correctly with generalization performance because the additional penalty term behave correctly and because this additional penalty term dominates the bound. It is therefore not clear to me if the paper provides the correct explanation in the introduction about what is ''really'' solving the problem.

**Questions:**

1) Do you think it would be possible to simplify the approach by using only one auxiliary trajectory (the omniscient one) in the proof?
2) Can you provide more clarifications about the point 2) in the weaknesses above?

---

> ### Author Response · Authors · 2024-11-21
>
> We thank the reviewer for their feedback and will respond to the raised questions below.
> We have revised the manuscript after these feedbacks. Modifications corresponding to your feedbacks are colored in **blue**.
>
> ### **W2.1: The new term may worsen the bound. The proof involves two auxiliary trajectories, complicating the proof.**
>
> Thank you very much for illustrating a potential suboptimal case for our bound.
>
> - Regarding **the new terms**:
>   - It is indeed possible that the new terms worsen so much that the improvement on the trajectory terms cannot compensate.
>   - However, the omniscient trajectory and the new term are not fixed and can be flexibly optimized. If the new terms introduced by the optimization in Theorem 2 is too bad, then one can simply set $\Delta g\_t = 0$, and roll back to the existing bounds. As a result, our bound is at least as good as existing bounds even the additional term worsens too much.
>   - Moreover, since omniscient trajectory is optimized in the instance level as discussed in Sec 3.1, such rollback can also be cone in the instance level. That is, one does not need to compute Theorem 2 and computing Neu et al.'s Prop. 8 and select one of them *uniformly* in the population level, i.e., computing $\min(\mathbb{E}(\text{terms from Theorem 2}), \mathbb{E}(\text{terms from Prop. 8}))$. Instead, one can **pointwisely** compute $\mathbb{E}[\min(\text{terms from Theorem 2}, \text{terms from Prop.8})]$, which is provably smaller than any of Theorem 2 and Prop. 8 in the face of bad Theorem 2 terms.
> - Regarding the **complexity of the two trajectories**: We totally agree with you that the two trajectories are more complicated. However, we believe such complexity is manageable since the two trajectories have clear and decoupled roles, which are also historically inherited from existing constructions.
>   - Omniscient trajectory **optimizes the bound**. Its role is similar to the noise variances or covariances in the work of [Neu et al., 2021] and [Wang and Mao, 2021]. It can be replaced by existing constructions in proofs.
>   - SGLD-like trajectory **bounds the MI** of other trajectories. Its role is exactly the same as in existing works. It can be replaced by other techniques for bounding MI in proofs.
>   As a result, the proof can be decomposed into two major steps and then addressed independently using their respective methods. This decoupled nature of the proof leads to manageable complexity.
>
>   Moreover, in some cases, one can get rid of the SGLD-like trajectory, thus simplifying the proofs. This is elaborated in our response to your Q2.1.
>
> We have added the discussion in **Lines 280-281** and **Lines 323-326**.

---

> ### Author Response · Authors · 2024-11-21
>
> ### **W2.2: The tendency w.r.t. learning rate is still incorrect.**
>
> Thank you for pointing that out. Although trajectory term's dependence on learning rate is still qualitatively incorrect, we argue that the wrong dependence is fixed partially and quantitatively.
>   - From Figures 1(b) and C.1(b) for existing bounds, one can see that the curves for different learning rates are well separated. In contrast, from Figure 3(b) for our bound, the curves for different learning rates are much closer and some of them twist together. As a result, the twisting indicates an alleviation to the wrong negative dependence and we conclude that our technique partially fixes the wrong dependence to learning rate.
>
> We also conjecture the reason for why tendency is fully corrected w.r.t. batch size but not  w.r.t. the learning rate:
>   - After optimization, the trajectory term is a product between weight variances and flatness. As a result, larger variance increases the trajectory term.
>   - Increasing learning rate increases step sizes and increases the weight variance. Therefore, increasing learning rate worsens weight variance and improves flatness at the same time, which may cancel each other. It turns out that the worsen weight variance overpowers the improved flatness, making the dependence on learning rate still incorrect. However, since the flatness is improved, the worsening of trajectory term is slowed down, leading to the partial improvement.
>   - Smaller batch size also potentially increases the weight variance. However, it turns out that it is not the case:  The trajectory term of Neu et al.'s Prop. 8 is exactly the weight variance, if one restricts the noises to be isotropic and does not divide the variance by the $\Sigma$ or $\sigma$. This form is exactly diagramed in Figure C.1(b). Interestingly, Figure C.1(b) shows that the weight variance is almost unrelated to batch size. As a result, decreasing batch size will not increase the weight variance and the improvement on flatness dominates and fully corrects the trajectory term.
>   - Therefore, the competition between flatness and covariance can explain the experimental results. We consider the inadequacies of flatness exploitation is indeed behind the wrong tendencies of existing bounds.
>
> We have added such explanations to **Sec. 4** (Lines 461-465) as well as **Appendix C.4.1** (Lines 1377-1394).
>
> ### **Q2.1: Do you think it would be possible to simplify the approach by using only one auxiliary trajectory (the omniscient one) in the proof?**
>
> Yes. In some cases the auxiliary trajectories can be simplified into one.
> - An example is Theorem 3, whose proof only uses omniscient trajectory without SGLD-like trajectory. This is because that SGLD-like trajectory is only used to bound the MI for other trajectories. However, in Theorem 3, the omniscient trajectory has a very special form: (conditionally) constant. As a result, its MI can be easily computed (0) and SGLD-like trajectory is no longer needed.
> - More generally, if one can restrict the omniscient trajectory to be some special form such that its MI can be easily bounded, then SGLD-like trajectory is no longer needed, simplifying the proof.
> - We have added this potential in **Remark D.4**.
>
> ### **Q2.2: Can you provide more clarifications about the point 2) in the weaknesses above?**
>
> It is explained in the response to W2.2.

---

> > ### Comment · Reviewer_6UMN · 2024-12-03
> >
> > Thank you for the responses. I will keep my position on the side of acceptance.

---

> > > ### Author Response · Authors · 2024-12-03
> > >
> > > We sincerely thank the reviewer for encouraging us to supplement the discussion of experimental results and to explore the potential future development of the proposed technique. And also thank you very much for recommending acceptance. If you have more questions or remarks, please do not hesitate to post them.

---

### Official Review · Reviewer_CXNU · 2024-11-04

**Soundness:** 3
**Presentation:** 3
**Contribution:** 3
**Rating:** 8
**Confidence:** 4

**Summary:**

This paper studies improved information-theoretic generalization bounds for SGD by more effectively incorporating the concept of flatness. Specifically, the authors point out limitations in previous information-theoretic generalization bounds, which fail to align well with some generalization phenomena observed in deep learning, such as the tendency for flatter minima to generalize better. The authors then derive new information-theoretic generalization bounds by constructing an "omniscient trajectory", which allows for a better capture of the flatness of SGD solutions in the loss landscape. Empirical results demonstrate that these bounds improve upon previous ones. Additionally, the authors extend their analysis by applying techniques from previous works, such as the individual technique, to mitigate limitations of information-theoretic bounds in stochastic convex optimization (SCO) problems.

**Strengths:**

1. This paper introduces novel information-theoretic generalization bounds for SGD that improve upon previous results, particularly when SGD finds flat minima.

2. Constructing the auxiliary weight process twice, specifically through the omniscient trajectory, is a notable technical contribution in this field.

3. The results provide an explanation for the learnability of some SCO problems, where many previous information-theoretic bounds do not apply (e.g., non-vanishing bounds).

4. The paper is clearly written and easy to follow.

**Weaknesses:**

1. Some related works, whether cited or uncited in this paper, require further discussion in relation to these results. For example, [R2] also discusses limitations in [R1] and suggests that, if the SGD process can be well-approximated by SDEs (motivated by empirical observations), the auxiliary process could be the SDE approximation of SGD, with perturbed Gaussian noise depending on the training data and current state (i.e. depending on $S$ and $W_{t-1}$ at step $t$). Moreover, while not directly related to the generalization of SGD, [R3], which is missing in this paper, investigates the optimal covariance matrix of Gaussian noise in SGLD. Since these works also consider more complex geometric structures for anisotropic Gaussian noise, and emphasize the importance of alignment between flatness and the covariance of weight distributions for generalization, it would be beneficial to discuss whether the results here are comparable or related to them in any sense, such as [R2, Theorem 3.2].

[R1] Ziqiao Wang, and Yongyi Mao. "On the Generalization of Models Trained with SGD: Information-Theoretic Bounds and Implications." International Conference on Learning Representations 2022.

[R2] Ziqiao Wang, and Yongyi Mao. "Two Facets of SDE Under an Information-Theoretic Lens: Generalization of SGD via Training Trajectories and via Terminal States." The Conference on Uncertainty in Artificial Intelligence 2024.

[R3] Bohan Wang, et al. "Optimizing information-theoretical generalization bound via anisotropic noise of SGLD." Advances in Neural Information Processing Systems 2021.

2. In Section 3.4, assuming that $\nabla\hat{\mathcal{L}}_S(W_T)\approx0$ could limit the applicability of your results in modern deep learning. For example, in some settings, especially for large-scale neural networks, models can perform well without vanishing gradient norms, as noted in [R4]. Can the results here explain generalization in such cases?

[R4] Jingzhao Zhang, et al. "Neural network weights do not converge to stationary points: An invariant measure perspective." International Conference on Machine Learning 2022.

3. In the proof of Theorem 1, particularly in Eq. (B.4), why not let $\Omega(\tilde{w}\_{0:t-1})=\tilde{w}\_{t-1}-\mathbb{E}[g\_t]-\mathbb{E}[\Delta g\_t]$? This choice seems valid if the expectation is taken over a carefully chosen distribution. Such a choice would result in an additional term, $-\mathbb{E}[\Delta G]$ in the trajectory term of Corollary B.1. Could you clarify whether this alternative choice would obtain a tighter bound?

4. In Theorem 3, while I do believe the contribution to information-theoretic generalization bounds here is novel, it seems similar in insight to [R5]: namely, if the learning algorithm is stable, information-theoretic generalization bounds can explain the learnability of these CLB problems. The main distinction in Theorem 3 is that the stability parameter is explicitly expressed in terms of $T$ and $\eta$, as in Bassily et al. (2020), while [R5] simply uses the stability parameter. In other words, both this work and [R5] explain the generalization of stable algorithms (e.g., GD with smooth or non-smooth convex Lipschitz loss) but may not be able to overcome the limitations highlighted in CLB counterexamples in [R6].

[R5] Ziqiao Wang, and Yongyi Mao. "Sample-conditioned hypothesis stability sharpens information-theoretic generalization bounds." Advances in Neural Information Processing Systems 2023.

[R6] Idan Attias, et al. "Information Complexity of Stochastic Convex Optimization: Applications to Generalization, Memorization, and Tracing." International Conference on Machine Learning 2024.

Minor comments:

1. Some citation errors should be corrected: Wang \& Mao (2021) was published in ICLR 2022, not NeurIPS 2021. The evaluated variant of CMI bounds is first introduced in Section 6.2 of the arXiv version of Steinke \& Zakynthinou (2020). Additionally, Hafez-Kolahi et al. (2020) did not use evaluated variants of CMI, so the sentence "the evaluated variants (Hafez-Kolahi et al., 2020; Harutyunyan et al., 2021; Wang \& Mao, 2023b)" is incorrect. A missing reference here is [R7], which applies individual e-CMI bounds to obtain new generalization bounds. Accordingly, the correct sentence should be "the evaluated variants (Harutyunyan et al., 2021; Hellström & Durisi, 2022; Wang & Mao, 2023b)" to reflect $f$-CMI, e-CMI and ld-CMI.

[R7] Fredrik Hellström, and Giuseppe Durisi. "A new family of generalization bounds using samplewise evaluated CMI." Advances in Neural Information Processing Systems 2022.

2. The Hessian notation is inconsistent: in Section 2.2, $\hat{H}\_w$ and $H\_w$ are used to denote the Hessians at $w$, but in Line 218, $\hat{H}\_s$ is used, and in Theorem 2, $\hat{H}\_s(w)$ and $H\_\mu(w)$ are used. Please clarify these notations for consistency.

**Questions:**

1. How did you estimate the trace of the Hessian (or other matrices) and the inverse of the matrix in your experiments?

2. In Eq. (1), it seems the expression could potentially be simplified by setting $\xi\sim\mathcal{N}(\gamma,\Sigma)$, is this valid?

3. In Eq. (3), what specifically does $H$ represent in $I+H/2\lambda C$? Also, the notation for $H(W)$ is unclear here; you state that "$H(w)$ is defined as $\Delta H(w)$ or $\hat{H}_S(w)$", yet in Appendix B.5, it seems that $H(w)$ should correspond to $\Delta H(w)$. Furthermore, regarding the definition for $C$, does $H(W_T)$ refer to the population Hessian? The notations about Hessian become somewhat confusing due to their inconsistency.

4. In Line 356, the phrase "such $\Delta g_t$ is optimal under ..." appears. Where is the justification for this optimality provided in the text?

4. In the proof of Theorem 3, could you clarify why the CMI term in Eq. (D.4) takes the form $I(W_T+\sum_{t=1}^T\Delta g_t^i;Z_i|Z_{-i})$? I may have overlooked something, but it seems that the construction might no longer require the Gaussian noises.

5. Is there potential to extend your analysis to SGD with momentum? Neu et al. (2021) give a method for extending their analysis to SGD with momentum, though they acknowledge limitations in demonstrating any advantage for SGD with momentum over vanilla SGD using their method.


**I enjoyed reading this paper, and I would be happy to increase my score if the authors could adequately address my concerns.**

---

> ### Author Response · Authors · 2024-11-21
>
> We thank the reviewer for their feedback and will respond to the raised questions below.
> We have revised the manuscript after these feedbacks. Modifications corresponding to your feedbacks are colored in **orange** or **red**.
>
> ### **W1.1: Discussions and comparisons with similar notions of alignment from existing works are missing.**
>
> Thank you very much for the suggestion. We have added discussions and comparisons with the works you mentioned in the **Introduction** (Lines 136-137), **Sec. 3.4** (Lines 404-407) (concisely) and **Appendix** **B.8.1** (in detail). They are summarized as the following:
> - **Relation to [R2]**: Briefly, our alignment is similar to [R2]'s in terms of the alignment between state-dependent weight/gradient statistics and state-dependent local geometry. Our alignment differs from [R2]'s by directly using Hessians instead of gradient/weight covariances as proxies. This allows us to better "leverage flatness".
> - **Relation to [R3]**: [R3] intends to optimize the noise structure of SGLD with information-theoretic bounds as a proxy to the true risk. If we put [R3]'s information-theoretic bound and their algorithm together, the consequent results' similarities and differences with our alignment are similar to [R2]. Nevertheless, we find more similarities and differences on how local geometry in the alignments affects a trade-off between lower information and lower losses. Details can be found in **Appendix B.8.1**.
>
> ### **W1.2: The assumption that $\nabla \hat{\mathcal{L}}_S(W_T) \approx 0$ is too strong.**
>
> Thank you for pointing out this issue. However, as we intended, the assumption is introduced only to simplify computation in approximation of Sec 3.4. It is only used in Sec 3.4 and is not required by Theorem 1, Theorem 2 or later experiments. Therefore, the main results Theorem 1 and 2 work with $\|\nabla \mathcal{L}_S\| \gg 0$.
>
> We have added an explicit remark on the limited use of this assumption (**Line 383, Lines 386-388**). We have also added extra results without this assumption at the end of **Appendix B.8** (second draft).
>
> ### **W1.3: Theorem 1 can be further tightened.**
>
> Thank you very much for such careful examination and your efforts and exploration to refine the results.
>
> The modification is indeed valid because given the omniscient trajectory, its perturbation expectation is a constant and can be depended on by $\Omega$. It would result in an extra $-\mathbb{E}[\Delta G]$, which indeed tightens Theorem 1 since $\min_{a} \mathbb{E}[\|X - a\|^2]$ is minimized at $a=\mathbb{E}[X]$.
>
> We also have tried this alternative before submission. However, the omniscient trajectory in Theorem 1 is not the final result and must be optimized later. Unfortunately, this extra term  makes optimization much harder. The expectation $\mathbb{E}[\Delta G]$ depends on all $\Delta G$ of all trajectory $W_{0:t}$ instances. As a result, we have to consider the omniscient trajectory for *other* instances even if we are optimizing the omniscient trajectory for one instance, as done in  Appendix B.6. This may be addressed by more advanced optimization techniques, e.g., the variational method, at the cost of complicating the proof very much. According to experiments, the current Theorem 1 and simple convex optimization is already efficient enough.
>
> We have included the discussion in **Remark B.1** at the end of Theorem 1's proof.

---

> > ### Author Response · Authors · 2024-11-21
> >
> > ### **W1.4: The contribution of Theorem 3 is very similar to [R5].**
> >
> > Thank you very much for the careful comparison.
> >
> > Our Theorem 3 is indeed similar to [R5] in contribution as generalization bounds. But we believe they have important differences when put in the context of their motivations. Before we elaborate on the differences, we must clarify the similar part of the motivation of [R5] and our Theorem 3.
> >
> > - **Clarification on motivation**: Although the results are about stable algorithms' generalization, we must emphasize that the **research focus** of [R5] (at least for its CLB section) and our Theorem 3 is **information-theoretic (IT) bounds** instead of stable algorithms' generalization, as the latter is already proved by stability-based arguments. [R5] and our Theorem 3 both focus on the drawback pointed out by [R6,R8,R9] that **IT bounds cannot explain** the generalization of stable algorithms, and try to defend the IT theories by improving them. Therefore, it is more important to compare our contributions according to  how results defend or improve IT theories, instead of how results prove the generalization of stable algorithms.
> >
> > - **Contribution differences**: [R5] only partially defends and improves IT theories on **two** CLB subproblems, while our improvement is on **all** CLB problems.
> >
> >   - In terms of defending IT theories, [R5] improved IT bounds by proving stable algorithms' generalization on **only two subproblems** of CLB problems, one of which is the counterexample of [R8], which only partially defends IT theories on CLB problems.
> >   - However, [R8]'s subsequential work [R6] proposes other counterexamples. Since more counterexamples can be found by later works, one cannot fully defend IT bounds by solving CLB subproblems one by one.
> >   - To end the arms race, one must prove generalization on **all** CLB problems using IT bounds. This is exactly what our Theorem 3 has done, marking its advantage over [R5] in contribution.
> >   - We have added an explicit remark on the 2-vs-all difference in **Sec 5 (Lines 502-505)**.
> >
> > ### **Minor Weaknesses**
> >
> > #### **Citations errors.**
> >
> > Thank you very much for carefully examining the citations. We have fixed such mistakes as you advised (**Line 052, Line 410**).
> >
> > #### **Inconsistent notations.**
> >
> > Thank you very much for pointing them out. The Hessian notations in Sec 2.2 are typos, which should be the same as in Theorem 2. The dependence to $w$ is omitted for simplicity in Line 218. We have corrected typos and written the dependences explicitly (**Line 202 and Line 225**).

---

> ### Author Response · Authors · 2024-11-21
>
> ### **Q1.1: How did you estimate the trace of the Hessian (or other matrices) and the inverse of the matrix in your experiments?**
>
> - For Hessian traces, we use PyHessian [R10], as in [R1].
> - We did not compute Hessian inverses explicitly because they are too large to store and compute. We directly compute inverse-Hessian-vector product (iHVP) in the bounds. Specifically, iHVPs are computed using conjugate gradient method [R11], which only requires Hessian-vector products. We repeatedly run conjugate gradients to obtain iHVP $\hat{u} \approx H^{-1} v$ between $H$ and $v$ until $\|H \hat{u} - v\|^2$ is smaller than $1\%$ of $\|v\|^2$ or the maximum iteration 20 is reached.
> - Explanations have been added in **Sec C.1** in our second draft.
>
> ### **Q1.2: Can the expression in Eq. (1) be further simplified?**
>
> Thank you very much for your careful examination and your efforts to refine the notations. The simplified definition is indeed valid.
>
> However, we intentionally keep $\gamma$ and $\Sigma$ separate because they will be used differently. In later Theorems, $\gamma$ will be substituted by the (translational) perturbation corresponding to the omniscient trajectory, while $\Sigma$ corresponds to the (Gaussian) perturbation form the SGLD-like trajectory (see Theorem 1 in Line 334-343). To echo this correspondence, we intentionally write $\gamma$ and $\Sigma$ saperately.
>
> ### **Q1.3: Confusing notations in Eq. (3) and Appendix B.5**
>
>
> Here is our clarification on the original notation:
> - In $I + H/2\lambda C$, $H$ is the "$H(W_T)$" that "is defined as $\Delta H(W_T)$ or $\hat{H}_S(W_T)$". We omitted the "default" dependence of $W_T$.
> - In Appendix B.5,  $H(w)$ is still $\Delta H(w)$ or $\hat{H}_S(w)$. This confusion presumably arises from the first formula (L936-L940, first draft) of Appendix B.5 (correct us if it is not the case), because $H$ and $C$ are used after it. But we did not introduce $H$ or $J$ in the first formula. Instead, it only recalls Corollary B.1 and conducts second-order approximation. $H$ and $J$ are only inserted at the second formula (L957, first draft) in Appendix B.5.
> - In $C$, $H$ is still "$H(W_T)$" that "is defined as $\Delta H(W_T)$ or $\hat{H}_S(W_T)$"
>
> According to your feedback, we have refined the notation for these various Hessians:
> - We have removed the omission of the dependence on $W_T$.
> - We have given $H$ a new symbol that drastically differs from other Hessians: $\tilde{H}$. We also force $\tilde{H}$ to be $\Delta H(W_T)$ or $\hat{H}_S(W_T)$, i.e., $\tilde{H}$ is tied up with $W_T$.
> - We have also split $H$ or $\tilde{H}$ into two notions $\tilde{H}\_{\text{flat}}$ and $\tilde{H}\_{\text{pen}}$: According to the reviewer aqMJ's suggestion on experiments, we have allowed existing bounds to explicitly depend on validation sets to estimate population Hessians in the flatness term for better comparison and rerun some experiments. To align with the existing bounds, we must make the flatness term also depend on population Hessians. In the first draft, the population Hessian have already exist in the flatness term, but it is not considered in the surrogate optimization of the omniscient trajectory, which involves $C$ that uses $H$, selected as $\hat{H}_S(W_T)$ in experiments. Therefore, $C$ and $H$ are better to depend on $H\_{\mu}(W\_T)$. However, the estimation requires $H$ to avoid dependence on $H\_{\mu}(W\_T)$, otherwise we have to invert population statistics, which is harder to estimate. Therefore, we split the old $H$ into two: $\tilde{H}\_{\text{flat}}$ and $\tilde{H}\_{\text{pen}}$. Both can be chosen from $\Delta H(W\_T)$ or $\hat{H}\_S(W\_T)$ independently.
> - Lastly, we have explicitly explained in the second draft when and why we introduce $\tilde{H}\_{\text{flat}}, \tilde{H}\_{\text{pen}}$ and $J$ (Lines 1050-1057). We have also added a hint that they are not yet introduced in the first formula (**Line 1042**).
> - The modifications have been applied to the second draft on every presence of $H$. With drastically different notations, explicit dependencies and remarks, the above confusions should be addressed.
>
> ### **Q1.4: How to justify the optimality of $\Delta g_t$ in Line 356?**
>
> The perturbation $\Delta g_t$ is optimal essentially due to the convexity of the squared $L_2$ norm. The details can be found in the newly added **Appendix B.4** (second draft).

---

> ### Author Response · Authors · 2024-11-21
>
> ### **Q1.5: Could you clarify the CMI term in Eq. (D.4)? Is it true that the proof of Theorem 3 no longer needs Gaussian noises?**
>
> Thank you very much for carefully reading the proofs. Here are our clarifications:
> - **Regarding the "CMI" bound**: It is in fact not a CMI bound, but an individual-sample bound as hinted at by the paragraph before Eq. (D.4). It essentially starts from the vanilla individual-sample bound with terms $I(\dots; Z\_{i})$. After that, recall if $C$ is independent to at least one of $A, B$, then $I(A; B) \le I(A; B \mid C)$. Applying this with $C = Z\_{-i} \perp Z\_{i}$ leads to $I(\dots; Z\_{i}) \le I(\dots; Z\_{i} \mid Z\_{-i})$. We use this form because we want to recover stability arguments, which essentially involves keeping $Z\_{-i}$ the same while resampling $Z\_{i}$. By putting $Z\_{-i}$ in the condition, one can make it intuitively "given" or "the same". By putting $Z\_{i}$ in the conditional MI, one can make it intuitively "randomly change" or "resampled".
> - **Regarding the missing Gaussian noises in Theorem 3**: It is correct that the construction in Theorem 3 no longer requires Gaussian noises or SGLD-like trajectory. This is because their function is to bound the MI of other trajectories since their proposal by Neu et al (2021). However, in Theorem 3, the omniscient trajectory has a very special form, i.e., (conditionally) constant. Its (conditional) MI is then very easy to compute, i.e. 0. Therefore, Gaussian noises are no longer needed. This example also inspires us that we can restrict the form of omniscient trajectory so that its MI can be easily bounded to get rid of the Gaussian noises and simplify the proof. Thank you very much for leading us to such inspiration. We have added explanation of Gaussian noises's exclusion in **Remark D.4** in the proof of Theorem 3.
>
> ### **Q1.6: Is there potential to extend your analysis to SGD with momentum?**
>
>
> Yes, it can be done. Theorem 1 and 2 are already applicable for SGD with momentum and one only needs to turn on momentum when running experiments. This is because they allow algorithm to access all past weights as in Sec 2.1 and recover all past gradients. As a result, this formulation covers momentum, Adam or other adaptive algorithms. However, the bound only captures the relationship between the distribution of output weight and generalization without modeling the relationship between momentum and output weights. As a result, the bound does not explicitly depend on the momentum and only the distribution of weights can be implicitly affected by it. Therefore, the advantage of momentum can only be demonstrated through numerical experiments.
>
> We have added the discussion in **Lines 196-198**.
>
> ## **Additional References:**
>
> [R8] Mahdi Haghifam, et al. "Limitations of information-theoretic generalization bounds for Gradient Descent methods in Stochastic Convex Optimization." COLT 2023
>
> [R9] Roi Livni. "Information theoretic lower bounds for information theoretic upper bounds." Advances in Neural Information Processing Systems 2024.
>
> [R10] Zhewei Yao, et al. "PyHessian: Neural Networks Through the Lens of the Hessian." IEEE International Conference on Big Data (Big Data) 2020
>
> [R11] Mathieu Dagreou, et al. "How to compute hessian-vector products?" ICLR Blogposts 2024, https://iclr-blogposts.github.io/2024/blog/bench-hvp/#inverse-hessian-vector-products-ihvps-in-optimization.

---

> > ### Comment · Reviewer_CXNU · 2024-11-25
> >
> > Thanks for your detailed responses!
> >
> > I have some additional questions and comments regarding Theorem 3:
> >
> > 1. This should be a straightforward question, but it may arise from my forgetting some of the notation used in the paper: Could you elaborate on why $W_T+\Sigma_T^i=\mathbb{E}[W_T|Z_{-i}]$ in Line 1989 (of your revised version)? If this is correct, then using the original individual bound in Theorem 3 should be sufficient, as $I(\mathbb{E}[W_T|Z_{-i}];Z_i)=0$ due to the independence between $\mathbb{E}[W_T|Z_{-i}]$ and $Z_i$.
> >
> > 2. Regarding the comparison to [R5], I’d like to clarify my earlier concern: The main argument in [R5] about the CMI bound in CLB is that as long as uniform stability can explain generalization, their refined CMI bound can also provide an explanation (see [R5, Corollary 5.1]). Thus, their result is not limited to demonstrating the learnability of just the two specific examples explicitly provided.
> >
> > Your Theorem 3 focuses on CLB problems solved by GD, and it is known that GD, with certain $T$ and $\eta$ is uniformly stable for those CLB problems. In this sense, both your work and [R5] demonstrate that the learnability of CLB problems optimized by a stable GD can be explained within the IT generalization framework. However, note that [R5] considers more general stable algorithms, which may extend beyond the case of GD, making their result potentially applicable to broader scenarios.
> >
> > More importantly, [R6] highlights a "CMI-accuracy tradeoff" in certain CLB examples: for  **any** $\epsilon$-learner, whose generalization error is $\leq O(\epsilon)$ ($\epsilon$ vanishes with increasing $n$ increasing), then the (individual) CMI or IOMI of this learner  $\geq\Omega(\frac{1}{\epsilon^2})$. This is a strong result, as it applies to *all* $\epsilon$-learners in their examples, not just GD with specific $T$ and $\eta$. Consequently, since your Theorem 3 is constrained to GD and [R5] focuses on stable learning algorithms, neither can fully overcome the limitations highlighted by [R6] in their CLB examples. I hope the authors could clarify this point in the paper.
> >
> > 3. Given the specific choice in your construction, Theorem 3 no longer seems related to flatness. Have you considered emphasizing "omniscient trajectory" in the title instead of "flatness"? The omniscient trajectory in your construction seems highly flexible, with flatness being only a special case. This is, of course, a subjective suggestion of mine.

---

> > > ### Author Response · Authors · 2024-11-27
> > >
> > > Thank you very much for the detailed clarification on your concerns.
> > >
> > > ## **Q1.7: Could you elaborate on Line 1989? If it is correct, are original individual bounds already sufficient?**
> > >
> > > ### **Details of $W\_T + \Gamma\_T^i = \mathbb{E}[W\_T \mid Z\_{-i}]$**
> > >
> > > In the proof of Theorem 3, we select $\Delta g\_t^i = g\_t - \mathbb{E}[g\_t \mid Z\_{-i}]$. After the correction, we have
> > >
> > > $$
> > > \begin{aligned}
> > >     W\_T + \Gamma\_T^i
> > >     =&  \left(W\_0 - \sum\_{t=1}^T g\_t\right) + \sum\_{t=1}^T \Delta g\_t^i & (\text{Definition of iterative algorithm and } \Gamma\_T^i)\\\\
> > >     =&  \left(W\_0 - \sum\_{t=1}^T g\_t\right) + \sum\_{t=1}^T (g\_t - \mathbb{E}[g\_t \mid Z\_{-i}])\\\\
> > >     =&  W\_0 - \sum\_{t=1}^T \mathbb{E}[g\_t \mid Z\_{-i}]
> > > \end{aligned}
> > > $$
> > >
> > > There is an implicit assumption of Bassily et al.'s stability bound: $W\_0$ is fixed (but can be any value, see their Algorithm 1 for their Theorem 3.2). We follow this assumption and arrive at
> > >
> > > $$
> > > \begin{aligned}
> > >     W\_T + \Gamma\_T^i
> > >     =&  W\_0 - \sum\_{t=1}^T \mathbb{E}[g\_t \mid Z\_{-i}]\\\\
> > >     =& \mathbb{E}\left[W\_0 - \sum\_{t=1}^T g\_t \mid Z\_{-i}\right]\\\\
> > >     =&  \mathbb{E}[W\_T \mid Z\_{-i}] & (\text{Definition of iterative algorithms})
> > > \end{aligned}
> > > $$
> > >
> > > We have added the assumption that $W\_0$ is fixed in the newest revision.
> > >
> > > ### **The use of original individual bounds**
> > >
> > > To some extent, the original individual bounds are indeed sufficient after we have derived $W\_T + \Gamma\_T^i = \mathbb{E}[W\_T \mid Z\_{-i}]$. This is what exactly the proof is doing: applying the original individual bound on the **omniscient trajectory**, as in Eq.(D.4).
> > >
> > > However, without the omniscient trajectory, i.e., if one directly applies the individual-sample bound to the original trajectory, the consequential MI term $I(W\_T; Z\_{i} \mid Z\_{-i})$ cannot be reduced to $0$ because $W\_T$ depends on $Z\_{i}$. This is why we use the omniscient trajectory here, i.e., to get rid of the dependence on $Z\_i$.
> > >
> > > To sum up, it is indeed sufficient to using the original individual bound, but only with the help of the omniscient trajectory.
> > >
> > > ## **Q1.8: Comparison with [R5]**
> > >
> > > Thank you for further clarifying your concerns, which involve essentially the following aspects:
> > > - **Q1.8.1**: [R5]'s refined CMI bound already recovers uniform stability and thus extends to all CLB problems.
> > > - **Q1.8.2**: Our bound only handles GD, while [R5] handles stable algorithms
> > > - **Q1.8.3**: [R6] highlights CMI-accuracy trade-off instead of solely the drawback of MI bounds. This trade-off happens to **any** $\epsilon$-learners and prove positive results for stable algorithms are not sufficient.
> > >
> > > We will address these concerns in the following:
> > >
> > > ### **Q1.8.1**
> > >
> > > Thank you very much for your clarification. [R5]'s CMI bound can indeed explain more general stable algorithms and on all CLB problems as well. We are sorry to misinterpret the result because [R5] seems to emphasize on "examples" a lot.
> > >
> > > Then contributions are indeed very similar. Yet, our technique is simpler and very consistent across different applications. The improved CMI of [R5] in their Theorem 4.1 and Corollary 5.1 involves a lot of fancy CMIs with complex random processes. In contrast, our result uses only individual MIs. The structure of our technique is consistent over all applications (constructing omniscient trajectory, bounding it and optimizing it). Moreover, our bound provides intuitive partial solution to the CMI-accuracy trade-off for more $\epsilon$-learners (See response to your Q1.8.3), while the method of [R5] would arguably struggle (see discussions below). Surprisingly, a similar construction of the omniscient trajectory is used in the proof for the $\epsilon$-learners, further demonstrating the simplicity and consistency of our results.
> > >
> > > The results of [R5] also seems tied up with stability, which replaces the sub-Gaussianity coefficient in the (C)MI bounds. As a result, although [R5] applies to general stable algorithms, its applicability seems restricted to the stable algorithms. In contrast, our technique is not restricted to stable algorithms. As a result, our technique can yield a new information measure that vanishes for some "well-behaved" $\epsilon$-learners. See our response to Q1.8.3 for more details.
> > >
> > > We have corrected the comparison with [R5] in Sec 5 in the newest revision, emphasizing their contribution on all CLB problems.

---

> > > ### Author Response · Authors · 2024-11-27
> > >
> > > ### **Q1.8.2**
> > >
> > > Our bound for CLB problems can also extend to stable algorithms, although we did not state this explicitly in Theorem 3 to save space and fit in the 10 pages.
> > >
> > > The proof of Theorem 3 can be outlined as: "resemble the uniform stability argument" using omniscient trajectories and then apply the stability bound of Bassily et al. (2020).
> > > Specifically for the first part, before Line 1997, we have not relied on any specific properties of GD, except for two assumptions:  the initial weight is fixed and $W\_{0:T}$ always stay in $\mathcal{W}$ where the loss is defined. After these steps, we have arrived at that the iterative algorithm's generalization can be bounded by
> > >
> > > $$
> > > \begin{aligned}
> > >     \frac{2L}{n} \sum\_{t=1}^n \mathbb{E}[\\|\Gamma^i\_T\\|]
> > >     =& \frac{2L}{n} \sum\_{i=1}^n \mathbb{E}[\\|W\_T - \mathbb{E}[W\_T \mid Z\_{-i}]\\|]\\\\
> > >     \le&  2L \frac{1}{n} \sum\_{i=1}^n \mathbb{E}\_{Z\_{-i}}[\\|W\_T - W'\_T\\| \mid Z\_{-i}]  & (\text{Corollary A.1})
> > > \end{aligned}
> > > $$
> > >
> > > Since $\mathbb{E}\_{Z\_{-i}}[\\|W\_T - W'\_T\\| \mid Z\_{-i}]$ measures the expected difference between two random weights produced given the same other $n-1$ samples, it is equivalent to the weight difference after the $i$-th sample is resampled, which is the expectation version of uniform stability. As a result, $\frac{1}{n} \sum\_{i=1}^n \mathbb{E}\_{Z\_{-i}}[\\|W\_T - W'\_T\\| \mid Z\_{-i}]$ is upperbounded by the uniform stability.
> > > Only in the last step, we use introduce GD by using Bassily et al. (2020)'s stability of GD.
> > >
> > > Therefore, our Theorem 3 can be extended to more general stable algorithms by only writing that explicitly in the statement. This is already done by in Proposition 3 of Appendix D.5, where we prove our bound recovers stability bounds under smooth losses.
> > > We have added explicit include of stable algorithms in Theorem 3 in the main text and in the Appendix D.4.

---

> ### Author Response · Authors · 2024-11-27
>
> ### **Q1.8.3**
>
> We saw [R6] together with [R8] and [R9] as a line of works demonstrating the drawbacks of MI bounds and, in this context, regarded the inherent CMI-accuracy trade-off as secondary and misinterpreted your concern on address the "limitations". Thank you very much for your clarification on your emphasis.
>
> Regarding the trade-off, we believe our Theorem 3 and omniscience technique have provided an intuitive solution/refinement to this trade-off: Even though stable $\epsilon$-learners have large (individual) (C)MI, they are quite close to a learner that has low (individual) (C)MI.
>
> We first check Theorem 3 is about $\epsilon$-learners with gentle assumptions.
> Theorem 3 shows that stable algorithms has vanishing excess generalization error. Therefore, if we assume the optimal generalization error is $0$, then Theorem 3 says stable algorithms has vanishing generalization errors, and are $\epsilon$-learners when $n = \Omega(1/\epsilon^2)$.
>
> Then, we interpret the proof and the result of Theorem 3 as the following: The stable $\epsilon$-learner (original trajectory $W\_{0:T}$) is quite close to (measured by the penalty term / loss difference, which are finally bounded  the stability) low-individual-MI learners (the omniscient trajectory $\breve{W}\_{0:T}$ with $0$ individual MI).
> This can be seen from the proof if we move the selection of the omniscient trajectory to the first step of the proof: we first select the omniscient trajectory with zero individual MI, and then measure the difference by the penalty term, which finally is bounded by the stability.
>
> Therefore, our result can be seen as an intuitive refinement to the CMI-accuracy trade-off. This refinement happens between the accuracy and a new (very rate-distortion style) information measure that combines the approximation to a low-information learner and how low-information the latter is. After replacing CMI with the new information measure, the trade-off is improved: the two components can vanish at the same time.
>
> For general $\epsilon$-learners, we have added **Theorem 4 in Appendix D.5** in the newest revision. It replaces the CMI in the information-accuracy trade-off with a rate-distortion style information measure induced by our omniscient trajectory.
> Using the same construction of the omniscient trajectory, it shows for $\epsilon$-learners that are also "$O(\epsilon)$-optimizers", the new information measure vanishes together with excess generalization risk.
> By replacing the information measure, Theorem 4 partially breaks through the trade-off.
> By the construction of omniscient trajectory and the proof, the previous intuition extends to more general learners: **for $O(\epsilon)$-optimizing $\epsilon$-learners, although they have high CMI, they are quite close to learners with low MI**.
>
> However, this result is still partial and preliminary because it assumes "$O(\epsilon)$-optimizers", i.e., the excess optimization error is at most $O(\epsilon)$.
> Nevertheless, we believe it is quite intuitive and gentle.
> Intuitively, it assumes overfitting, which is quite similar to [R6]'s Lemma 6.3 and their main results on the CMI that emphasize the correlation between the learned weights and the training set.
> In practice, learners almost always overfit. If some learner have larger training loss than testing loss, it must be quite strange. Therefore, we believe our assumption on the optimization error is quite gentle and what it excludes are mainly weird learners.
>
> Therefore, our results now provide a direct response (although still partial) to the CMI-accuracy trade-off.
>
> We have added remarks on the intuitive break-through of information-accuracy trade-off in **Lines 515-518 in Sec 5**, and included the results for $O(\epsilon)$-optimizing $\epsilon$-learners in **Lines 518-522 in Sec 5 and Theorem 4 in Appendix D.5**.
>
> ## **Q1.9: Theorem 3 is not related to flatness**
>
> Yes, we have considered and tried emphasizing the omniscient trajectory instead of the flatness.
> However, we found we must write a main story with two parallel lines, one for flatness in deep learning and one for CLB problems in the more classical setting.
> Two parts seem quite unrelated and may makes our manuscript lack of focus. (We believed there are some common reasons behind the one-fits-two solution, but we had not figured it our before we had to start writing.)
>
> As a result, we had to choose one line as the main line and the other one line as the extension (Sec 5). Since MI bounds are mainly developed to understand the data-dependent generalization under overparameterization, we chose the flatness as the main line.

---

> ### Comment · Reviewer_CXNU · 2024-11-27
>
> Thank you for your responses and the additional results.
>
> My following comments are intended to share some thoughts in a more casual style, without giving any additional pressure to the authors during the rebuttal period.
>
> I believe the use of fixed initialization should be explicitly stated, and thanks for the clarification.
>
> Regarding [R5] and Theorem 3, I did not view the similarity in insights between these results as a negative aspect of the paper. However, further clarifying the key differences (e.g., proving strategy or the flexibility of the omniscient trajectory) and revising any potential points of misinterpretation (as you've done in the revision) can help avoid confusion for readers.
>
> Regarding [R6], while I did not verify all the proof details of Theorem 4, it aligns with my experience. In the development of information-theoretic generalization bounds, we have already known that IT bounds perform exceptionally well in the realizable setting (i.e., zero empirical risk) and with bounded loss. These insights can extend to cases of low empirical risk variance, although many of these results are based on CMI bounds, see [R12, R13, R7, R14]. For example, [R13] and [R14] show that e-CMI or ld-CMI bounds can exactly characterize the generalization error (i.e. $\textrm{gen. err.} =O(\textrm{CMI})$)  in the zero training loss and 0-1 loss setting.
>
> While I agree that considering small optimization errors is closer to practical scenarios, such as deep learning, I would like to emphasize that [R6], along with [R8] and [R9], aims to extend our understanding of IT bounds beyond low empirical risk and bounded loss settings. These works demonstrate that due to the dimension-dependent nature of information measures, there always exists a learner that causes IT bounds to fail, even if such learners are extremely weird.
>
> [R12] Mahdi Haghifam, et al. Towards a unified information-theoretic framework for generalization. NeurIPS 2021.
>
> [R13] Mahdi Haghifam, et al. Understanding generalization via leave-one-out conditional mutual information. ISIT 2022.
>
> [R14] Ziqiao Wang and Yongyi Mao. Tighter Information-Theoretic Generalization Bounds from Supersamples. ICML 2023
>
>
> I sincerely appreciate the authors’ efforts in addressing my concerns, and I am increasing my score to 8 accordingly.

---

> > ### Author Response · Authors · 2024-11-28
> >
> > We sincerely thank the reviewer for the constructive feedbacks and the detailed clarifications. The discussion with the reviewer is very inspiring, encouraging us to explore more results and build deeper connection with existing works. Thank you again for the insightful remarks in your latest comments. We also thank the reviewer very much for increasing the score.
> >
> > If you have further questions or remarks, please do not hesitate to post them. We are very willing to continue the constructive and inspiring discussion.

---

### Meta-Review · Area_Chair_CTXY · 2024-12-16

**Metareview:**

The work improves on recent advances in information theoretic generalization bounds by using the flatness geometry typically associated with SGD training. There is novelty in the technical approach used, as noted by all the reviewers, and empirical results are considerable improvements over the state-of-the-art.

There was wide ranging discussions between all the reviewers and the authors. Several aspects related to the techniques used and the relationship of the current work with recent literature was brought up. The manuscript will benefit from capturing some of these discussions as remarks or in the appendix.

**Additional Comments On Reviewer Discussion:**

There was wide ranging discussions between all the reviewers and the authors. The reviewers and the authors were actively engaged.

---

### Decision · Program_Chairs · 2025-01-22

Accept (Poster)